# Collective dynamical regimes predict invasion success and impacts in microbial communities

Jiliang Hu ®[1,2], Matthieu Barbier ®[3,4], Guy Bunin ®[5] & Jeff Gore ®[1]✉

The outcomes of ecological invasions may depend on either characteristics of the invading species or attributes of the resident community. Here we use a combination of experiments and theory to show that the interplay between dynamics, interaction strength and diversity determine the invasion outcome in microbial communities. We find that the communities with fluctuating species abundances are more invasible and diverse than stable communities, leading to a positive diversity–invasibility relationship among communities assembled in the same environment. As predicted by theory, increasing interspecies interaction strength and species pool size leads to a decrease of invasion probability in our experiment. Our results show a positive correspondence between invasibility and survival fraction of resident species across all conditions. Communities composed of strongly interacting species can exhibit an emergent priority effect in which invader species are less likely to colonize than species in the original pool. However, if an invasion is successful, its ecological effects on the resident community are greater when interspecies interactions are strong. Our findings provide a unified perspective on the diversity–invasibility debate by showing that invasibility and invasion effect are emergent properties of interacting species, which can be predicted by simple community-level features.

Ecological invasions, characterized by the spread of non-native species into new environments, have important consequences for biodiversity, ecosystem function and habitat resilience[1]. Over decades, ecologists have sought to unravel the myriad factors influencing why some species invade successfully and why some of those have large impacts on resident species communities, while others do not. Ecologists have posited a range of determinants, from the fitness and adaptability of the invaders to the resilience and composition of native communities[2–4]. Among studies focusing on the invader species, many have sought to identify traits, such as growth and dispersal strategies, that may shape invasion outcomes[5]. Others have emphasized the role of the invaders' initial population size in the likelihood of establishment and spread[6,7]. Yet others have emphasized interactions with resident species; for example, the enemy release hypothesis that invasive species often succeed in new environments because they lack consumers or pathogens[8]. This has led to research on how properties of resident communities as a whole can determine the invasion outcome. For instance, the biotic resistance hypothesis suggests that communities with high native biodiversity are more resistant to invasion than less diverse communities, due to more efficient resource use or presence of natural enemies, but it is not consistently supported by empirical results[9–12]. Beyond the characteristics of invader species and resident communities, environmental conditions have been shown to play a crucial role in shaping the invasion outcome[1]. For example, theories

[1]Physics of Living Systems, Department of Physics, Massachusetts Institute of Technology, Cambridge, MA, USA. [2]Department of Mechanical Engineering, Massachusetts Institute of Technology, Cambridge, MA, USA. [3]CIRAD, UMR PHIM, Montpellier, France. [4]PHIM Plant Health Institute, Montpellier University, CIRAD, INRAE, Institut Agro, IRD, Montpellier, France. [5]Department of Physics, Technion—Israel Institute of Technology, Haifa, Israel. ✉e-mail: gore@mit.edu

such as the storage effect and the fluctuating resource availability hypothesis posit that environmental disturbances and fluctuations might favour invader species in specific periods[13–15].

More recently, the issue of ecological invasion has become salient in the study of microbial communities, ranging from soil and aquatic ecosystems to the human body[16–22]. These invasions can have profound impacts on ecosystem services and human health[16,17,19,20]. Pathogenic microorganisms can invade host-associated microbial communities, leading to infections and disease[19,23,24]. For example, the invasion of the pathogenic microorganism *Clostridium difficile* into the gut microbiota can lead to severe diseases, including diarrhoea and colitis[23,25]. Understanding the mechanisms underlying invasion success and ecological consequences can help to inform strategies for disease prevention, as well as the development of targeted therapies to control invasive pathogens[25,26]. Similar to larger-scale ecological systems, it has been suggested that microbial communities with higher diversity (number of species) are less likely to be invaded because diverse resident species may occupy all available niches by consuming all resources, leaving less room for invaders[18,27–29]. Furthermore, it was shown that facilitative and competitive interactions between microorganisms can favour and prevent successful invasions, respectively[27,30–32]. Parallel to observations in macroorganisms, external disruptions, such as antibiotic interventions or nutrient level shifts, can heighten the vulnerability of microbial communities to invasions[16,33–35].

While research in microbial invasions has made important strides, it remains unclear what characteristics of resident communities determine the success and impacts of an invasion[17,18,36,37]. Species diversity is an easily measured indicator, but its relationship to invasibility may not be straightforward, whereas species interactions are probably important but often difficult to quantify. A rarely emphasized property is the dynamics of the resident community: are the species abundances constant over time, consistent with a stable state or are they deterministically fluctuating? It is not obvious that we can characterize dynamics at the level of the community; yet, building upon the groundbreaking work of Robert May, ecologists have explored the possibility of community-wide emergent dynamics, which can be classified into only a few qualitatively distinct regimes and predicted from a few macroscopic parameters[10,38–43]. In a recent study[40], we experimentally assembled communities from various pools of microbial species in different conditions and confirmed that simple community-level features, including species pool size and interspecies interaction strength, determined distinct dynamical regimes characterized by the fraction of surviving species and the emergence of deterministic abundance fluctuations over time. As species pool size and strength of interactions increase, we found that microbial ecosystems transition between three distinct dynamical phases, from a stable equilibrium in which all species coexist to partial coexistence to the emergence of persistent fluctuations in species abundances[40].

Here we perform invasion experiments in diverse assembled microbial communities and observe that the foremost predictor of invasion outcomes appears to be the dynamical state of the resident community. We then use a combination of experiments and theory, exploring several dynamical regimes and spanning their control parameters (species pool size and interaction strength) to show that, taken together, they explain many features of invasibility and invasion effects. Communities of weakly interacting species reach a stable composition, where a fraction of the initial species pool survives, and further invasions display the same fraction of successes, only weakly perturbing resident species. Larger species pools and stronger interactions can give rise to fluctuating states, where species abundances fluctuate over time. We found that these fluctuating communities are more invasible and diverse than stable communities, leading to a positive diversity–invasibility relationship among communities assembled in the same environment and the same species pool size. These deterministic fluctuations in communities are chaotic dynamics or limit cycle

oscillations driven by interspecies interactions, rather than stochastic fluctuations driven by demographic noise. Finally, communities with strong interactions can also reach alternative stable states where invasions succeed more rarely than predicted by survival fraction, but strongly impact the resident community when they do. The lower invasion probability compared to the survival fraction suggests a priority effect, whereby earlier invaders preclude later ones from growing from small abundances, leading to situations where the sequence and timing of species introduction can influence invasion success[10,44,45].

Studying invasions through the prism of community-wide dynamical regimes allows us to connect several strands of ecological thinking, regarding what counts as a successful invasion, when factors such as population size and history matter, and what consequences invasions have on resident community structure and functioning[46,47]. Furthermore, it helps clarify the hypothesis that increased community diversity results in reduced invasion probability due to fewer available niches[18,27–29]. Within fixed conditions (given the same initial species pool size and environment), more diverse communities tend to be found in fluctuating states, and are actually more likely to be invaded. Depending on how we change conditions – for example, increasing species pool or reducing interaction strength – diversity may positively or negatively correlate with invasibility, providing one explanation for inconsistent observations[48–50]. Throughout these different conditions, however, the fraction of surviving species during the initial community assembly remains a better predictor of invasibility, displaying a universal positive correspondence with invasibility across all conditions, modulated by the presence of priority effects. Our results demonstrate that both invasibility and invasion effects are emergent properties, shaped by the interactions of resident species, which can be predicted by simple community-level features.

## Results and discussion

To experimentally characterize invasions in microbial communities, we built 17 different synthetic communities of size $S = 20$ using a library of 80 bacterial isolates from river and terrestrial environments (Fig. 1a and Supplementary Fig. 1). We exposed each community to daily cycles of growth and dilution into fresh media, with dispersal from the species pool ($S = 20$) to mimic species dispersal in natural habitats (Fig. 1a). After 6 days of culturing, we exposed each community to an invader species (Fig. 1a) and we continued to culture the communities for another 6 days with dispersal of all species on each dilution cycle (Fig. 1a,b). For each resident community, we performed seven to nine independent invasion tests with different randomly chosen invader species on day 6, and monitored the growth of the invader and resident species (Fig. 1b). Analysing species abundances through 16S sequencing, we found that 7% ± 2% of invasion tests were successful (relative invader abundance exceeded extinction threshold $8 \times 10^{-4}$ on the last day 12; the rationale behind the choice of extinction threshold is explained in the Supplementary Materials and Methods) (Fig. 1c and Supplementary Figs. 2 and 26). Although diverse ecosystems are typically thought to be more resistant to invaders[18,27–29], our experimental results display a significant ($P = 0.036$) positive correlation between invasion probability and community diversity, where the diversity is defined as the number of species that survived the assembly process over 6 days (correlation coefficient = 0.5; Fig. 1c). Among communities of low diversity (two to five surviving species), only 3% ± 2% of invasions were successful, whereas among communities of high diversity (six to nine surviving species) 13% ± 5% of invasions were successful. Throughout the manuscript, we used the standard error of the mean (s.e.m.) as the measure of dispersion. We therefore find that less diverse communities may resist invasions better than highly diverse ones under the same initial species pool size and nutrient conditions.

To better understand why the more diverse communities were more invasible, we next quantified the dynamics of the resident communities before invasion. We found that just under half (8/17) the

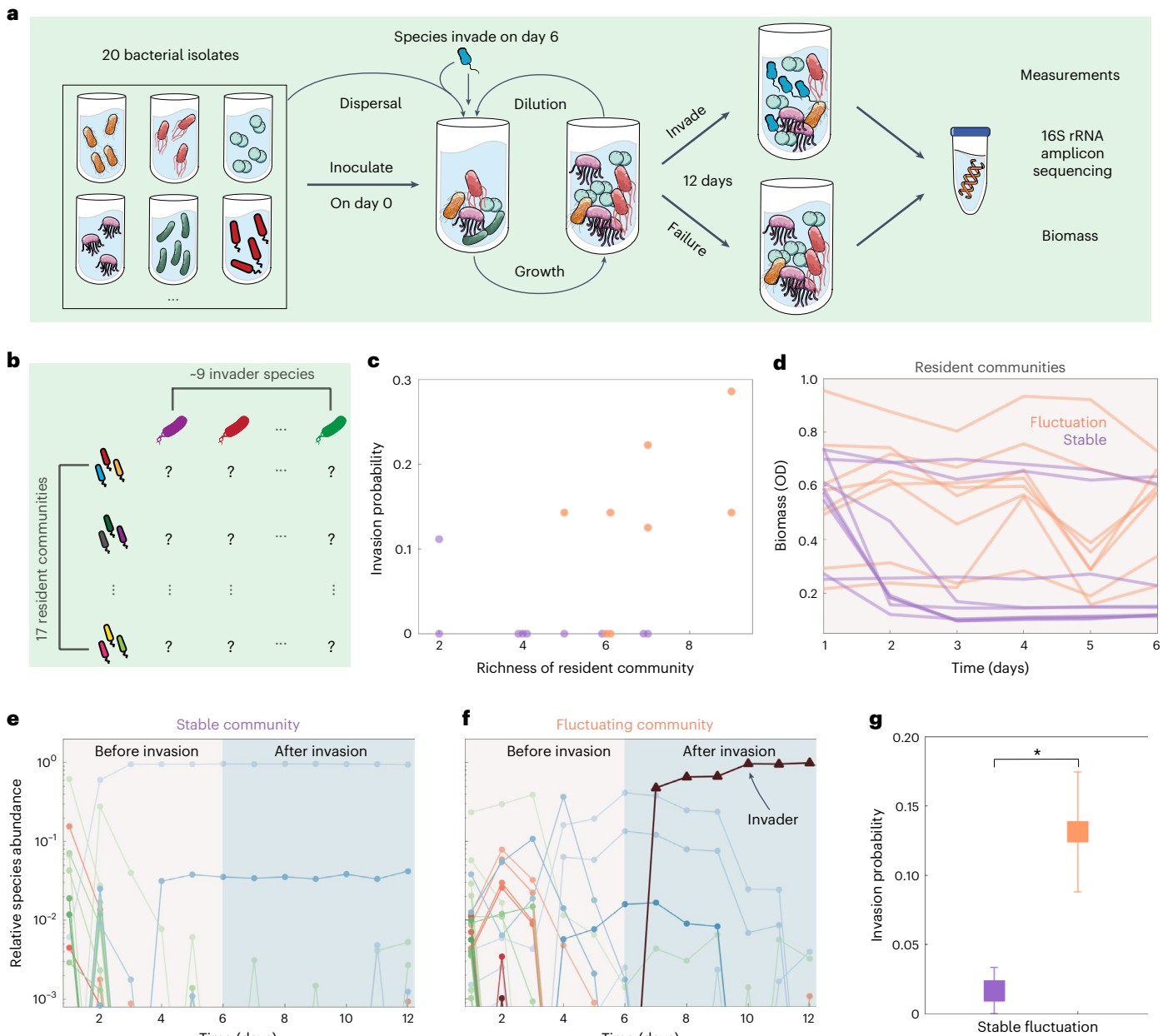

**Fig. 1 | Experiments using synthetic microbial communities.** The invasion probability in fluctuating communities is higher than stable ones, leading to a positive diversity–invasibility relationship. **a**, We used a library of bacteria to generate different synthetic communities with $S = 20$ species in the pool (under 'high' nutrient conditions; Methods). Communities underwent serial daily dilutions with additional dispersal from the pool. We introduced invader species to the resident communities on day 6 and continued to apply daily dispersal of invaders. Community composition and total biomass were monitored via 16S sequencing and optical density (OD). **b**, We formed 17 resident communities with different sets of species ($S = 20$). We added invader species outside the pool into the resident communities on day 6, and then measured the community compositions and biomass on day 12 to determine the outcome and effect of the invasions. **c**, The invasion probability in resident communities positively correlate with their richness (correlation coefficient = 0.5, $P = 0.047$) under the same species pool size and nutrient conditions. **d**, Out of the 17 resident communities, 8 reach fluctuation in biomass (orange) and the other 9 communities reach stable states (purple). **e**, Representative time course of relative species abundance via 16S sequencing show that the stable community was not invaded. **f**, The representative time course of relative species abundance shows that the invader successfully invades and grows in the fluctuating community. **g**, The invasion probability in fluctuating resident communities is statistically higher than that of stable communities (two independent samples two-sided Student's $t$-test, $P = 0.016$, the number of invasion tests is $n = 61$ (60) for fluctuating (stable) communities). Error bars, s.e.m.

resident communities displayed persistent and deterministic fluctuations in biomass and species composition, with the remainder reaching stable community states (Fig. 1d–f and Supplementary Figs. 3–12). We found that biomass fluctuations were highly correlated with species abundance fluctuations (Supplementary Fig. 12) and the classification of stable and fluctuating communities was robust to different methods (Supplementary Fig. 12). These deterministic fluctuations

in communities are chaotic dynamics or limit cycle oscillations driven by interspecies interactions, rather than stochastic fluctuations driven by demographic noise, because of the large population size regime in this study (Supplementary Materials and Methods). Consistent with our previous results, we found that the diversity of fluctuating communities is approximately twice the diversity in stable communities (Fig. 1c)[40]. Given this higher diversity in fluctuating communities, we

next analysed the invasibility of communities separately for the stable and fluctuating communities to determine if this could be driving the positive diversity–invasibility relationship that we observed. Indeed, we detected eight successful invasions out of 61 invasion tests to fluctuating communities, while there was only one single successful invasion out of 60 invasion tests to stable communities (Supplementary Fig. 2). Our results therefore show that the probability to successfully invade fluctuating communities (13% ± 4%) is statistically about eightfold larger than the probability of invading stable communities (1.7% ± 1.7%) (Fig. 1g). Our experimental tests of invasion demonstrate that, for fixed environment and species pool size, more diverse communities are more invasible because fluctuating communities are both more diverse and more susceptible to invasion. However, we will show later that, when species pool size or nutrient concentration is varied, this relationship does not always hold. This increased invasibility of fluctuating communities can be interpreted through the lens of niche theory, where fluctuating communities create fluctuating niche availability for invader species[13]. Temporal fluctuations in resource availability and environmental conditions allow invaders to exploit niches that may not be consistently available in stable communities[13–15].

To gain insight into these surprising relationships between diversity, stability and invasibility, we next studied invasions in the well-known generalized Lotka–Volterra (gLV) model, modified to include dispersal from a species pool:

$$\frac{dN_i}{dt} = N_i\left(1 - \sum_{j=1}^{S} \alpha_{ij}N_j\right) + D \qquad (1)$$

where $N_i$ ($N_j$) is the abundance of species $i$ ($j$) (normalized to its carrying capacity), $t$ is the time, $\alpha_{ij}$ is the interaction strength that captures how strongly species $j$ inhibits species $i$ (with self-regulation $\alpha_{ii} = 1$) and $D$ is the dispersal rate, which is set to $D = 10^{-5}$ (Supplementary Figs. 24 and 25). We simulated the dynamics of communities with different species pool sizes $S$ and competitive interaction matrices because competition is the dominant interaction type in our experiments[40]. We sampled the interaction strength from a uniform distribution $U[0,2 <\alpha_{ij}>]$, where $<\alpha_{ij}>$ is the mean interaction strength between species (predictions of this model are insensitive to the particular distribution chosen[40]). Modelling species interactions as a random interaction network captures species heterogeneity without assuming any particular community structure[10,38–40]. We introduced invaders into resident communities at $t = 10^3$ and continued to simulate the dynamics until $t = 2 \times 10^3$ to determine to invasion outcome.

Our simulations revealed a wide range of dynamics and invasion outcomes under strong interaction strength between species (Fig. 2a and Supplementary Fig. 31). Some successful invasions cause dramatic effects on the structures of resident communities, whereas other invasions only yield weak change in communities (Fig. 2a). Consistent with our experimental results (Fig. 1c,g), we found a positive correlation between invasion probability and richness (number of resident species coexisting before invasion) (Fig. 2b), which is because fluctuating communities exhibit larger invasion probability than stable communities under the same conditions (Fig. 2c). Our simulation results with the Lotka–Volterra model also predict that the invasion probability decreases when mean interaction strength $<\alpha_{ij}>$ and the species pool size $S$ increase (Fig. 2d–f). It is important to note that although fluctuating communities exhibit larger invasion probability than stable communities under the same conditions, stable communities can still yield larger invasion probability under weaker interaction strength $<\alpha_{ij}>$ or smaller species pool size $S$ (Fig. 2d–f). If we interpret these phenomenological interactions in terms of niche theory and resource competition[51], stronger interaction strength corresponds to larger niche overlap and greater resource consumption, making it harder for invaders to establish. Similarly, a larger species pool increases the total interaction (more niche overlap) between community species and

invader species, thereby inhibiting invasion more strongly[51]. We also developed a model that integrates explicit pH-mediated growth with the Lotka–Volterra framework, allowing interactions to be expressed as a function of pH modification. This new model suggests that the presence of pH effects increases the effective interspecies interaction strengths, but otherwise yields predictions similar to those of the canonical Lotka–Volterra model (Supplementary Fig. 23). In addition, we found that neither serial dilutions nor the existence of positive (facilitative) interspecies interactions qualitatively affects this result (Supplementary Figs. 28–30). The Lotka–Volterra model therefore explains why our diverse and fluctuating communities are susceptible to species invasion and makes new predictions regarding how invasibility would change with the size of the species pool and the strength of interspecies interactions (Fig. 2d–f).

To experimentally test the predicted dependence of invasion probability on interaction strength and species pool size, we tuned the interspecies interaction strength by tuning the concentration of supplemented glucose and urea in the culture medium[40,52,53]. As discussed in our previous work[40,52,53], increasing the concentration of supplemented glucose and urea leads to stronger strength of competitive interactions between bacterial species due to extensive modification of the media (for example, pH). We measured the invasion of about nine invader species to 15 synthetic resident communities under low nutrient conditions (weak interaction) and 25 communities under high nutrient (strong interaction) conditions. Consistent with our theoretical predictions, we found that increasing interaction strength leads to a decrease of invasion probability in resident communities (Fig. 3a). Specifically, the invasion probability was 56% ± 8% in low nutrient conditions (weak interaction), eightfold higher than the invasion probability of 7% ± 2% observed in high nutrient conditions (strong interaction) (Fig. 3a). We also decreased the species pool size from $S = 20$ to $S = 12$ and found that invasion probability increased to 85% ± 6% from 56% ± 8% in low nutrient conditions (weak interaction) (Fig. 3b), consistent with our theoretical predictions. We only observed stable communities under low nutrients (weak interaction) because fluctuations only emerge when species pool size and interaction strength are large enough to cross the stability boundary[40]. Our theory and experiment both indicate that increasing either interaction strength or species pool size leads to a decrease in community invasibility[10,18,27–29].

To unify different invasibility-richness relationships in the experiments depending upon how the richness is changed (by varying interaction strength, species pool size or dynamical regime) (Supplementary Fig. 13), we next analysed the dependence of invasion probability on the survival fraction of species in resident communities, defined as the fraction of species in the initial pool that survive the assembly process (on day 6 before invasion). The results show a strongly positive correlation of invasibility with survival fraction, where the correlation coefficient is 0.77 ($P = 3.4 \times 10^{-7}$) (Fig. 3c). Microbial communities cultured in low nutrient (weak interaction) media display both a larger invasion probability and larger survival fraction than communities under high nutrient (strong interaction) (Fig. 3c). Furthermore, fluctuating communities, which are easier to be successfully invaded, also exhibit larger survival fraction than stable communities under the same conditions (Figs. 1c and 3c). These results demonstrate that the survival fraction can serve as a unifying predictor of the invasibility of a resident community. Although it has been suggested that microbial communities with higher diversity are less likely to be invaded because they leave fewer available niches for invaders[18,27–29], our results indicate that this is only true when the diversity is increased by increasing the size of the species pool (Figs. 1c and 3c). However, if diversity is modulated by a change in interaction strength or stability, then more diverse communities are instead more invasible.

Despite the observed correspondence between invasion probability and survival fraction, we find that invasion probability under high nutrient (strong interaction) conditions is generally lower than

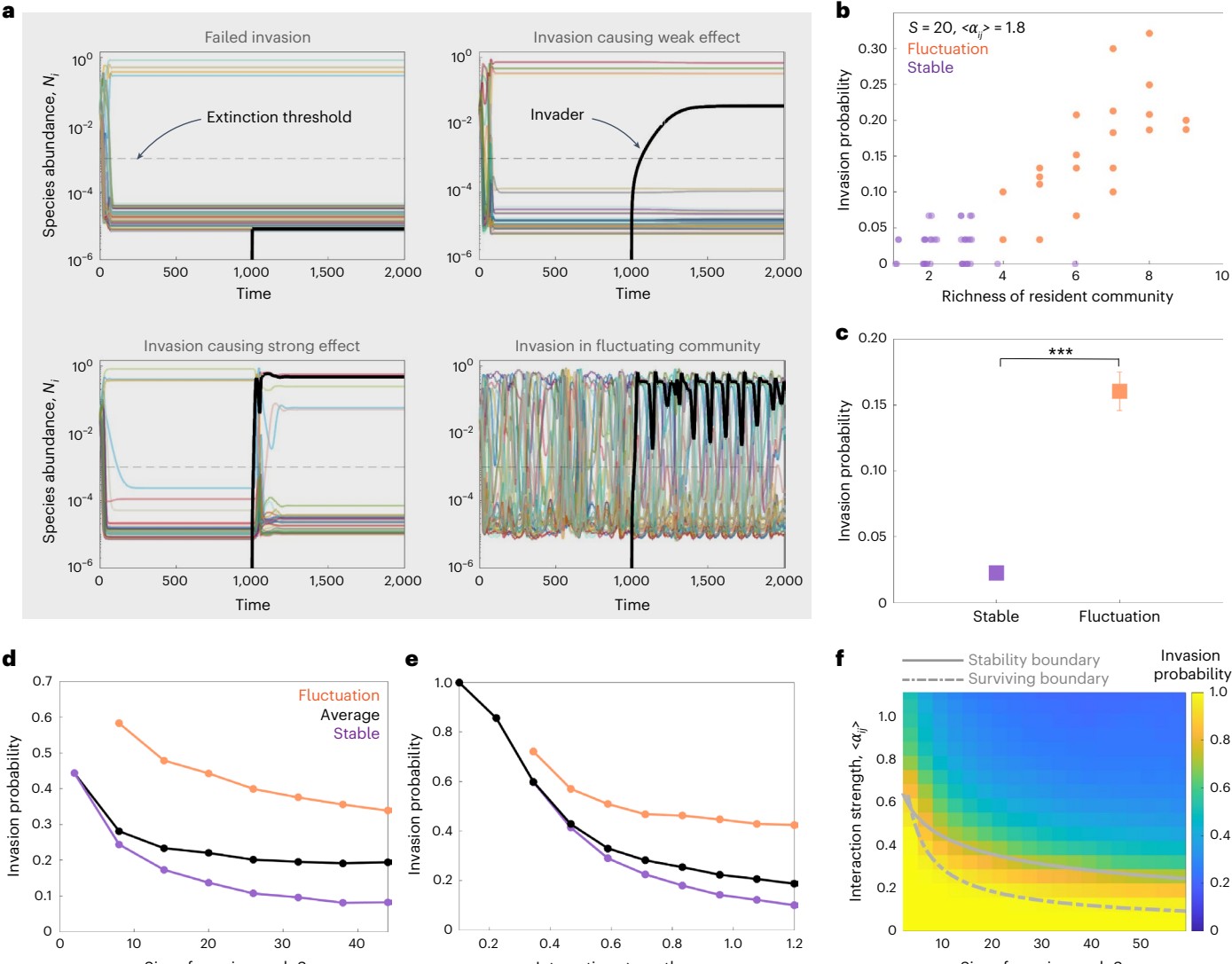

**Fig. 2 | The Lotka–Volterra model of invasion probability.** The model predicts a decrease of invasion probability when stability, interaction strength and species pool size of resident communities increase. **a**, Representative time series of species abundance in simulation show diverse invasion dynamics and outcome: invader species failed to grow in the community (top left, the black curve represents invader); an invader grows and only causes small effect on community composition (top right); an invader successfully invades and causes large change on community composition (bottom left); invasion to a fluctuating resident community (bottom right) ($<\alpha_{ij}>$ =0.6, $S$ = 32). The unit of time is defined as $1/r$, where $r$ represents the species' growth rate, set to 1 in all simulations. **b**, Consistent with experiments (Fig. 1c), the invasion probability of simulated resident communities positively correlates with their richness, which arises because fluctuating communities are more diverse and more invasible under

the same species pool size and average interaction strength. **c**, The invasion probability for fluctuating resident communities ($n$ = 8) is statistically higher than that for stable communities ($n$ = 9) ($P$ = 6.2 × 10⁻²⁹). Two-sided Student's $t$-test was performed. Error bars, s.e.m. **d**, Increasing species pool size leads to a decrease in invasion probability. Fluctuating communities (orange points) exhibit higher invasion probability than stable communities (purple points). **e**, Increasing interaction strength leads to a decrease in invasion probability. **f**, Increasing species pool size and interaction strength leads to a decrease in invasion probability. The communities experience the extinction of species and loss of stability when crossing the dashed grey line (surviving boundary) and solid grey line (stability boundary), respectively. The curves and colour maps depict the mean value over 1,000 simulations.

the survival fraction (most data points lie below the diagonal line in Fig. 3c). This discrepancy suggests the influence of priority effects or alternative stable states, where the order of species arrival significantly impacts community structure[44,45]. Specifically, early-arriving species in strongly interacting communities may establish dominance, reducing the likelihood of later-arriving invaders to establish successfully. Further discussion on how priority effects and alternative stable states explain this reduced invasion probability is provided in Supplementary Fig. 22 (refs. 10,40).

To understand the reason for different diversity–invasibility relationships when varying interaction strength, species pool size

or dynamical regime (Figs. 1c, 2 and 3), we sampled resident communities along different paths on the phase diagram (Fig. 2f). We simulated invasions to these resident communities and found different diversity–invasibility relationship along different paths (Fig. 4a). The results show a positive diversity–invasibility relationship when only varying interaction strength while fixing species pool size or randomly sampling communities under the same parameters of species pool size and interactions (Fig. 4a). On the contrary, a reversed negative or non-monotonic diversity–invasibility relationship was observed when varying species pool size while fixing interaction strength (Fig. 4a). Despite these conflicting diversity–invasibility relationships, after

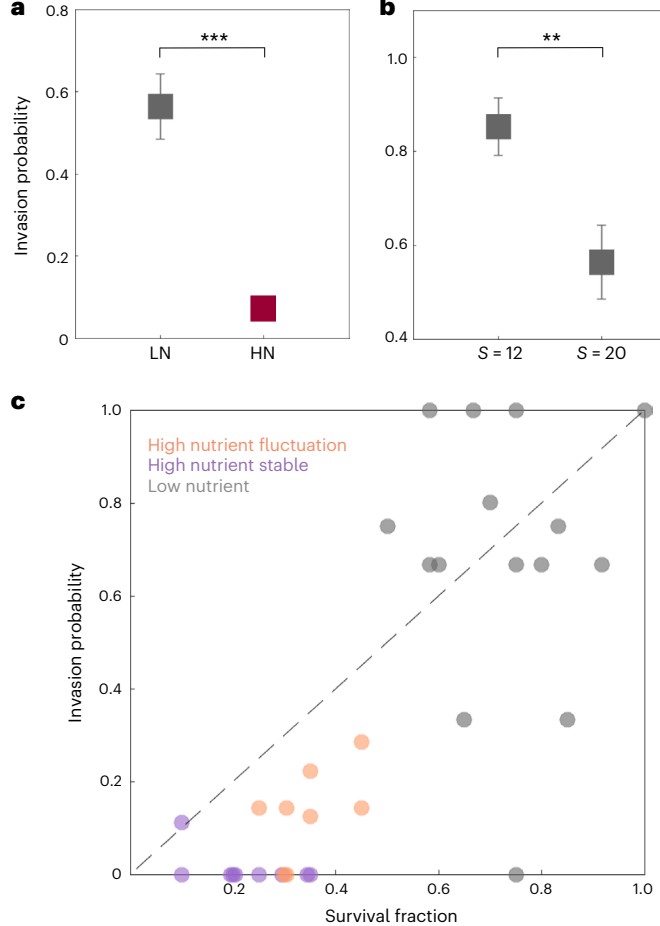

**Fig. 3 | Increasing interaction strength and species pool size leads to lower invasion probability of resident communities in experiment. a**, The invasions to resident communities under low nutrient (weak interaction) exhibit statistically higher invasion probability than communities under high nutrient (strong interaction) ($P = 4.5 \times 10^{-13}$, the number of invasion tests is $n = 120$ (39) for high (low) nutrient). Two-sided Student's $t$-test was performed. **b**, The invasions to resident communities under smaller species pool size ($S = 12$) exhibit statistically higher invasion probability than for communities under larger species pool size ($S = 20$) ($P = 0.007$, the number of invasion tests is $n = 39$ (34) for $S = 20$ (12), all communities were cultured under low nutrient, weak interaction). Two-sided Student's $t$-test was performed. Error bars, s.e.m. **c**, The invasion probability positively correlates with survival fraction (before invasion) across different communities and nutrient conditions (each point represents one community; correlation coefficient = 0.77, $P = 3.4 \times 10^{-7}$). The points corresponding to communities under high nutrient (strong interaction) are below the diagonal line, showing the invasion probability of communities under high nutrient (strong interaction) are generally smaller than their survival fraction, which indicates the priority effect under strong interaction strength. HN, high nutrient; LN, low nutrient.

scaling richness with species pool size to get the survival fraction, we found that all communities collapsed to a universal line in which the invasion probability is approximately equal to the survival fraction (Fig. 4b). The deviation from the exact collapse in small survival fraction regime (bottom left of Fig. 4b) indicates priority effect under strong interaction. Our results indicate that survival fraction determines invasibility, whereas diversity–invasibility relationship can be qualitatively different depending upon the origin of different diversity in different communities.

The emergence of the priority effect in experiments (Fig. 3c) was also found in the Lotka–Volterra model under different regimes of interaction strength and species pool size. We quantified the priority

effect by calculating the difference between survival fraction of resident species and the invasion probability of species that invade after the resident communities have assembled, where the difference was normalized by survival fraction (Fig. 4c). We found that there is no clear priority effect in the small species pool size and weak interaction regime, where species in the initial pool and invader species display similar probability of colonizing in the communities (Fig. 4c). Consistent with our experimental results, increasing species pool size and interaction strength in the model leads to the emergence of priority effect in the phase where communities reach fluctuation or alternative stable states (Fig. 4c). Simulations indicate that the priority effect originated from alternative stable states or limits cycle oscillations in the strongly interacting phase, whereas chaotic fluctuations display no significant priority effect (Supplementary Fig. 14), which can be explained by its ergodicity[41,43,54] (technical discussion in Supplementary Fig. 14).

We also investigated the idea that successful invasions can cause strong or weak effects on resident community structure depending on how invaders interact with resident species[10,30,31]. Our simulations predict that invasions have a larger impact on the composition of resident communities when the interactions are stronger, where the invasion effect is quantified as the proportion of change in surviving species before the invasion ($t = 10^3$) and after the invasion ($t = 2 \times 10^3$) (invasion effect = 1 − (number of overlapping species/total number of species)) (Fig. 4d). To understand the effect of a successful invasion in the experiment, we analysed the change of biomass and species composition before and after the invasions (Fig. 5). The community biomass displays relatively small changes after invasion under weak interactions (low nutrient regime, inset of Fig. 5a and Supplementary Figs. 6 and 7). In the strong interaction regime (high nutrient), we found that stable communities typically transitioned from low biomass states to high biomass states after successful invasions, whereas the biomass of fluctuating communities continued to fluctuate over a similar range (Fig. 5a–c and Supplementary Figs. 3–5). Averaging across both stable and fluctuating communities, we found that community biomass under strong interaction displayed a larger fold change ($2.9 \pm 0.8$) after successful invasion than those under weak interaction ($1.15 \pm 0.03$) (Fig. 5c and Supplementary Fig. 27). We calculated the invasion effect in experiment by comparing surviving species between invaded communities and control communities without adding invaders (Supplementary Figs. 15–18). This analysis on surviving species indicated that successful invasions cause stronger change in the community composition under strong interaction (invasion effect = $53\% \pm 6\%$) than weak interaction ($39\% \pm 2\%$) (Fig. 5d), which is consistent with the simulation results with the Lotka–Volterra model (Fig. 4d). The effect size on community composition caused by increasing from low nutrient (weak interaction) to high nutrient (strong interaction) conditions is 0.14, with a 95% confidence interval of [0.021, 0.259]. We also observed a weak positive correlation between the invasion effect and the final abundance of invaders in the experiment and simulation (Supplementary Fig. 33). The growth of invader species influences the community structure more dramatically when it has a stronger interaction with other resident species, and the strong interplay between resident species can also cause stronger secondary effects on other resident species when their abundances change[10,47].

Although our study was primarily focused on community-level properties that determine invasibility and invasion effect, we also analysed properties of the invader species that correlated with invasibility and invasion effect. Perhaps surprisingly, we did not observe a significant correlation between a species' ability to invade and that species growth in monoculture (Supplementary Fig. 19). For example, a *Pseudomonas* sp. (invader 4) and an Enterobacterales sp. (invader 7) were the two most successful invader species (16 of 35 and 6 of 11 invasions, respectively), yet displayed growth in monoculture that was typical of the group of nine invaders that were tested. In addition, a *Bacillus* sp. (invader 6) had the highest monoculture growth rate

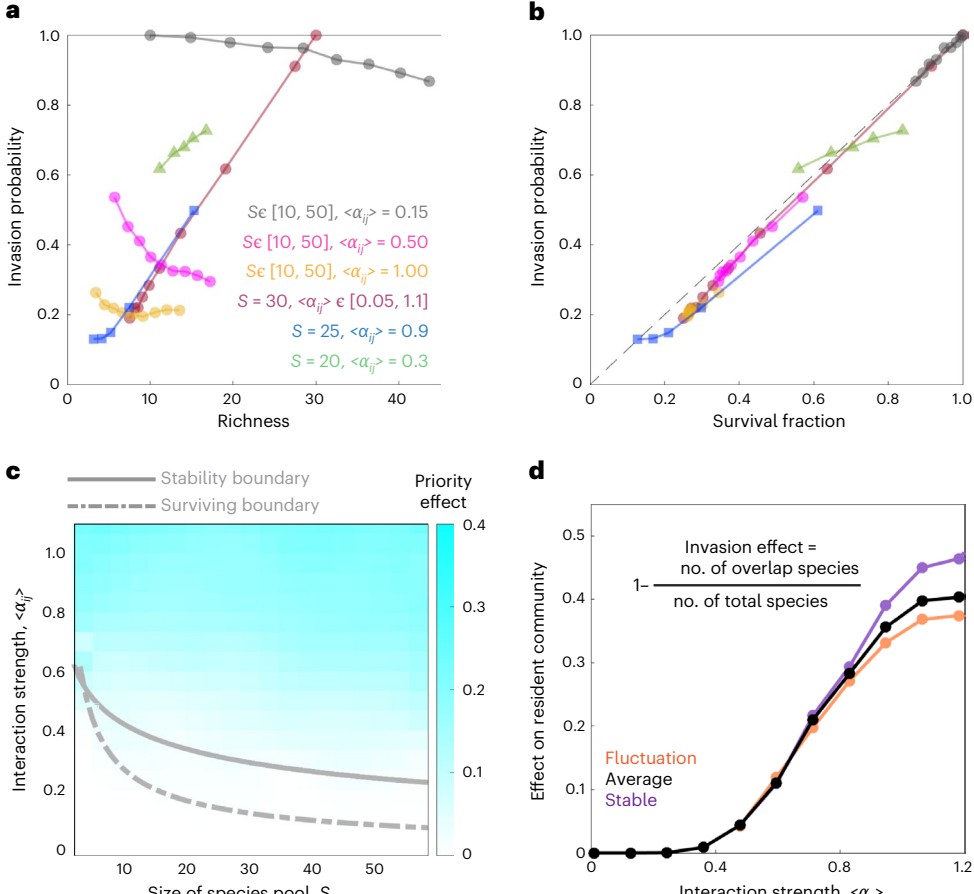

**Fig. 4 | Predictions of the Lotka–Volterra model.** The model predicts a universal correspondence between invasion probability and survival fraction, the emergence of priority effects and stronger invasion effects when increasing interaction strength. **a**, The dependence of invasion probability on final richness of resident communities is qualitatively different depending upon how the richness is changed. Invasion probability positively correlates with richness when varying interaction strength or when randomly sampling communities with a fixed species pool size and interaction strength distribution. Invasion probability can decrease with community diversity when varying species pool size. **b**, Invasion probability is approximately equal to the survival fraction of species in the resident communities, no matter how we change richness, species pool or interaction strength. **c**, Increasing species pool size and interaction strength leads to the emergence of priority effect, where the invasion probability of resident communities is smaller than their species survival fraction. The communities experience the extinction of species and loss of stability when crossing the dashed grey line (surviving boundary) and solid grey line (stability boundary), respectively. **d**, Successful invasions cause larger effect on species composition in the resident communities under stronger interaction strength. The curves depict the mean value over 1,000 simulations.

among all invaders yet was a poor invader (2/37). We also observed that a *Pseudomonas* sp. (invader 2) and a *Pedobacter* sp. (invader 3) could occasionally invade communities despite being subject to a strong Allee effect that prevented the species from growing from an initially small inoculum (Supplementary Fig. 17). Furthermore, we did not observe significant correlation between the invasion effect and invader properties either (Supplementary Fig. 20). Whether for the invasion probability of resident communities or different invaders, we found an absence of correlation between invasion probability and invasion effect (Supplementary Fig. 21). Interestingly, invaders that are phylogenetically closer to resident species tend to achieve higher post-invasion abundances (Supplementary Fig. 32). Taking these together, we therefore found that monoculture growth properties were surprisingly ineffective at predicting the success of a species as an invader.

Our findings show that invasibility and invasion effects can be statistically predicted by simple community-level features including the dynamical regime, species pool size and interaction strength of the community. As predicted by our theory, increasing community diversity leads to stronger resistance to invaders only when varying species pool size and fixing community stability and environmental conditions

(including interaction strength), which is consistent with the biotic resistance hypothesis[9–11]. We demonstrated that, when diversity is tied to increased dynamic fluctuations or reduced interaction strength, more diverse communities might instead exhibit decreased resistance to invasion (Figs. 1c and 3c). Our results emphasize that only by concurrently considering the effects of interaction strength and stability can the diversity of native communities be used to predict invasion probability; diversity alone is insufficient for such predictions. By normalizing richness with species pool size, we obtained the survival fraction, a unified predictor that closely approximates invasion probability across different conditions (Figs. 3c and 4b). This survival fraction is influenced by factors such as species pool size, interaction strength and stability (Figs. 2b and 3c). Our previous findings indicate that, on average, increasing species pool size and interaction strength both decrease the overall survival fraction[40]. We also observed that increasing species pool size and interaction strength leads to the emergence of some fluctuating communities[40]. These fluctuating communities, despite the general trend, exhibit a higher survival fraction compared to stable communities assembled from the same species pool size and nutrient concentrations (interaction strength)[40]. This suggests that while stronger interactions and larger species pools typically reduce

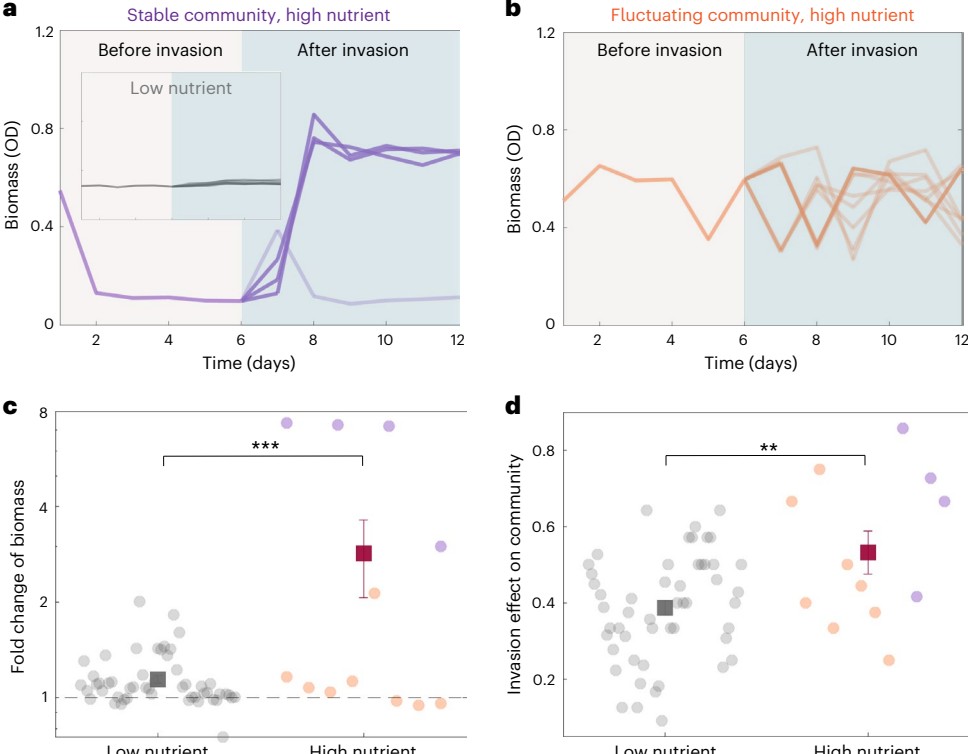

**Fig. 5 | Increasing interaction strength leads to a stronger effect on resident communities under invasion success. a**, The stable communities under high nutrient conditions (strong interaction) experienced a large increase in biomass after successful invasions (dark purple curves). Inset shows that the invasions under low nutrient (weak interaction) only cause weak effect on community biomass as compared with high nutrient (strong interaction). **b**, The time course of fluctuating community biomass under high nutrient (strong interaction) before invasion and after invasion, where dark and light orange curves represent successful and failed invasions, respectively. **c**, The invasions to resident communities under low nutrient (weak interaction) cause statistically lower fold change of biomass than communities under high nutrient (strong interaction) ($P = 2.9 \times 10^{-5}$, the number of successful invasions is $n = 51$ (11) for low (high) nutrient). The successful invasions statistically tend to increase the biomass of resident communities under different conditions. Two-sided Student's $t$-test was performed. **d**, The invasions to resident communities under low nutrient (weak interaction) cause statistically lower effect on species composition change than communities under high nutrient (strong interaction) ($P = 0.0038$, the number of invasion tests is $n = 51$ (11) for low (high) nutrient). Two-sided Student's $t$-test was performed. Error bars, s.e.m.

survival, the dynamic nature of fluctuating communities allows them to maintain higher survival fractions.

Applying the insights developed here to natural communities requires that we draw a parallel between the three recognized types of diversity in ecology—alpha, beta and gamma diversity—and the three species number variables we have investigated in our study: richness, survival fraction and species pool size[55]. Specifically, richness and species pool size can be seen as analogues for alpha (local diversity) and gamma (regional diversity) diversities, respectively. Beta diversity, defined as the ratio between regional and local diversity, is the reciprocal of the survival fraction. Consequently, our discovery of a universal positive relationship between invasibility and survival fraction suggests an overarching negative correlation between invasibility and beta diversity. While directly measuring the survival fraction in natural communities can be challenging, the ratio of local richness to regional richness in natural habitats may serve as an approximation of survival fraction[9,12,48,55], acting as a singular predictor for invasion probability. That prediction is nevertheless affected by the presence or absence of priority effects. Building upon our earlier discoveries regarding emergent phases in communities[40], our current work suggests that priority effects are most pronounced in the theoretically predicted phase of alternative stable states, matching our empirical observations of stable states found under conditions of strong interactions and a large species pool (Figs. e2 and 3c).

Beyond the deterministic fluctuations observed under large population sizes in this work, it is important to study invasions under

stochastic dynamics driven by demographic noise in subsequent research. Theory shows that demographic noise can drive stochastic transitions between alternative stable states, leading to another type of community fluctuations[56–59]. Our definition of the invasion effect in this work focuses on the impact on total biomass and community composition. We do not study the invasion effect on community function and cannot rule out the possibility that the community remains functionally unchanged as a result of functional redundancy between the invader and resident species replaced by the invader. Future research needs to include analysis of functional traits and ecosystem processes to fully understand the functional impact of invasions.

Our invasion experiments in synthetic microbial communities under controlled conditions have shown that, before any other feature of invader or resident species, the qualitative dynamical regime of the resident community is a central factor that informs all other predictions. The distinct regimes that we found, and the relationships between various predictions, were all compatible with a theory governed only by a few community-level parameters of (pool) diversity and interaction strength. Future work is necessary to determine whether these community-level features can predict invasion outcomes across spatiotemporal scales, environmental conditions and organism types.

## Methods
### Microbial community construction
We constructed a diverse microbial library of 80 bacterial isolates from soil, tree leaves and Charles River water samples. This library includes

isolates from five phyla: Proteobacteria, Firmicutes, Bacteroidota, Actinobacteriota and Cyanobacteria. For each experimental community, species were randomly chosen from this library, with species pool sizes varying across conditions to test the impact on invasibility and community stability. All bacterial isolates were precultured in base medium (BM) before constructing synthetic communities.

## Culturing conditions

We used two nutrient conditions: low nutrient (low interaction strength) and high nutrient (high interaction strength). The low nutrient BM consisted of $1\,g\,l^{-1}$ of yeast extract, $1\,g\,l^{-1}$ of soytone, 10 mM sodium phosphate and trace elements. The high nutrient medium was supplemented with $5\,g\,l^{-1}$ of glucose and $4\,g\,l^{-1}$ of urea to increase interaction strength by amplifying resource competition and promoting environmental pH fluctuations.

All communities were incubated in 96-deep-well plates at 30 °C with constant shaking at 1,200 rpm. To minimize evaporation, plates were kept in acrylic boxes. Each day, communities underwent a 30-fold serial dilution in fresh medium and dispersal from species pools at a rate of $10^{-5}$, applied to mimic natural dispersal events and maintain community diversity.

## Experimental design for invasion studies

Invasions were introduced into each community on day 6 after pre-establishing community structures through six daily cycles of growth and dispersal. For each invasion test, we selected one invader species from the library and added it to the resident community at a $10^{-3}$ ratio of its monoculture to resident volume. The communities were monitored over another 6 days post-invasion with continued daily dilution cycles, measuring invasion success by tracking changes in species abundances and community composition.

## Biomass and species abundance measurements

Biomass was measured daily using optical density (OD) at 600 nm on a Varioskan Flash plate reader, with 150 µl samples taken from each well. The remaining samples were stored at −80 °C for DNA extraction. Community compositions were monitored through 16S ribosomal RNA sequencing, performed at the Environmental Sample Preparation and Sequencing Facility at Argonne National Laboratory. We used the DADA2 pipeline to obtain amplicon sequence variants (ASVs), with taxonomic identities assigned through the SILVA database (v.132). Species richness was defined as the number of ASVs with relative abundances ≥0.08%.

## Data analysis for invasion success and community dynamics

Invasion success was defined by the final relative abundance of the invader species, with a threshold of 0.08% as a cutoff for successful establishment. For communities reaching steady states, fluctuations in species abundance and biomass over time were categorized as stable or fluctuating. A standard deviation threshold of 0.05 OD across days 4 to 6 was used to distinguish these states.

## Lotka–Volterra model simulations

We modelled community dynamics using a gLV framework with species pool dispersal rates set at $10^{-5}$. The model includes dispersal from the external species pool to simulate natural community dynamics, with interaction strength $\alpha_{ij}$ sampled from a uniform distribution $U[0, 2<\alpha_{ij}>]$ to reflect interspecies competition. The survival fraction and stability of each simulated community were analysed on the basis of abundance fluctuations around a threshold set at $8 \times 10^{-4}$, aligning with experimental detection limits.

## Statistical analysis

Statistical comparisons of invasion probability, community diversity and survival fraction were performed using Student's $t$-tests or correlation analyses, as appropriate. For figures requiring error bars, the mean and s.e.m. are presented, with specific test details provided in each legend. All simulations were run in MATLAB using Runge–Kutta numerical integration with a step size of 0.05, ensuring consistent results across 1,000 simulations for each parameter set.

## Reporting summary

Further information on research design is available in the Nature Portfolio Reporting Summary linked to this article.

## Data availability

Isolates and communities are available upon request. All data are available in the Supplementary Information and via Dryad at https://doi.org/10.5061/dryad.8gtht76xz (ref. 60).

## Code availability

All codes used for simulation and analysis in this publication are available via GitHub at https://github.com/Jiliang-Hu/Collective-dynamical-regimes-predict-invasion.

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

# Article

## Acknowledgements

J.G. acknowledges funding support from the Schmidt Polymath Award and the Sloan Foundation.

## Author contributions

J.H. and J.G. conceived the study. J.H. performed the experiments and the theoretical modelling. All authors analysed the data and wrote the manuscript.

## Competing interests

The authors declare no competing interests.

## Additional information

**Extended data** is available for this paper at https://doi.org/10.1038/s41559-024-02618-y.

**Correspondence and requests for materials** should be addressed to Jeff Gore.

Bacteria-Firmicutes-Bacilli-Lactobacillales-Streptococcaceae-Lactococcus
Bacteria-Firmicutes-Bacilli-Lactobacillales-Leuconostocaceae-Leuconostoc
Bacteria-Firmicutes-Bacilli-Lactobacillales-Leuconostocaceae-Leuconostoc
Bacteria-Firmicutes-Bacilli-Exiguobacterales-Exiguobacteraceae-Exiguobacterium
Bacteria-Firmicutes-Bacilli-Bacillales-Planococcaceae-Lysinibacillus
Bacteria-Firmicutes-Bacilli-Staphylococcales-Staphylococcaceae-Staphylococcus
Bacteria-Firmicutes-Bacilli-Bacillales-Bacillaceae-Bacillus
Bacteria-Firmicutes-Bacilli-Lactobacillales-Streptococcaceae-Lactococcus
Bacteria-Firmicutes-Bacilli-Bacillales-Planococcaceae-NA
Bacteria-Firmicutes-Bacilli-Staphylococcales-Staphylococcaceae-Staphylococcus
Bacteria-Firmicutes-Bacilli-Bacillales-Bacillaceae-Bacillus
Bacteria-Firmicutes-Bacilli-Exiguobacterales-Exiguobacteraceae-Exiguobacterium
Bacteria-Firmicutes-Bacilli-Lactobacillales-Leuconostocaceae-Leuconostoc
Bacteria-Firmicutes-Bacilli-Exiguobacterales-Exiguobacteraceae-Exiguobacterium
Bacteria-Firmicutes-Bacilli-Bacillales-Bacillaceae-Bacillus
Bacteria-Firmicutes-Bacilli-Lactobacillales-Streptococcaceae-Lactococcus
Bacteria-Firmicutes-Clostridia-Lachnospirales-Lachnospiraceae-Lachnospiraceae$_N$K4A136$_g$roup
Bacteria-Firmicutes-Bacilli-Bacillales-Planococcaceae-Lysinibacillus
Bacteria-Firmicutes-Clostridia-Lachnospirales-Lachnospiraceae-Agathobacter
Bacteria-Proteobacteria-Gammaproteobacteria-Enterobacterales-Enterobacteriaceae-Raoultella
Bacteria-Proteobacteria-Gammaproteobacteria-Enterobacterales-Enterobacteriaceae-Klebsiella
Bacteria-Proteobacteria-Gammaproteobacteria-Enterobacterales-Enterobacteriaceae-Pluralibacter
Bacteria-Proteobacteria-Gammaproteobacteria-Aeromonadales-Aeromonadaceae-Aeromonas
Bacteria-Proteobacteria-Gammaproteobacteria-Enterobacterales-NA-NA
Bacteria-Proteobacteria-Gammaproteobacteria-Xanthomonadales-Xanthomonadaceae-Stenotrophomonas
Bacteria-Proteobacteria-Gammaproteobacteria-Enterobacterales-Erwiniaceae-Pantoea
Bacteria-Proteobacteria-Alphaproteobacteria-Rhizobiales-Rhizobiaceae-Ochrobactrum
Bacteria-Proteobacteria-Gammaproteobacteria-Pseudomonadales-Pseudomonadaceae-Pseudomonas
Bacteria-Proteobacteria-Gammaproteobacteria-Enterobacterales-Enterobacteriaceae-Escherichia/Shigella
Bacteria-Proteobacteria-Gammaproteobacteria-Pseudomonadales-Moraxellaceae-Acinetobacter
Bacteria-Proteobacteria-Gammaproteobacteria-Pseudomonadales-Pseudomonadaceae-Pseudomonas
Bacteria-Proteobacteria-Gammaproteobacteria-Burkholderiales-Oxalobacteraceae-Herbaspirillum
Bacteria-Proteobacteria-Gammaproteobacteria-Pseudomonadales-Pseudomonadaceae-Pseudomonas
Bacteria-Proteobacteria-Gammaproteobacteria-Xanthomonadales-Xanthomonadaceae-Stenotrophomonas
Bacteria-Proteobacteria-Gammaproteobacteria-Burkholderiales-Oxalobacteraceae-Undibacterium
Bacteria-Proteobacteria-Gammaproteobacteria-Enterobacterales-Enterobacteriaceae-NA
Bacteria-Proteobacteria-Gammaproteobacteria-Enterobacterales-Enterobacteriaceae-Raoultella
Bacteria-Proteobacteria-Gammaproteobacteria-Aeromonadales-Aeromonadaceae-Aeromonas
Bacteria-Proteobacteria-Gammaproteobacteria-Burkholderiales-Comamonadaceae-Acidovorax
Bacteria-Proteobacteria-Gammaproteobacteria-Enterobacterales-Enterobacteriaceae-Citrobacter
Bacteria-Proteobacteria-Gammaproteobacteria-Enterobacterales-Erwiniaceae-Pantoea
Bacteria-Proteobacteria-Gammaproteobacteria-Pseudomonadales-Pseudomonadaceae-Pseudomonas
Bacteria-Proteobacteria-Gammaproteobacteria-Enterobacterales-Enterobacteriaceae-Klebsiella
Bacteria-Proteobacteria-Gammaproteobacteria-Enterobacterales-Enterobacteriaceae-Enterobacter
Bacteria-Proteobacteria-Gammaproteobacteria-Pseudomonadales-Pseudomonadaceae-Pseudomonas
Bacteria-Proteobacteria-Gammaproteobacteria-Pseudomonadales-Pseudomonadaceae-Pseudomonas
Bacteria-Proteobacteria-Gammaproteobacteria-Aeromonadales-Aeromonadaceae-Aeromonas
Bacteria-Proteobacteria-Gammaproteobacteria-Aeromonadales-Aeromonadaceae-Tolumonas
Bacteria-Proteobacteria-Gammaproteobacteria-Enterobacterales-NA-NA
Bacteria-Proteobacteria-Gammaproteobacteria-Enterobacterales-Enterobacteriaceae-Pluralibacter
Bacteria-Proteobacteria-Gammaproteobacteria-Pseudomonadales-Pseudomonadaceae-Pseudomonas
Bacteria-Proteobacteria-Gammaproteobacteria-Xanthomonadales-Xanthomonadaceae-Stenotrophomonas
Bacteria-Proteobacteria-Gammaproteobacteria-Aeromonadales-Aeromonadaceae-Tolumonas
Bacteria-Proteobacteria-Gammaproteobacteria-Enterobacterales-Erwiniaceae-Pantoea
Bacteria-Proteobacteria-Gammaproteobacteria-Burkholderiales-Oxalobacteraceae-Undibacterium
Bacteria-Proteobacteria-Gammaproteobacteria-Pseudomonadales-Moraxellaceae-Acinetobacter
Bacteria-Proteobacteria-Gammaproteobacteria-Enterobacterales-Enterobacteriaceae-Citrobacter
Bacteria-Proteobacteria-Gammaproteobacteria-Enterobacterales-Enterobacteriaceae-Escherichia/Shigella
Bacteria-Proteobacteria-Alphaproteobacteria-Rhizobiales-Rhizobiaceae-Ochrobactrum
Bacteria-Proteobacteria-Gammaproteobacteria-Enterobacterales-Enterobacteriaceae-NA
Bacteria-Proteobacteria-Gammaproteobacteria-Burkholderiales-Comamonadaceae-Acidovorax
Bacteria-Bacteroidota-Bacteroidia-Flavobacteriales-Weeksellaceae-Chryseobacterium
Bacteria-Bacteroidota-Bacteroidia-Flavobacteriales-Weeksellaceae-Empedobacter
Bacteria-Bacteroidota-Bacteroidia-Flavobacteriales-Weeksellaceae-Empedobacter
Bacteria-Bacteroidota-Bacteroidia-Sphingobacteriales-Sphingobacteriaceae-Sphingobacterium
Bacteria-Bacteroidota-Bacteroidia-Flavobacteriales-Weeksellaceae-Empedobacter
Bacteria-Bacteroidota-Bacteroidia-Flavobacteriales-Weeksellaceae-Empedobacter
Bacteria-Bacteroidota-Bacteroidia-Sphingobacteriales-Sphingobacteriaceae-Pedobacter
Bacteria-Bacteroidota-Bacteroidia-Cytophagales-Spirosomaceae-Flectobacillus
Bacteria-Bacteroidota-Bacteroidia-Cytophagales-Spirosomaceae-Flectobacillus
Bacteria-Bacteroidota-Bacteroidia-Flavobacteriales-Weeksellaceae-Chryseobacterium
Bacteria-Bacteroidota-Bacteroidia-Flavobacteriales-Flavobacteriaceae-Flavobacterium
Bacteria-Bacteroidota-Bacteroidia-Bacteroidales-Williamwhitmaniaceae-Acetobacteroides
Bacteria-Bacteroidota-Bacteroidia-Flavobacteriales-Weeksellaceae-Empedobacter
Bacteria-Bacteroidota-Bacteroidia-Flavobacteriales-Weeksellaceae-Empedobacter
Bacteria-Bacteroidota-Bacteroidia-Flavobacteriales-Flavobacteriaceae-Flavobacterium
Bacteria-Bacteroidota-Bacteroidia-Flavobacteriales-Weeksellaceae-Chryseobacterium
Bacteria-Actinobacteriota-Actinobacteria-Streptomycetales-Streptomycetaceae-Streptomyces
Bacteria-Actinobacteriota-Actinobacteria-Micrococcales-Microbacteriaceae-Curtobacterium
Bacteria-Cyanobacteria-Cyanobacteriia-Chloroplast-NA-NA

**Extended Data Fig. 1 | See next page for caption.**

**Extended Data Fig. 1 | Taxonomic identity of the bacterial isolates.** The identities have been inferred from the ASV (Methods) of 16S sequencing, which allow the classification of the 80 isolates down to the genus level. Colors are consistent with those in the main text and other supplementary figures. Species belonging to the Firmicutes phylum are assigned different shades of blue, Proteobacteria species are assigned different shades of green, Bacteroidota species are assigned different shades of red, Actinobacteriota species are assigned different shades of purple, Cyanobacteria species are assigned different shades of yellow.

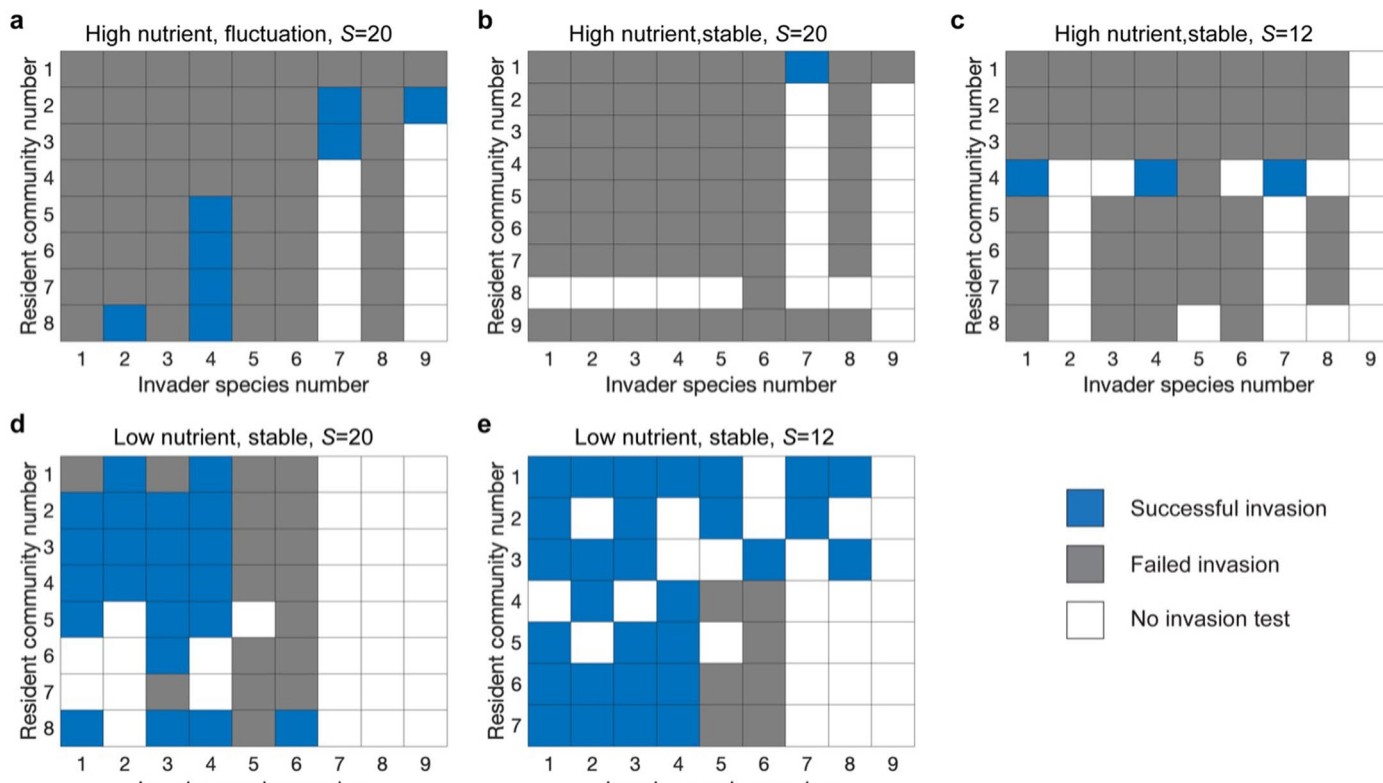

**Extended Data Fig. 2 | Introducing different invaders into different resident communities and measuring the invasion outcome through 16 s sequencing.** The invasion outcome matrices show that increasing nutrient and species pool size lead to a decrease in invasion probability. Specifically: (a) and (b) show the invasion outcomes of fluctuating and stable communities, respectively, under high nutrient conditions with an initial species pool size of $S = 20$; (c) shows the invasion outcomes of stable communities under high nutrient conditions with an initial species pool size of $S = 12$; (d) and (e) depict the invasion outcomes of stable communities under low nutrient conditions with initial species pool sizes of $S = 20$ and $S = 12$, respectively.

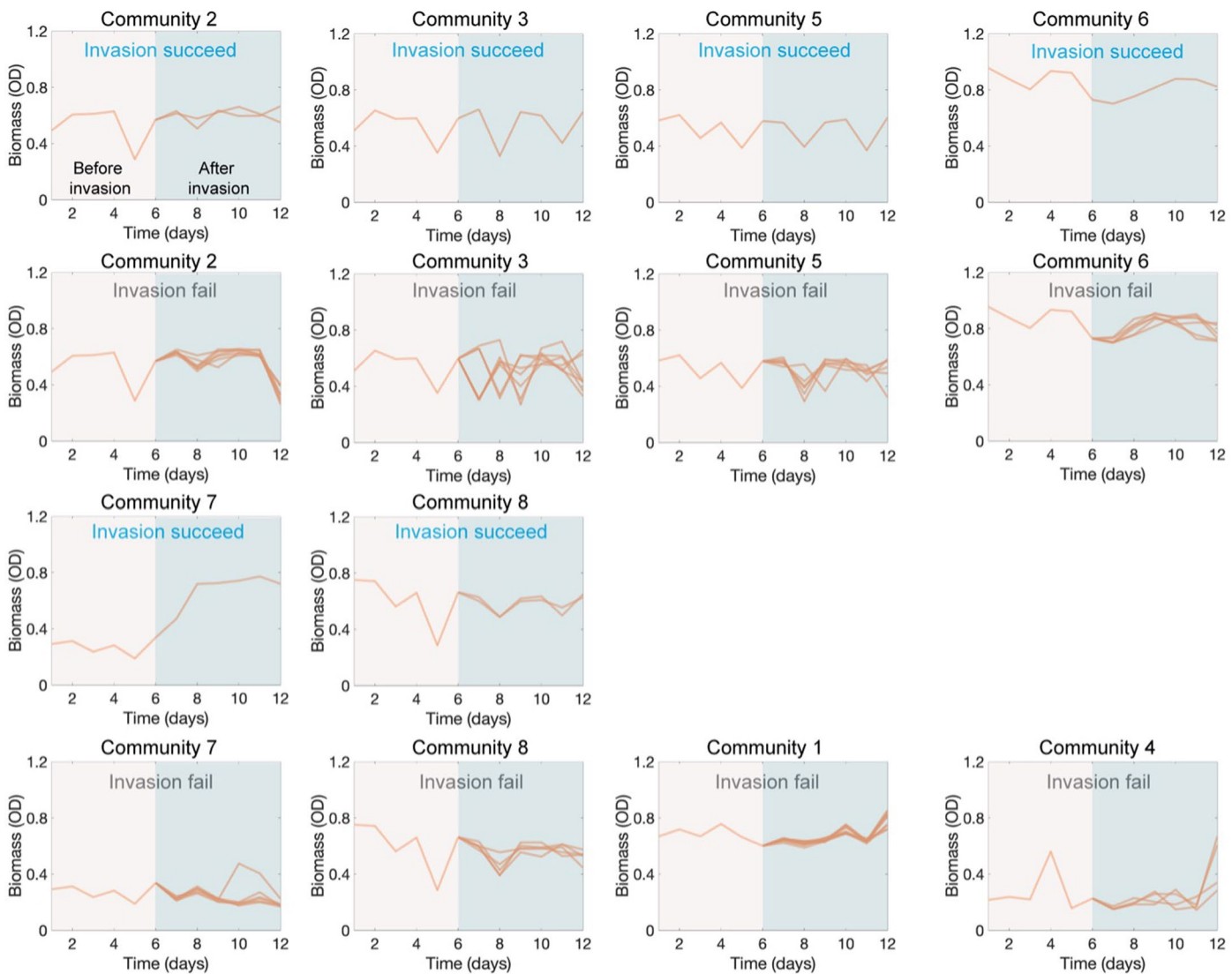

**Extended Data Fig. 3 | Time series for the biomass of the fluctuating communities with species pool size $S = 20$ under strong average interaction strength (high nutrients concentration).** Each panel shows the time series for the OD (600 nm) of one fluctuating community with species pool size $S = 20$ under high nutrient. The invaders were introduced on day 6, and the time series of successful invasions and failed invasions for the same communities were displayed in different panels.

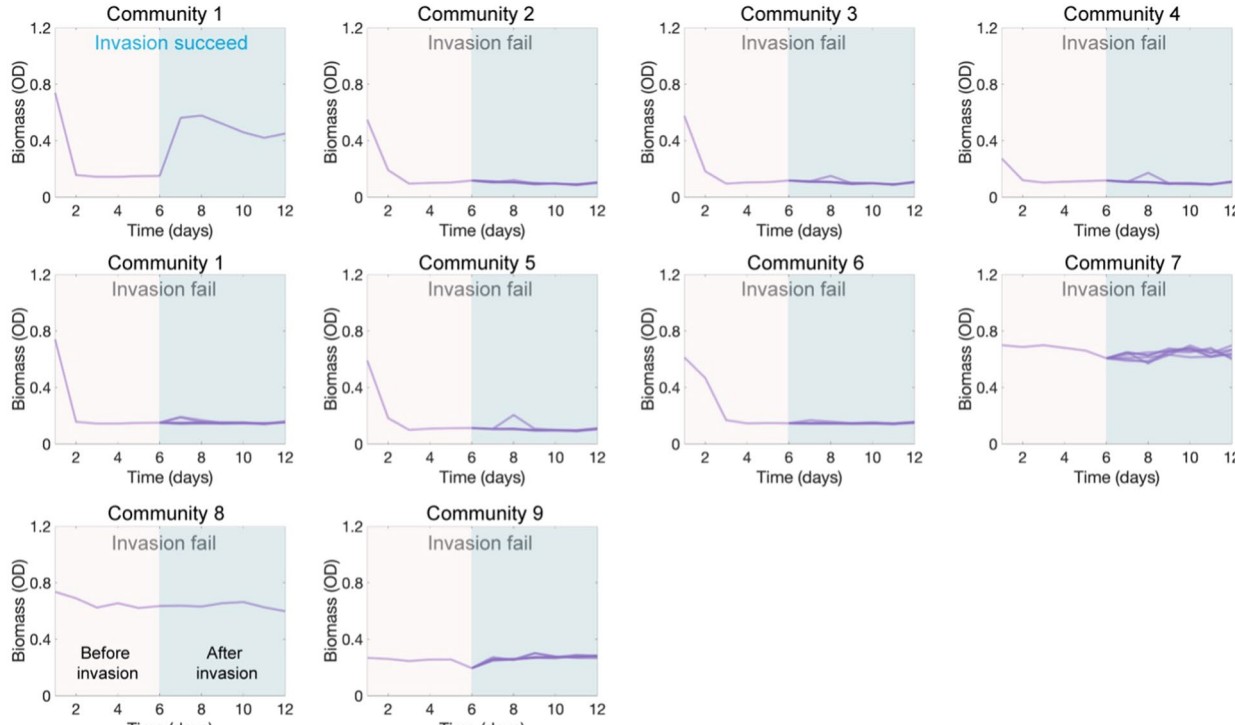

**Extended Data Fig. 4 | Time series for the biomass of the stable communities with species pool size $S$ = 20 under strong average interaction strength (high nutrients concentration).** Each panel shows the time series for the OD (600 nm) of one stable community with species pool size $S$ = 20 under high nutrient. The invaders were introduced on day 6, and the time series of successful invasions and failed invasions for the same communities were displayed in different panels.

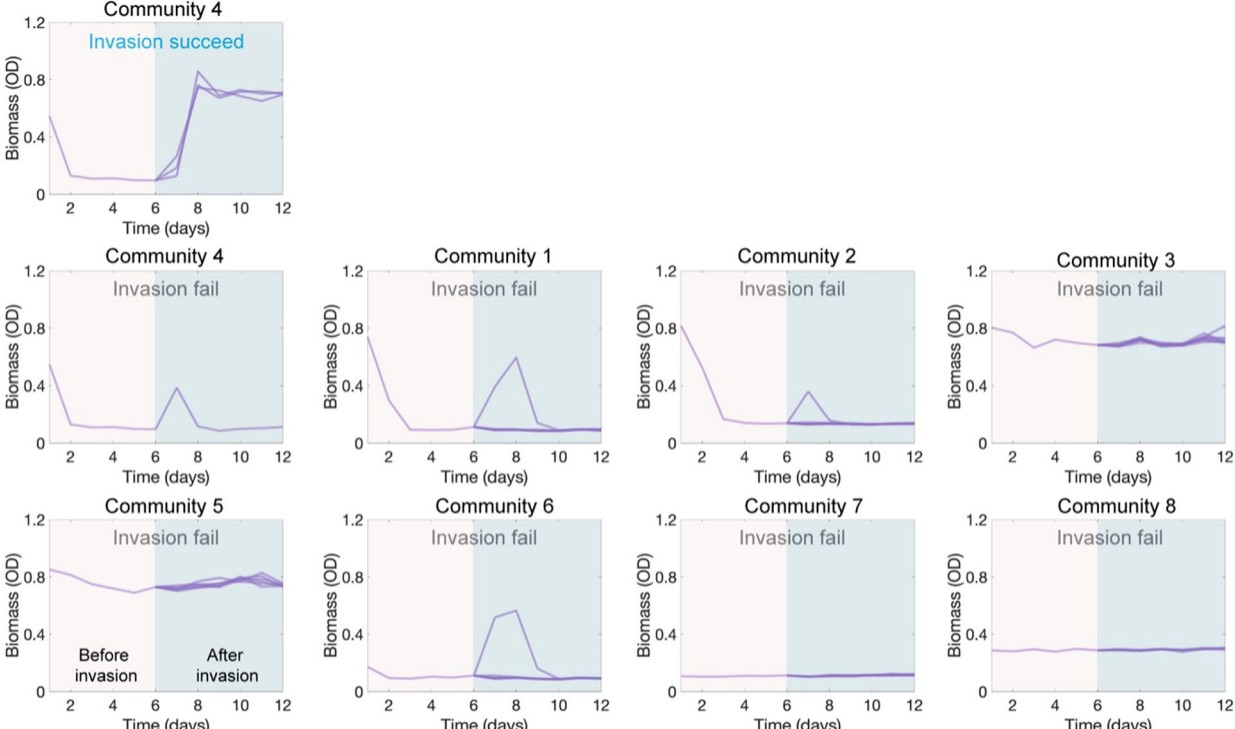

**Extended Data Fig. 5 | Time series for the biomass of the stable communities with species pool size $S$ = 12 under strong average interaction strength (high nutrients concentration).** Each panel shows the time series for the OD (600 nm) of one stable community with species pool size $S$ = 12 under high nutrient. The invaders were introduced on day 6, and the time series of successful invasions and failed invasions for the same communities were displayed in different panels.

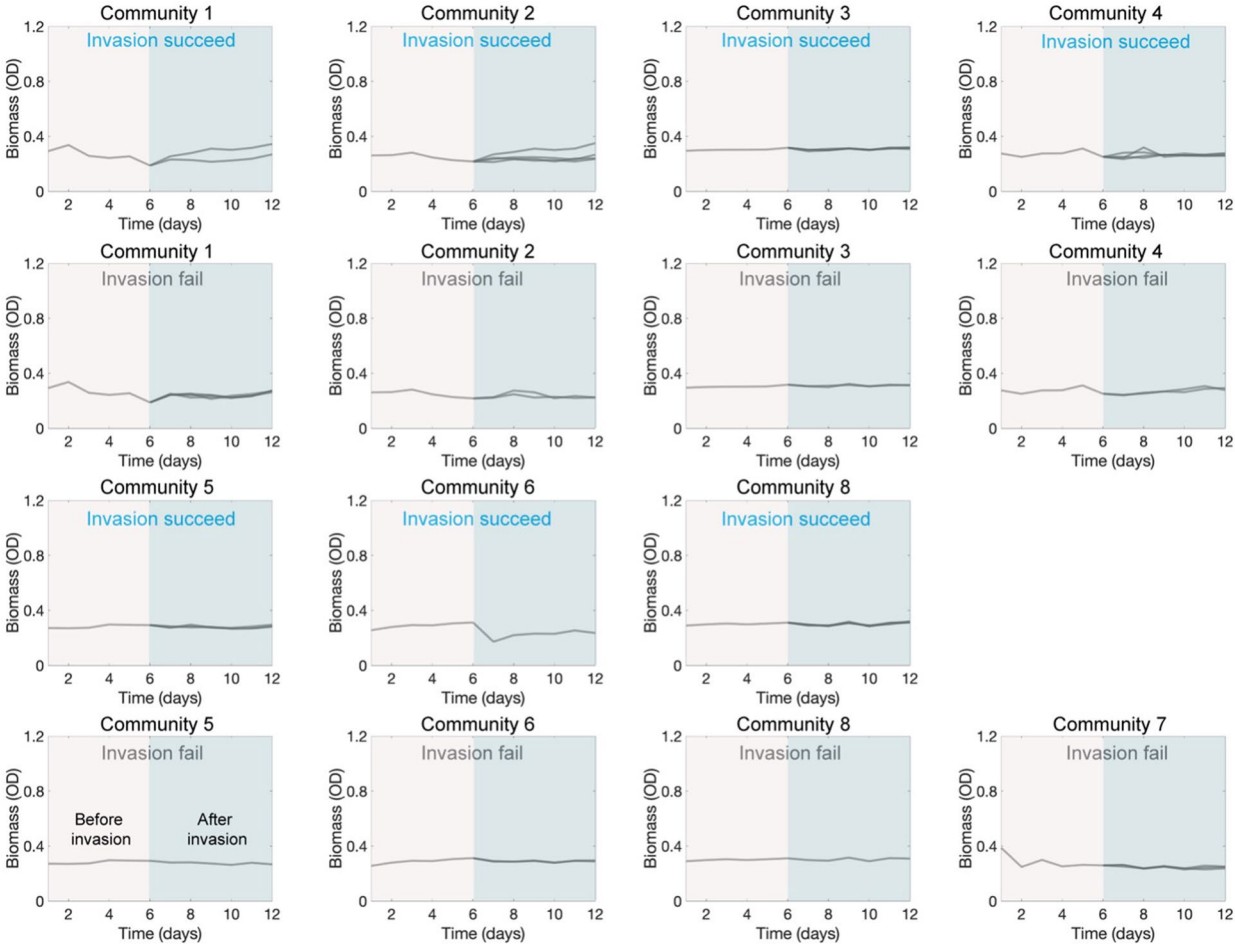

**Extended Data Fig. 6 | Supplementary Fig. 6. Time series for the biomass of the stable communities with species pool size $S$ = 20 under weak average interaction strength (low nutrients concentration).** Each panel shows the time series for the OD (600 nm) of one stable community with species pool size $S$ = 20 under low nutrient. The invaders were introduced on day 6, and the time series of successful invasions and failed invasions for the same communities were displayed in different panels.

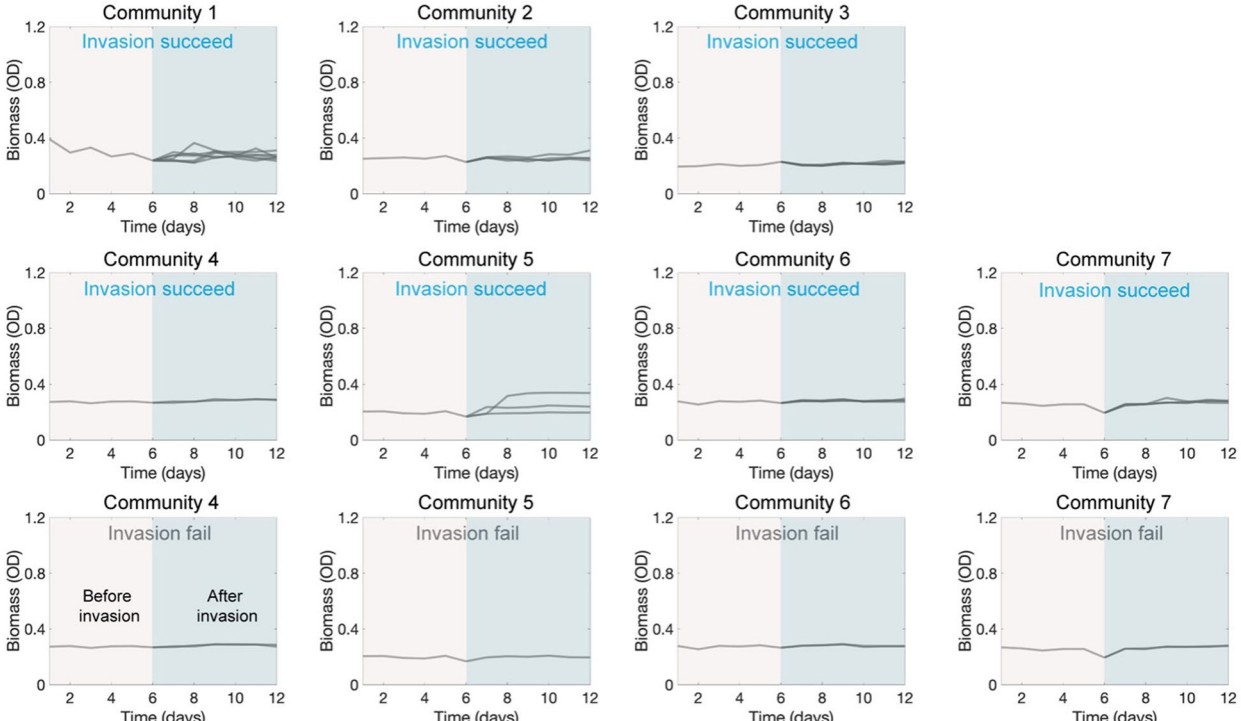

**Extended Data Fig. 7 | Time series for the biomass of the stable communities with species pool size $S$ = 12 under weak average interaction strength (low nutrients concentration).** Each panel shows the time series for the OD (600 nm) of one stable community with species pool size $S$ = 12 under low nutrient. The invaders were introduced on day 6, and the time series of successful invasions and failed invasions for the same communities were displayed in different panels.

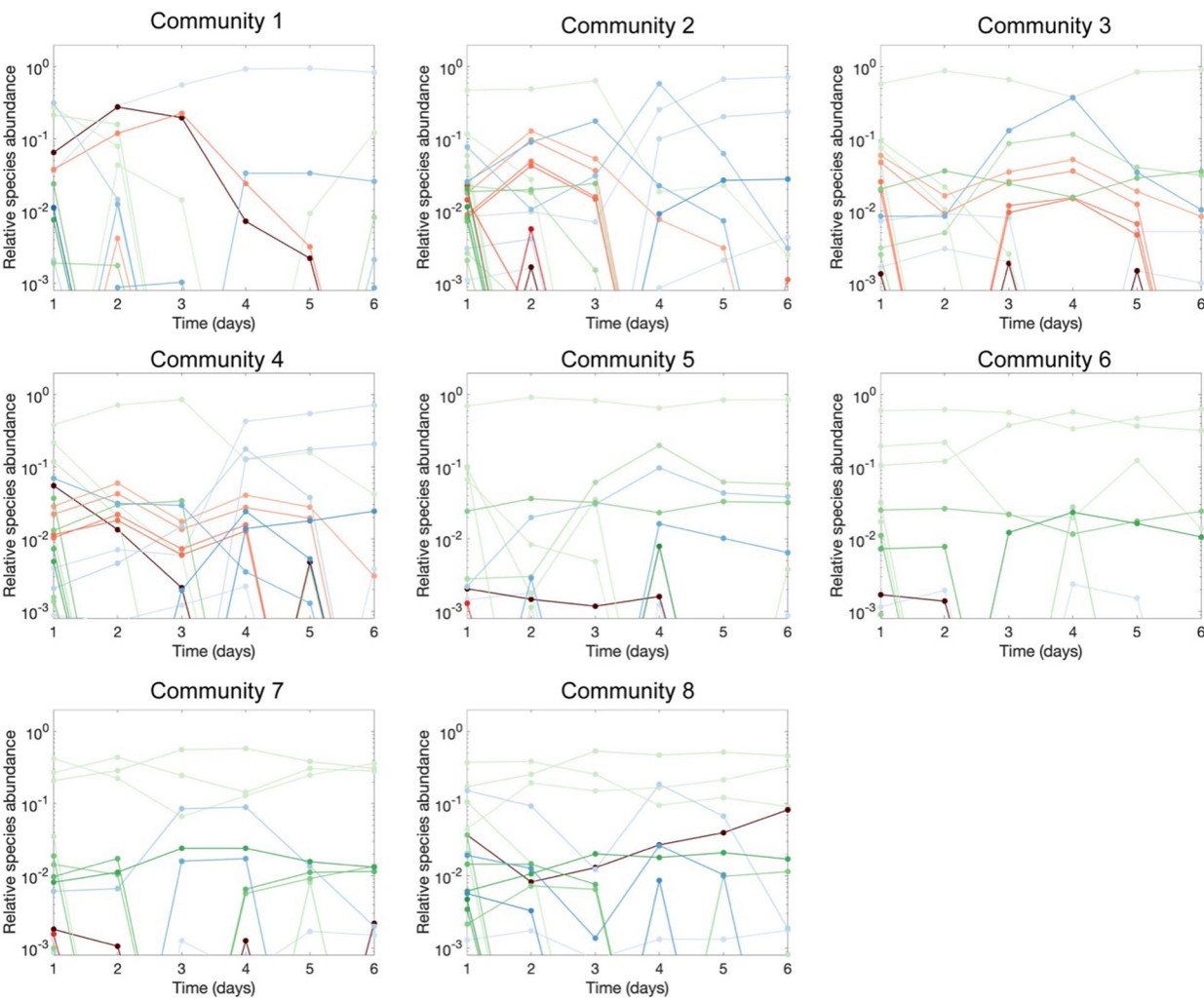

**Extended Data Fig. 8 | Time series for the relative species abundances of the fluctuating communities with species pool size *S* = 20 under strong average interaction strength (high nutrients concentration).** Each panel shows the time series for the relative species abundances of one fluctuating community before introducing invaders, where species pool size *S* = 20 under high nutrient.

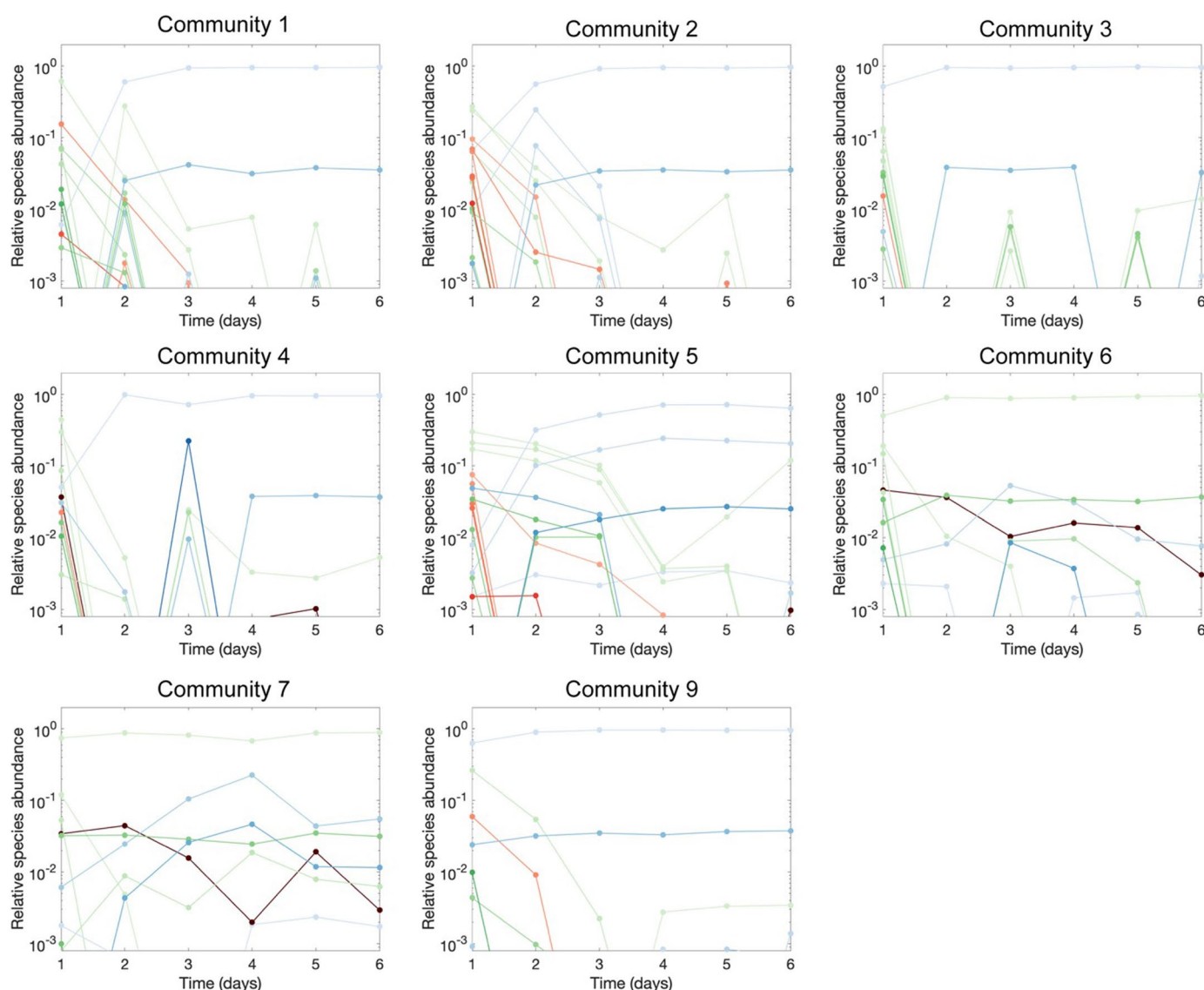

**Extended Data Fig. 9 | Time series for the relative species abundances of the stable communities with species pool size $S$ = 20 under strong average interaction strength (high nutrients concentration).** Each panel shows the time series for the relative species abundances of one stable community before introducing invaders, where species pool size $S$ = 20 under high nutrient.

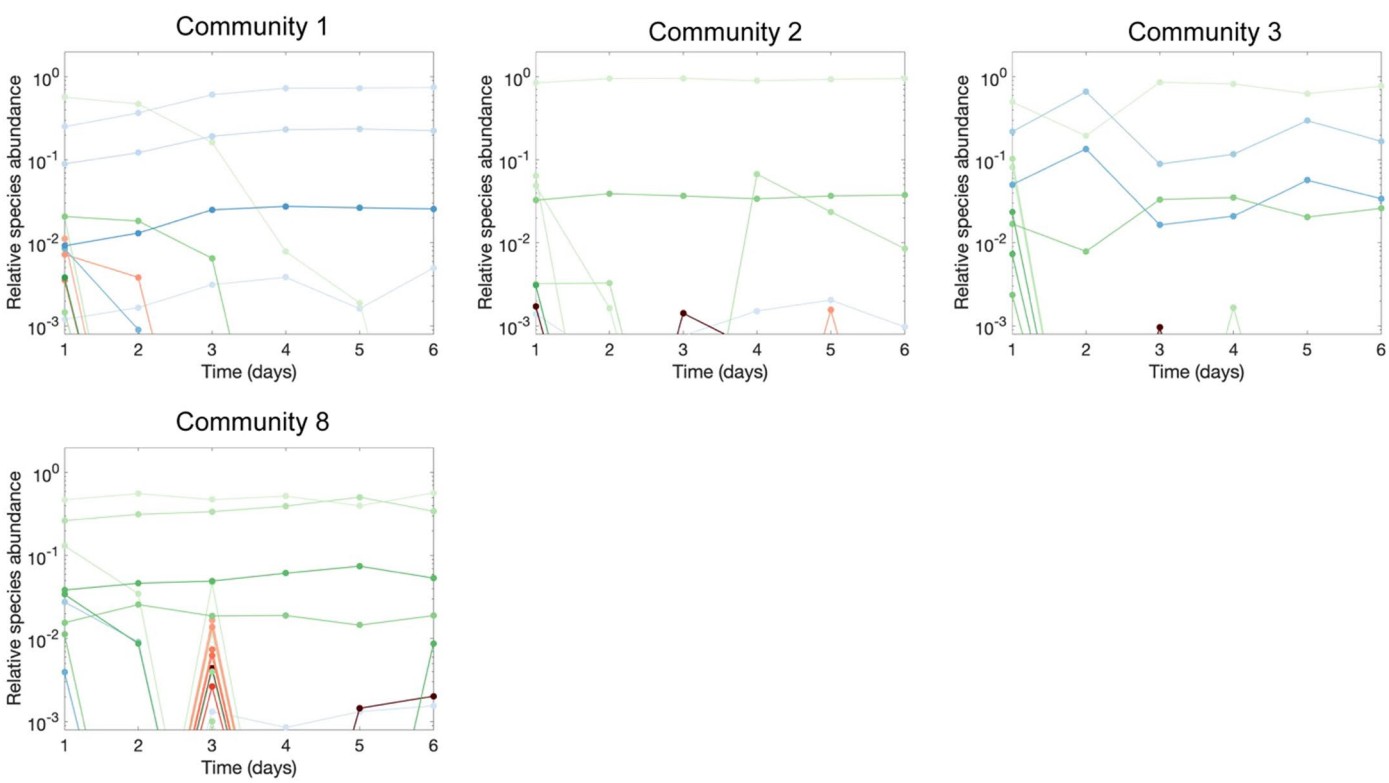

**Extended Data Fig. 10 | Time series for the relative species abundances of the stable communities with species pool size** $S$ = **12 under strong average interaction strength (high nutrients concentration).** Each panel shows the time series for the relative species abundances of one stable community before introducing invaders, where species pool size $S$ = 12 under high nutrient.

# Reporting Summary

## Statistics

For all statistical analyses, confirm that the following items are present in the figure legend, table legend, main text, or Methods section.

| n/a | Confirmed | |
|---|---|---|
| ☐ | ☒ | The exact sample size ($n$) for each experimental group/condition, given as a discrete number and unit of measurement |
| ☐ | ☒ | A statement on whether measurements were taken from distinct samples or whether the same sample was measured repeatedly |
| ☐ | ☒ | The statistical test(s) used AND whether they are one- or two-sided *Only common tests should be described solely by name; describe more complex techniques in the Methods section.* |
| ☐ | ☒ | A description of all covariates tested |
| ☐ | ☒ | A description of any assumptions or corrections, such as tests of normality and adjustment for multiple comparisons |
| ☐ | ☒ | A full description of the statistical parameters including central tendency (e.g. means) or other basic estimates (e.g. regression coefficient) AND variation (e.g. standard deviation) or associated estimates of uncertainty (e.g. confidence intervals) |
| ☐ | ☒ | For null hypothesis testing, the test statistic (e.g. $F$, $t$, $r$) with confidence intervals, effect sizes, degrees of freedom and $P$ value noted *Give P values as exact values whenever suitable.* |
| ☒ | ☐ | For Bayesian analysis, information on the choice of priors and Markov chain Monte Carlo settings |
| ☒ | ☐ | For hierarchical and complex designs, identification of the appropriate level for tests and full reporting of outcomes |
| ☐ | ☒ | Estimates of effect sizes (e.g. Cohen's $d$, Pearson's $r$), indicating how they were calculated |

*Our web collection on statistics for biologists contains articles on many of the points above.*

## Software and code

Policy information about availability of computer code

| | |
|---|---|
| Data collection | The simulations in this study were conducted using MATLAB 2024. |
| Data analysis | We used the DADA2 R package to obtain amplicon sequence variants (ASVs). Taxonomic identities were assigned to the ASVs using the SILVA reference database (version 132). |

For manuscripts utilizing custom algorithms or software that are central to the research but not yet described in published literature, software must be made available to editors and reviewers. We strongly encourage code deposition in a community repository (e.g. GitHub). See the Nature Portfolio guidelines for submitting code & software for further information.

## Data

Policy information about availability of data

All manuscripts must include a data availability statement. This statement should provide the following information, where applicable:
- Accession codes, unique identifiers, or web links for publicly available datasets
- A description of any restrictions on data availability
- For clinical datasets or third party data, please ensure that the statement adheres to our policy

All data are available in the supplementary materials, and the raw sequencing data have been deposited in Dryad: DOI: 10.5061/dryad.8gtht76xz

# Research involving human participants, their data, or biological material

Policy information about studies with [human participants or human data](). See also policy information about [sex, gender (identity/presentation), and sexual orientation]() and [race, ethnicity and racism]().

| | |
|---|---|
| Reporting on sex and gender | *Use the terms sex (biological attribute) and gender (shaped by social and cultural circumstances) carefully in order to avoid confusing both terms. Indicate if findings apply to only one sex or gender; describe whether sex and gender were considered in study design; whether sex and/or gender was determined based on self-reporting or assigned and methods used.* <br> *Provide in the source data disaggregated sex and gender data, where this information has been collected, and if consent has been obtained for sharing of individual-level data; provide overall numbers in this Reporting Summary. Please state if this information has not been collected.* <br> *Report sex- and gender-based analyses where performed, justify reasons for lack of sex- and gender-based analysis.* |
| Reporting on race, ethnicity, or other socially relevant groupings | *Please specify the socially constructed or socially relevant categorization variable(s) used in your manuscript and explain why they were used. Please note that such variables should not be used as proxies for other socially constructed/relevant variables (for example, race or ethnicity should not be used as a proxy for socioeconomic status).* <br> *Provide clear definitions of the relevant terms used, how they were provided (by the participants/respondents, the researchers, or third parties), and the method(s) used to classify people into the different categories (e.g. self-report, census or administrative data, social media data, etc.)* <br> *Please provide details about how you controlled for confounding variables in your analyses.* |
| Population characteristics | *Describe the covariate-relevant population characteristics of the human research participants (e.g. age, genotypic information, past and current diagnosis and treatment categories). If you filled out the behavioural & social sciences study design questions and have nothing to add here, write "See above."* |
| Recruitment | *Describe how participants were recruited. Outline any potential self-selection bias or other biases that may be present and how these are likely to impact results.* |
| Ethics oversight | *Identify the organization(s) that approved the study protocol.* |

Note that full information on the approval of the study protocol must also be provided in the manuscript.

# Field-specific reporting

Please select the one below that is the best fit for your research. If you are not sure, read the appropriate sections before making your selection.

☒ Life sciences    ☐ Behavioural & social sciences    ☐ Ecological, evolutionary & environmental sciences

For a reference copy of the document with all sections, see [nature.com/documents/nr-reporting-summary-flat.pdf](http://nature.com/documents/nr-reporting-summary-flat.pdf)

# Life sciences study design

All studies must disclose on these points even when the disclosure is negative.

| | |
|---|---|
| Sample size | The sample size for the study was based on previous literature and experimental feasibility. For each experimental condition, we built 17 different synthetic microbial communities using a library of 80 bacterial isolates. Each community was exposed to multiple invasion tests (7-9 per community), which allowed us to detect statistically significant differences in invasion success rates. |
| Data exclusions | No data were excluded from the analyses. |
| Replication | All attempts at replication were successful. |
| Randomization | Samples and bacterial isolates were randomly allocated into experimental groups. Invader species were randomly chosen for each invasion test to minimize any potential biases in species selection or community assembly. |
| Blinding | Blinding was not relevant to this study because the bacterial species in the resident community and the invader were randomly chosen for each experiment. |

# Reporting for specific materials, systems and methods

We require information from authors about some types of materials, experimental systems and methods used in many studies. Here, indicate whether each material, system or method listed is relevant to your study. If you are not sure if a list item applies to your research, read the appropriate section before selecting a response.

## Materials & experimental systems

| n/a | Involved in the study |
|-----|----------------------|
| ☒ ☐ | Antibodies |
| ☒ ☐ | Eukaryotic cell lines |
| ☒ ☐ | Palaeontology and archaeology |
| ☒ ☐ | Animals and other organisms |
| ☒ ☐ | Clinical data |
| ☒ ☐ | Dual use research of concern |
| ☒ ☐ | Plants |

## Methods

| n/a | Involved in the study |
|-----|----------------------|
| ☒ ☐ | ChIP-seq |
| ☒ ☐ | Flow cytometry |
| ☒ ☐ | MRI-based neuroimaging |

## Plants

| | |
|---|---|
| Seed stocks | *Report on the source of all seed stocks or other plant material used. If applicable, state the seed stock centre and catalogue number. If plant specimens were collected from the field, describe the collection location, date and sampling procedures.* |
| Novel plant genotypes | *Describe the methods by which all novel plant genotypes were produced. This includes those generated by transgenic approaches, gene editing, chemical/radiation-based mutagenesis and hybridization. For transgenic lines, describe the transformation method, the number of independent lines analyzed and the generation upon which experiments were performed. For gene-edited lines, describe the editor used, the endogenous sequence targeted for editing, the targeting guide RNA sequence (if applicable) and how the editor was applied.* |
| Authentication | *Describe any authentication procedures for each seed stock used or novel genotype generated. Describe any experiments used to assess the effect of a mutation and, where applicable, how potential secondary effects (e.g. second site T-DNA insertions, mosiacism, off-target gene editing) were examined.* |

