## [Peer Review File · Nature Ecology & Evolution]

Collective dynamical regimes predict invasion success and impacts in microbial communities

Corresponding Author: Professor Jeff Gore

Version 0:

Decision Letter:

17th April 2024

Dear Professor Gore,

Thank you for your patience while your Article "Collective dynamical regimes predict invasion success and impacts in microbial communities" was under review. We have now received three reports. As you will see from the comments copied below, the reviewers find your work of considerable potential interest but have raised quite substantial concerns. In light of these comments, we cannot accept the manuscript for publication but would be very interested in considering a revised version that addresses these concerns.

We hope you will find the reviewers' comments useful as you decide how to proceed. Please bear in mind that we will be reluctant to approach the reviewers again in the absence of major revisions. In particular, while we do not expect additional experimental work, addressing the concerns raised to the reviewers' satisfactions will likely require additional modelling work, as well as further information on the methodology. Regarding the latter, if you are concerned about space limitations in the main manuscript file, feel free to use the supplementary files to provide all the information required by the reviewers (with clear guidance as to where find a specific piece of information).

If you choose to revise your manuscript taking into account all reviewer and editor comments, please highlight all changes in the manuscript text file, preferably in Microsoft Word format.

* Include a "Response to reviewers" document detailing, point-by-point, how you addressed each referee comment. If no action was taken to address a point, you must provide a compelling argument. This response will be sent back to the referees along with the revised manuscript.

* If you have not done so already we suggest that you begin to revise your manuscript so that it conforms to our Article format instructions at <http://www.nature.com/natecolevol/info/final-submission>. Refer also to any guidelines provided in this letter.

Link Redacted

If you wish to submit a suitably revised manuscript we would hope to receive it within 6 months. If you cannot send it within this time, please let us know. We will be happy to consider your revision so long as nothing similar has been accepted for publication at Nature Ecology & Evolution or published elsewhere.

Nature Ecology & Evolution is committed to improving transparency in authorship. As part of our efforts in this direction, we are

now requesting that all authors identified as 'corresponding author' on published papers create and link their Open Researcher and Contributor Identifier (ORCID) with their account on the Manuscript Tracking System (MTS), prior to acceptance. This applies to primary research papers only. ORCID helps the scientific community achieve unambiguous attribution of all scholarly contributions. You can create and link your ORCID from the home page of the MTS by clicking on 'Modify my Springer Nature account'. For more information please visit www.springernature.com/orcid.

Thank you for the opportunity to review your work.

[redacted]

Reviewers' comments:

Reviewer #1 (Remarks to the Author):

In this manuscript, Hu et al. combine experiments and theory to elucidate the factors determining community invasions. They find that while diversity-invasibility relationships can be both positive and negative depending on the factors regulating diversity, the fraction of surviving species directly predicts invasibility. This manuscript leverages their recent insights on the dynamical phases of microbial ecosystems to provide a timely and relevant exploration of invasions. The results are interesting and the research is carried out soundly. My comments are primarily aimed at making the manuscript easier to follow.

My first main comment concerns the exposition of the different diversity-invasibility relationships between Fig. 1 and Fig. 2/3, which on a first read was not immediately obvious. In Fig 1, more species and higher interaction strength makes for fluctuations and greater diversity, and fluctuating communities are more likely to be invaded; but both theory and Fig. 3 show a negative correlation between invasion probability and pool size/interaction strength. I naively expected increased pool size and stronger interactions to translate to a higher change of fluctuations (as in their recent paper), and thus greater richness and higher invasion probability. It is later revealed where this apparent contradiction comes from, but I believe it would be immensely helpful if this contradiction was highlighted early on. This issue is made more severe by the use of three different terms to describe the probability of successful invasions (invasion probability and invasion resistance, also invasibility) – especially because these have opposite meanings, I had to read this section of the manuscript carefully to understand its meaning. A single term would be easier to understand.

My second main comment concerns the notion of priority effects. Here, as I understand it, the authors use this term basically to mean that a community of strongly interacting species has established itself, which makes invasions less likely (e.g., because all niches are filled or because antagonistic interactions are balanced out) but I am not convinced that this is the sense in which most ecologists tend to use that term; to me, the more common use appears to be referring to particular species whose presence or absence steers the community into different alternative stable states. Here, however, the community is already in a stable state and a new species is trying to invade, and thus "priority effect" does not apply in my view. I suggest a different way of describing this effect here, or, at the very least, the sentence in line 244-246 needs more explanation. Similarly, I find the use of "alternative stable states" (I251) to be unclear since that is not really the subject of this paper.

Minor comments:

L110: "The lower probability" – compared to what?

L174: "including species interaction strength" – is it clear that different synthetic communities can be characterized by the same average interaction strength? Perhaps some initial compositions have strong antibiotic producers or very similar metabolic profiles

L227: correlation on -> with

L239: Why should we expect a 1:1 correspondence between survival fraction and invasion probability? This only becomes clear in Fig 4b

L280: ergodicity likely needs a bit more explanation to be clear

L285: definition of invasion effect is slightly unclear. Is this the same definition as in L296?

L317: SI Fig 17 does not show an Allee effect for invaders 2/3

Fig 2f/4c: Legend missing for the gray lines

Fig 4a: it is very hard to parse the different colors

Reviewer #2 (Remarks to the Author):

Summary

Overall, I found that the study addressed interesting questions and the choice to connect experiments with minimal models of community dynamics seemed like the appropriate choice. However, several decisions made and details about the experiment seem unclear. For example, it is unclear to me how 1) a generalized Lotka-Volterra (gLTV) serves as an appropriate model for pH-mediated effects, how fluctuating/stable communities were classified, whether their experimentally-imposed migration rate was sufficient to induce deterministic fluctuations.

The visualizations are effective and thorough. However, the authors often make claims about the interpretation of visualizations that need to be backed up by statistical tests. I have attempted to identify reasonable statistical tests when a test is lacking. There are also several instances where further clarification could help a broad readership understand the study. These instances are pointed out under both Major and Minor Comments.

Major Comments

There is no explicit consideration of resources in their model. This makes sense in some respects, as the gLV can be derived in the adiabatic limit of certain consumer-resource models. Furthermore, their experimental manipulations of invasion are demographic rather than environmental, so it makes sense that a phenomenological model that does not explicitly consider microscopic interactions of resource consumption is sufficient. Progress has recently been made using phenomenological models to investigate the macroecological consequences of demographic manipulations (Shoemaker et al., 2023), which may provide outside support for the authors' modeling decisions. At a minimum the authors should provide additional justification for their modelling choices and why they are appropriate for their mode.

However, the authors also manipulate resource concentration as a means to manipulate the strength of competitive interactions. This manipulation is done by increasing resource concentration, changing the pH, which is a feature of the environment that does not appear in the model and does not appear to be measured. So, while their invasion manipulations are demographic, it reads as though the difference in invasion outcomes between high and low nutrient treatments is driven by the environment. In the manuscript's current form, I imagine that it would be difficult for the reader to understand how the manipulation of resource concentration can be modeled as an increase in the strength of competitive interactions in a phenomenological gLV.

The connection between interaction coefficients and growth-mediated pH is difficult to see given that pH tends to be incorporated into terms related to growth, both for models such as Monod growth (e.g., Houtsma et al., 1996) as well as Michaelis–Menten kinetics (chapter 10 of Cornish-Bowden, 2012). I suppose that pH-dependent competition coefficients could be derived if one started with a consumer-resource model where growth terms were pH dependent and took the adiabatic limit. At minimum a form of the gLV needs to be derived so the reader can see the connection between increased resource concentration change in pH increase in the strength of competitive interactions. However, it is also unclear that a phenomenological gLV is appropriate since the pH changes within a transfer cycle due to the growth of the community. Since the change in pH within a transfer cycle is on the same timescale of the growth rate, I do not see how a phenomenological gLV can serve as an appropriate model.

My understanding is that the term “fluctuations” in the context of this study means fluctuations induced by chaotic dynamics. I understand that these are not meant to be viewed as stochastic fluctuations, but in my experience, I have found that only a fraction of eco/evo readers understand the distinction between fluctuations induced by deterministic vs. stochastic effects. It would aid the typical reader if the authors clarified what form of fluctuations they are focused on in the Introduction, specified the type of fluctuation whenever the term is used in the manuscript (e.g., “fluctuations” “deterministic fluctuations”), and briefly addressed in the Discussion how the consideration of stochastic fluctuations might factor into their subsequent research efforts.

It is worth briefly mentioning in the supplement why continued migration from the regional pool was a necessarily experimental detail for this study. Specifically, how the absence of experimentally-imposed migration would correspond to a gLV with zero migration, the analytic results of which would be decidedly less rich (e.g., no chaos, etc.). Broad readership may not pick up on this detail.

Related, did the authors perform any experiments without daily migration from the regional pool? Contrasting the results of this study with that type of experiment would strengthen their claims and provide further opportunities to test the utility of the gLV. Alternatively, if there are there any published experiments on invasion analysis using a similar experimental setting without imposing migration, then such results could be verbally compared to the results of their experiment.

Is the experimentally-imposed migration rate sufficient to induce chaotic dynamics in the context of their gLV model? I searched for this detail in the supplement but did not find it.

I would also encourage the authors to make both their code and raw data available on public repositories to ensure transparency and reproducibility.

Lines 148-149: It would help the reader if justification for the extinction threshold was provided in the main text. In lines 81-82 of the supplement the authors describe the lower bound on detection due to sequencing, but it is not clear how they arrived at this particular value. It would also help if there was a comparison between the chosen threshold and the inverse of the typical total abundance of the community (all cells), the threshold for true extinction, to provide context for the detection limitation.

Lines 208-211: I am unaware of models where the competition coefficients between species can be examined as a function of media modification. If such a model exists, could the authors briefly describe it in their supplement to help the reader understand how environmental variables (e.g., pH) map onto competition coefficients in a gLV model?

Lines 275-280: How are alternative stable states determined here?

Fig. 1d: Why are all but two of the “fluctuating” communities decreasing in biomass from day four to five. Could this simultaneous decrease in biomass be driven by experimental details rather than by the intrinsic dynamics of the communities?

Fig. 1g; lines 380-382: What statistical test was performed here?

Fig. 2f: What do the dashed and solid grey lines represent? Could the authors include a legend in the figure?

Fig. 4c: Could the authors add an axis label for the colorbar in the revision?

Fig. 4d: "Effect on resident community" is a nice term for those unfamiliar with these types of experiments, but it still reads as somewhat vague. Could the authors describe what this term means in the figure legend and provide an equation for how the quantity was estimated in the supplement?

Fig. 5: What timepoints are used to calculate the fold change? Are the same timepoints used for each community? Is the outcome of the statistical test robust to the choice of timepoints?

Supplement line 52: The sampling detection limit is $\sim 10^{-4}$ but the chosen dilution rate was 10^{-5} . How does the difference between dilution rates and detection limit shape the observed outcomes of the experiment? If it does, it may be worth describing the outcomes in different parameter limits to the reader (i.e., sampling limit much greater than dilution rate and vice-versa).

Supplement: Lines 52, 63: I am unfamiliar with the term "dispersal rate" used to describe the process of diluting cultures in a serial transfer experiment. The method used by the authors is different from the standard process of serial dilution, so I think some justification of the term as well as an explanation of how it connects to the mathematical definition of dispersal (Eq. S1) in the supplement is warranted.

Supplement lines 100-119: Does the absence of serial dilution being explicitly encoded into the simulation impact the results? Serial dilutions have been explicitly incorporated into mechanistic (Marsland et al., 2020) and phenomenological (Shoemaker et al., 2023) models of community dynamics in an experimental context. Could the authors provide justification of why serial dilution does not need to be explicitly encoded into their simulation?

Supplement lines 109-111: What are the units of time and how do they compare to the timescales of the experiments? Is the timescale of the simulation informed by the timescale of the experiment?

Supplement lines 147-148: It's unclear what the CV represents. My understanding is that each CV is calculated over time for each species for a given simulation iteration. Since the dynamics are deterministic and the interaction coefficients are a form of quenched disorder, what random variable is the CV calculated over? Does the distribution of CVs demonstrate a clear bimodality with the 10^{-3} cutoff representing the valley between the two peaks? It would help the reader if 1) additional context was provided for the CV and what it represents as well as 2) distributions of simulated CVs plotted to demonstrate the intuition behind classifying communities into stable/fluctuating classes.

Supplement lines 160-163, Fig. S12c, d: Is some degree of correlation expected since biomass was used to calculate the abundance of each species? This seems similar to the case presented in Garud et al., where the same parameter factors into both sides of the relationship (pg. 16 and 17 of supplement; 2019). Here the authors partitioned synonymous sites into two categories, providing two estimates of the quantities of interest. These two quantities are conditionally independent of the parameter that factors into both sides of the proposed relationship due to the Poisson thinning property. One quantity is then used to calculate the left side of the relationship while the other is used to calculate the right side. Is a similar analysis appropriate in this case given that total biomass factors into both sides of the quantities examined in Fig. S12c, d?

An alternative option could be to examine the relationship between the CV of biomass (OD) and the CV of relative abundances for each ASV, allowing for the identification of community members that disproportionately contribute to the fluctuations in biomass.

Fig. S2: Is there any phylogenetic structure to these results? Does the placement of a species on the phylogeny relative to the phylogenetic composition of a given community determine species invariability?

Fig. S12: "The standard deviation of community biomass over day 5, day 6 and day 7" this detail is confusing since the timeseries represented in Fig. 1d ends at day six. What was the rationale behind using only these three timepoints? Some justification of using these three timepoints is needed in the Supplement.

Fig. S13: The figure is a useful visualization, but in its current form it is difficult to validate the claims made in the legend.

- "Invisibility positively correlates with richness when varying interaction strength" Is a claim being made here about the direction in which the correlation changes with a change in interaction strength? It's not clear to me whether this claim is about the existence of positive correlations among treatments or the difference in positive correlations between treatments.

- "Invisibility positively correlates with richness when randomly sample $S=20$ communities under high nutrient, due to fluctuating communities display larger richness and larger invasion probability." I am having a hard time parsing this statement. Is a claim being made about correlation being higher in high nutrient $S=20$ fluctuating communities relative to high nutrient $S=20$ stable communities?

- The legend could use some retooling for clarity. Furthermore, if the claims are about the increase in correlation in one treatment vs. another, then additional statistical analyses are necessary. It seems like the authors are making three claims, which would require three tests. The question is what statistical model to use. A full regression analysis to examine the increase in slope between two treatments while controlling for potential confounders could be seen as necessary. Alternatively, the authors could test for the difference between two correlation coefficients using Fisher's Z statistic with a null distribution obtained by permuting

community identity for a given pair of treatments (Snedecor & Cochran, 1989). This statistic was recently used to test for the change in correlation coefficients in experimental microbial communities (Eq. 21 in Shoemaker et al., 2023).

Fig. S15, 16: Some type of statistical test is necessary to establish the claim that invasions lead to changes in community composition. There are multiple ways to accomplish this task and the authors may have their own idea. One immediate option strikes me: calculate a paired t-statistic for each community between the control and successfully invaded community, demonstrating that absolute value of the t-statistics is significantly greater than zero. Null distributions could be obtained by permuting control/invade labels of each species within each community.

Fig. S17, S18: Similar to Fig. S15 and S16, but now there are different invader species for each community. The authors could perform a paired t-test for each invader species, pooling observations across communities. A null could be generated for each invader species by permuting control/invade labels within each species within each community.

Figs. S15-S18: Is there any relationship between the relative abundance of the invading species and the change in relative abundance of the remaining species between the control/invade treatments? This may not be the most appropriate analysis for compositional data, but you plot the relative abundance of the invading species vs. the mean difference in relative abundance between treatments for the remaining species, do you see a clear relationship?

Fig. S19-S21: The authors' claims about the lack of correlations appear correct but I think the correlation coefficient should be provided along with the non-significant P-value. Statistical significance can be assessed by permuting x and y vectors. Permutations should be constrained on species identity for Fig. S19.

Minor Comments

Both passive voice and active voice are used throughout the main manuscript and supplement. Consult the journal style guide and use the appropriate voice throughout the manuscript.

Line 76: Should "occupy all available niches and resources" be "occupy all available niches by consuming all resources"?

Line 85-86: Given the experiment performed, would be more apt to say that the question is whether the dynamics are stationary with respect to deterministic fluctuations?

Line 192: Should "effect" be plural?

Line 235: Is "niches and resources" redundant here? My understanding that the niche is resource niche in the context of this study.

Lines 304-305: "stronger secondary effect" "stronger secondary effects"

Lines 378-379: Grammar.

Lines 379-380: "invasion probability to" "invasion probability of"

Line 389: "invaders successfully invade" or "an invader successfully invades" ?

Line 437: "high nutrient" "high nutrient conditions" or something similar.

Supplement lines 75-76: Please summarize the DADA2 parameters used in your script and make your DADA2 pipeline available in a public code repository.

Supplement lines 79-81: Please make your raxml code available in a public repository.

Supplement lines 123-125: This sentence reads as if steady state is defined as the state where community properties change with time. Is this supposed to be the case?

Supplement lines 144-145: How is a window of time of 100 units analogous to a timescale of 24 hours in the experiment? Were growth rates in the simulation parameterized to correspond to growth rates in the experiment?

Supplement lines 167-168: Can you plot the results from varying the choice of time window?

Supplement lines 174: "Algorithm" should be plural.

Fig S1: Would an alternative color scale help the reader? It does not appear that colors are assigned based on taxonomy. Assigning different shades of a given color to the species belonging to a given phylum may help with visualization (e.g., Firmicutes get different shades of blue, etc.).

Figs. S8-S11: Can you plot the extinction threshold as a dashed/dotted horizontal line?

Fig. S12: Consider plotting the cutoff of 0.05 as a horizontal dashed line for reference in subplots a and b and as a vertical dashed line in subplots c and d.

Fig. S14: How were priority effects quantified? What does the y-axis of the priority effect plot represent?

References

- Cornish-Bowden, A. (2012). *Fundamentals of enzyme kinetics* (4th, completely revised and greatly enlarged edition ed.). Wiley-Blackwell.
- Garud, N. R., Good, B. H., Hallatschek, O., & Pollard, K. S. (2019). Evolutionary dynamics of bacteria in the gut microbiome within and across hosts. *PLOS Biology*, 17(1), e3000102. <https://doi.org/10.1371/journal.pbio.3000102>
- Houtsma, P. C., Kant-Muermans, M. L., Rombouts, F. M., & Zwietering, M. H. (1996). Model for the combined effects of temperature, pH, and sodium lactate on growth rates of *Listeria innocua* in broth and Bologna-type sausages. *Applied and Environmental Microbiology*, 62(5), 1616–1622.
- Marsland, R., Cui, W., Goldford, J., & Mehta, P. (2020). The Community Simulator: A Python package for microbial ecology. *PLOS ONE*, 15(3), e0230430. <https://doi.org/10.1371/journal.pone.0230430>
- Shoemaker, W. R., Sánchez, Á., & Grilli, J. (2023). Macroecological laws in experimental microbial systems (p. 2023.07.24.550281). *bioRxiv*. <https://doi.org/10.1101/2023.07.24.550281>
- Snedecor, G. W., & Cochran, W. G. (1989). *Statistical methods* (8th ed). Iowa State University Press.

Reviewer #3 (Remarks to the Author):

In this manuscript, Hu et al. aim to identify characteristics of microbial communities that determine their invasibility. They first carried out experimental invasions with assembled communities, finding that more diverse communities are more invulnerable. To further explore this positive invulnerability-diversity relationship, a Lotka-Volterra model was used to predict the effect of changing interspecies interaction strength and species pool size on invulnerability. This showed that decreasing interaction strength and pool size increased invulnerability. This was confirmed experimentally by changing the concentration of glucose and urea to tune interaction strength, and by changing species pool size. These three key determinants of invulnerability (interaction strength, pool size, dynamical regime) determine the survival fraction, defined as the fraction of the initial species pool that survives the assembly process. The survival fraction correlates positively with community invulnerability, serving as a unifying predictor. They find that under strong interaction strength, the invasion probability is lower than the survival fraction, indicating a priority effect. The strength of interactions is also shown to determine the impact of invasion on the resident community. Finally, the properties of invaders were briefly discussed.

I see this as a valuable and novel contribution to the field. While the long-standing biotic resistance hypothesis predicts that more diverse communities should be less invulnerable due to niche filling, empirical evidence for this is mixed. By considering how diversity is achieved, the authors show that invulnerability can be predicted from community features, which has not been done previously. A particular strength of the manuscript is the integration of modelling and experiments to gain mechanistic insight into relationships observed experimentally.

I have two major comments regarding the tuning of interspecies interactions and determining the impact of invasion. In both cases, limitations of the findings of the manuscript need to be made more explicit.

The authors state that increasing interaction strength decreases invasion probability. This is done experimentally by increasing the strength of competitive interactions. However, previous work (ref 30) has demonstrated that the type of interaction between the invader/residents, or between residents, impacts the invasion outcome. For example, positive interactions (e.g. facilitation) between the invader and resident community increases invasion probability (as posited by the diversity begets diversity hypothesis). Increasing interaction strength in this case could increase community invulnerability, which is a possibility not considered by this study. While I recognise that experimental manipulation of interaction strength is difficult, I would recommend running the model with a distribution of interactions that considers positive interactions, to show the generality of their conclusions. If this has already been done (this is not clear) this should be included in the supplement.

The limitations of the study in determining the impact of successful invasions on the structure of the resident community should be more explicit. The impact of successful invasions on the structure of the resident community was measured experimentally by the fold change in biomass, and the invasion effect on the community using 16s sequencing data. However, the fold change in biomass provides little insight on invasion impact. It is not possible to distinguish between the change in biomass of the resident community, and the increase in biomass of the successful invader. Thus, a lower change in biomass in the low nutrient regime may be due to the fact that the invader has less nutrients to grow on, and the converse for the high nutrient regime. Moreover, while species abundance data shows a statistically significant difference in community composition in the higher nutrient regime, it is important that the authors provide information about the effect size. For example, as a 95% confidence interval for the difference between the invasion success under high nutrient conditions and the invasion success under low nutrient conditions. Finally, it is hard to conclude what the actual effect of invasion is using the selected measures. For example, though species composition may have changed, the invader could be functionally redundant with the species that were excluded, resulting in no change to community functioning. While a comprehensive assessment of invasion impact is out of the scope of this paper, I would recommend for these limitations to be discussed explicitly in the manuscript.

Minor points:

- I strongly recommend including explicit definitions of key terms, which would improve clarity of the manuscript. For example, the definition of community diversity as the number of species that survive the assembly process is not made explicit in line 152. What 'rich dynamics' (line 191) or richness (line 195, 222 and others) refers to is also unclear to me.
- The findings of this study could be embedded more explicitly into existing concepts in invasion ecology, which is dominated by

niche theory and resource competition. For example, in lines 175-176 the authors state that fluctuating communities are more invisable. The basis of this relationship is not made clear to the reader - I assumed that this means more niches are open at any given point in time. Moreover, why strong interspecies interactions and a larger species pool decrease invasibility is not discussed. It would also be helpful to discuss this more in the conclusion.

- I think that for those unfamiliar with previous work from this group, how interaction strength, species pool size and dynamical regime combine to determine survival fraction is difficult to understand. More explanation about how these features interface is necessary for clarity of the manuscript, since they are not independent. For example, it is not immediately clear why there are stable/fluctuating regimes for communities under high nutrient conditions, and not for low nutrient conditions

- The measure of dispersion used is not included in the text of the manuscript (i.e., on line 148, 153, 154, 172 and more, parameter estimates are given as mean +/- X, where X is a measure of dispersion, but which measure it is is not defined).

Defining the measure of dispersion is crucial to allow for interpretation of the findings. Some figure legends mention that error bars represent the standard error of the mean, which I assume is the measure used by the authors, but this should be clearly stated in the text of the manuscript.

- State how invaders were chosen (line 146) - I assume they were chosen at random?

- In the section discussing the model (lines 178-190), explicitly state at what relative abundance the invader is introduced relative to the community

- In the representative time series of fig 2a, invasions are shown to cause a weak effect on the community when the invader grows to a low abundance relative to the rest of the community, whereas the invader causes a strong effect when it grows to high abundance. Is this true across other simulations, or in experiments? If this is the case, it would suggest that there is another factor that influences invasion effect aside from interaction strength.

- I would include (strong interaction) or (weak interaction) every time high/low nutrient conditions are mentioned for clarity. This is missing now in line 240 and others

- Remove "while resident community determining invasion outcome" from line 322-323. Invader properties have not been sufficiently explored in this manuscript to claim they don't play an important role in determining invasion outcome.

Version 1:

Decision Letter:

Dear Jeff,

As per my previous email, we received the reviewers' comments on your revised manuscript "Collective dynamical regimes predict invasion success and impacts in microbial communities" (NATECOLEVOL-24020551A) and we decided in principle to publish it in Nature Ecology & Evolution, pending minor revisions and compliance with our editorial and formatting guidelines.

We are now performing detailed checks on your paper and will send you a checklist detailing our editorial and formatting requirements in about a week. Please do not upload the final materials until you receive this additional information from us.

[redacted]

Reviewer #1 (Remarks to the Author):

The authors have addressed all my concerns, and added useful new analyses and discussions that clarify the manuscript and the modeling choices they made. I have no more concerns that would need to be resolved before publication.

As an extremely minor point, I found the spread of the points in Fig. 5c and d to be a bit large, to the point where my eye didn't immediately parse the two clouds as separate categories - but that's presumably a matter of taste.

Reviewer #1 (Remarks on code availability):

No readme files, but the data looks complete. Both the sequencing data analysis code and the simulation code is clear and well-documented. I did not attempt to run the code, but as far as I can tell, the provided code only gives the backbone of the simulations and does not include code needed to reproduce the figures directly.

Reviewer #2 (Remarks to the Author):

Review #2

This is an impressive revision, and the quality reflects a high level of effort from the authors. The authors addressed all my comments and gave thoughtful, much appreciated answers. I believe the manuscript should be published.

My sole major comment is that the GitHub repo appears to be incomplete. Some specifics:

- The file names do not have extensions. I believe that the files are written in matlab, but I do not use that language so I cannot be sure.

- There is no README file. Some information about the working environment (e.g., packages) and how to run the code would be helpful, as well as a link to the Dryad repository.

- There are functions called in the files that do not appear to be defined or imported from other files in the repo. For example, LV_compute_invasion is called in LV_Invade but I do not see it defined in that file or in any of the other files in the repo.

The Dryad repository looks good.

Below are some minor comments that do not require a response.

- The discussion of how pH mediated interactions can shape the strength of interaction in gLV is helpful. In general, I think a derivation of pH mediated interactions from the starting point of a consumer-resource model is needed, even if it's just for pedagogical reasons and all the heavy modeling is ultimately done using gLV, but such modeling efforts lie outside the scope of the paper. All-in-all the additions to the supplement make it a valuable resource.

- The elaboration on the connection to ergodicity in response to R1 was very helpful.

- The new color scheme should aid readers.

- "In our group, we have been investigating the dependence of community diversity and dynamics on resource concentration across various consumer-resource models (e.g., linear growth or Monod growth)." Very much looking forward to reading this.

- The added analyses for the survival threshold are rigorous. The sequencing depth justification is particularly useful since my group has encountered studies where the order of magnitude of demographic manipulations is not justified given the depth of sampling (i.e., # reads).

Reviewer #2 (Remarks on code availability):

My sole major comment is that the GitHub repo appears to be incomplete. Some specifics:

- The file names do not have extensions. I believe that the files are written in matlab, but I do not use that language so I cannot be sure.

- There is no README file. Some information about the working environment (e.g., packages) and how to run the code would be helpful, as well as a link to the Dryad repository.

- There are functions called in the files that do not appear to be defined or imported from other files in the repo. For example, LV_compute_invasion is called in LV_Invade but I do not see it defined in that file or in any of the other files in the repo.

I was unable to run the code as-is.

Reviewer #3 (Remarks to the Author):

I would like to thank the authors for their thorough revision of this manuscript. All my concerns have been addressed and I have no further comments.

Reviewer #3 (Remarks on code availability):

The code is usable but needs a README file for further clarity.

Version 2:

Decision Letter:

25th November 2024

Dear Professor Gore,

We are pleased to inform you that your Article entitled "Collective dynamical regimes predict invasion success and impacts in microbial communities", has now been accepted for publication in Nature Ecology & Evolution.

Over the next few weeks, your paper will be copyedited to ensure that it conforms to Nature Ecology and Evolution style. Once your paper is typeset, you will receive an email with a link to choose the appropriate publishing options for your paper and our Author Services team will be in touch regarding any additional information that may be required

Due to the importance of these deadlines, we ask you please us know now whether you will be difficult to contact over the next month. If this is the case, we ask you provide us with the contact information (email, phone and fax) of someone who will be able to check the proofs on your behalf, and who will be available to address any last-minute problems. Once your paper has been scheduled for online publication, the Nature press office will be in touch to confirm the details.

Acceptance of your manuscript is conditional on all authors' agreement with our publication policies (see www.nature.com/authors/policies/index.html). In particular your manuscript must not be published elsewhere and there must be no announcement of the work to any media outlet until the publication date (the day on which it is uploaded onto our web site).

Please note that *Nature Ecology & Evolution* is a Transformative Journal (TJ). Authors may publish their research with us through the traditional subscription access route or make their paper immediately open access through payment of an article-processing charge (APC). Authors will not be required to make a final decision about access to their article until it has been accepted. [Find out more about Transformative Journals](https://www.springernature.com/gp/open-research/transformative-journals)

Authors may need to take specific actions to achieve [compliance](https://www.springernature.com/gp/open-research/funding/policy-compliance-faqs) with funder and institutional open access mandates. If your research is supported by a funder that requires immediate open access (e.g. according to [Plan S principles](https://www.springernature.com/gp/open-research/plan-s-compliance)) then you should select the gold OA route, and we will direct you to the compliant route where possible. For authors selecting the subscription publication route, the journal's standard licensing terms will need to be accepted, including [self-archiving and license to publish](https://www.nature.com/nature-portfolio/editorial-policies/self-archiving-and-license-to-publish). Those licensing terms will supersede any other terms that the author or any third party may assert apply to any version of the manuscript.

We welcome the submission of potential cover material (including a short caption of around 40 words) related to your manuscript; suggestions should be sent to Nature Ecology & Evolution as electronic files (the image should be 300 dpi at 210 x 297 mm in either TIFF or JPEG format). Please note that such pictures should be selected more for their aesthetic appeal than for their scientific content, and that colour images work better than black and white or grayscale images. Please do not try to design a cover with the Nature Ecology & Evolution logo etc., and please do not submit composites of images related to your work. I am sure you will understand that we cannot make any promise as to whether any of your suggestions might be selected for the cover of the journal.

You can generate the link yourself when you receive your article DOI by entering it here: <http://authors.springernature.com/share>.

[redacted]

P.S. Click on the following link if you would like to recommend Nature Ecology & Evolution to your librarian <http://www.nature.com/subscriptions/recommend.html#forms>

** Visit the Springer Nature Editorial and Publishing website at http://editorial-jobs.springernature.com?utm_source=ejP_NEcoE_email&utm_medium=ejP_NEcoE_email&utm_campaign=ejP_NEcoE for more information about our career opportunities. If you have any questions please click [here](mailto:editorial.publishing.jobs@springernature.com).

Reviewers' comments:

Reviewer#1 (Remarks to the Author):

In this manuscript, Hu et al. combine experiments and theory to elucidate the factors determining community invasions. They find that while diversity-invasibility relationships can be both positive and negative depending on the factors regulating diversity, the fraction of surviving species directly predicts invasibility. This manuscript leverages their recent insights on the dynamical phases of microbial ecosystems to provide a timely and relevant exploration of invasions.

The results are interesting and the research is carried out soundly. My comments are primarily aimed at making the manuscript easier to follow.

My first main comment concerns the exposition of the different diversity-invasibility relationships between Fig. 1 and Fig. 2/3, which on a first read was not immediately obvious. In Fig 1, more species and higher interaction strength makes for fluctuations and greater diversity, and fluctuating communities are more likely to be invaded; but both theory and Fig. 3 show a negative correlation between invasion probability and pool size/interaction strength. I naively expected increased pool size and stronger interactions to translate to a higher chance of fluctuations (as in their recent paper), and thus greater richness and higher invasion probability. It is later revealed where this apparent contradiction comes from, but I believe it would be immensely helpful if this contradiction was highlighted early on. This issue is made more severe by the use of three different terms to describe the probability of successful invasions (invasion probability and invasion resistance, also invasibility) – especially because these have opposite meanings, I had to read this section of the manuscript carefully to understand its meaning. A single term would be easier to understand.

We thank the reviewer for this helpful comment to improve the readability of the manuscript. When we sample many communities under the same species pool size and the same nutrient concentration (interaction strength), we found the fluctuating communities are more diverse and invasive than the stable ones. However, if the species pool size or nutrient concentration (interaction strength) varies in the experiment or simulation, then fluctuating communities may not be more diverse and invasive than stable ones, as shown in Fig. 3. A more accurate statement should be: The average community invasibility decreases with species pool size and interaction strength (Fig. 2f and Fig. 3). Fluctuating communities are on average more diverse and invasive than stable communities under the same nutrient condition and species pool size. We now clarify in the caption of Fig. 1 and 2 that the fluctuating communities are more invasive than stable ones under the same species pool size and nutrient conditions (average interactions), rather than a universal statement across different conditions. We follow the reviewer's comment and avoid using any "invasion resistance" in the manuscript, which has opposite meanings to "invasibility" and "invasion probability". We further clarify this point in the early text on page 4 of the main text to avoid confusion:

Our experimental tests of invasion demonstrate that, for fixed environment and species pool size, more diverse communities are more invasive because fluctuating communities are both more diverse and more susceptible to invasion. However, we will show later that when species pool size or nutrient concentration is varied, this relationship does not always hold.

We also highlight this point on page 5 of the main text:

It is important to note that although fluctuating communities exhibit larger invasion probability than stable communities under the same conditions, stable communities can still yield larger invasion probability under weaker interaction strength $\langle \alpha_{ij} \rangle$ or smaller species pool size S (Fig. 2d-f).

My second main comment concerns the notion of priority effects. Here, as I understand it, the authors use this term basically to mean that a community of strongly interacting species has established itself, which makes invasions less likely (e.g., because all niches are filled or because antagonistic interactions are balanced out) but I am not convinced that this is the sense in which most ecologists tend to use that term; to me, the more common use appears to be referring to particular species whose presence or absence steers the community into different alternative stable states. Here, however, the community is already in a stable state and a new species is trying to invade, and thus "priority effect" does not apply in my view. I suggest a different way of describing this effect here, or, at the very least, the sentence in line 244-246 needs more explanation. Similarly, I find the use of "alternative stable states" to be unclear since that is not really the subject of this paper.

We appreciate the reviewer for highlighting this important point. Our interpretation and use of the priority effect are in line with the reviewer's comments. In ecology, the "priority effect" refers to the phenomenon where the order and timing of species arrival in a community influence the subsequent structure and dynamics of that community. Specifically, the species that establish themselves first can significantly affect the community's composition and the success of later-arriving species. In our case we claim priority effect emergent under strong interaction because our model and experiment show that the later-arriving species (invader species) display lower surviving probability (invasion probability) than surviving probability of initial species assemble from the pool (early-arriving species) (as shown in Fig. 3c and Fig. 4c). Under low nutrient (weak interaction), the later-arriving species (invader species) display similar surviving probability (invasion probability) with surviving probability of initial species assemble from the pool (early-arriving species) (as shown in Fig. 3c and Fig. 4c), which indicate there is no priority effect under low nutrient (weak interaction).

Based on our experiments and simulations, we found that the existence of alternative stable states is closely related to the presence of priority effects. As demonstrated in previous theories (Fried, Shnerb, and Kessler 2017; Bunin 2017) and in our experiment (new Supplementary Fig. 22), we observed that the same set of species can reach different alternative stable states when starting from different initial compositions under high nutrient concentrations (strong interactions). In contrast, under low nutrient concentrations (weak interactions), the species consistently converge to a single stable state, meaning that the order of species arrival does not affect the final community structure, indicating the absence of a priority effect. In this context, we consider the priority effect to be equivalent to the presence of alternative stable states. The existence of alternative stable states implies that different initial compositions or the order of species arrival can lead to different final community compositions, which aligns with the definition of the priority effect. Conversely, when no alternative stable state exists, different initial compositions or the order of species arrival result in the same final community composition, which corresponds to the absence of a priority effect.

We therefore propose that the alternative stable states observed under high nutrient conditions exhibit a priority effect, which explains why invasion probability is generally lower than the survival fraction in these conditions (bottom left points in Fig. 3c). These stable communities strongly resist invasion due to their established alternative stable states. As the reviewer suggested, we've revised the manuscript to clarify the use of the terms "priority effect" and "alternative stable state".

Supplementary Fig. 22. Communities reach global stable states under low nutrient conditions (weak interaction) but reach alternative stable states under high nutrient conditions (strong interaction) when starting from different initial species compositions. Each bar represents a final community stable state after 7 daily dilution cycles starting from a particular initial condition. The communities assemble from the same species in the original pool ($S=12$ for the left and middle communities, $S=24$ for the right communities), under low-nutrient (weak interaction) and high-nutrient (strong interaction) conditions, respectively. For each initial condition, one of the species occupies 99% volume of the initial inoculum, while the other species in the pool together occupy only 1% volume of the initial inoculum. The results show that these very different initial species compositions lead to the same global stable state under low nutrient conditions (weak interaction) but result in alternative stable states under high nutrient conditions (strong interaction). This indicates the presence of alternative stable states and priority effect under high nutrient (strong interaction), where the initial composition influences the final community structure.

We revised the paragraph on page 6 and 7 with an additional explanation as the reviewer suggested:

Despite the experimentally observed correspondence between invasion probability and survival fraction, we note that the invasion probability for communities under high nutrient (strong interaction) conditions is usually lower than their survival fraction (i.e. the majority of points on the bottom left are below the diagonal line on Fig. 3c). We now discuss how this can be interpreted in terms of priority effects or alternative stable states. If the assembly of species in the community does not depend on the order of species arrival (i.e., no effect of history), the survival fraction of species assembled from the initial pool should be statistically equal to the survival probability of

the species that invade the communities later. In ecology, a "priority effect" refers to a situation in which the community structure is influenced by the order and timing of species' arrival (Sprockett, Fukami, and Relman 2018; Debray et al. 2022). We thus interpret the mismatch between invasion probability and survival fraction under high nutrient concentrations (strong interactions) as evidence of priority effects in the community assembly under strong interactions in our experiment. Early-arriving species can dominate by making it more challenging for subsequent invaders to establish, leading to a lower invasion probability than the survival fraction of species in the original pool. Under weak interactions, however, the colonization probability of invader species is similar to the probability of a species in the initial pool surviving the process of community assembly (Fig. 3c) (Case 1990). An emergent priority effect in communities composed of strongly interacting species can be explained by the presence of alternative stable states with different species compositions, which we observe in additional experiments (Supplementary Fig. 22). Early-arriving species establish dominance, making it more difficult for later-arriving invaders to successfully establish themselves, by inhibiting their growth at low abundance. These alternate stable states thus explain why the invasion probability is generally lower than the survival fraction from the initial pool under high nutrient (strong interaction) conditions (Case 1990; Hu et al. 2022).

Minor comments:

1) L110: "The lower probability" – compared to what?

We appreciate the reviewer for pointing out the need for clarification. The text should specify that the lower invasion probability is compared to the survival fraction. We have revised the manuscript to make this comparison explicit. The revised Text is:

"The lower invasion probability compared to the survival fraction suggests a priority effect, whereby earlier invaders preclude later ones from growing from small abundances."

2) L174: "including species interaction strength" – is it clear that different synthetic communities can be characterized by the same average interaction strength? Perhaps some initial compositions have strong antibiotic producers or very similar metabolic profiles

We thank the reviewer for this valuable question. We have clarified this point by specifying "for fixed species pool size and species interaction strength regime (nutrient concentrations)." We describe community conditions based on sampling species interactions from a given distribution. For example, in Fig. 2b, communities were sampled from the same species pool size ($S=20$) and a uniform distribution of interaction α_{ij} with a mean interaction of 1.8. Each sampled community has a unique but similarly distributed mean interaction. Our previous work shows that average pairwise interactions increase from low to high nutrient concentrations (Fig. 2B of Hu et al. *Science* 2022). Although the statistical distribution may vary with limited species sampling, species cultured in the same high nutrient concentration are in a strong interaction strength regime compared to those in a weak interaction strength regime under low nutrient concentrations. We have updated the manuscript on page 4 to reflect this clarification:

Our experimental tests of invasion demonstrate that, for fixed species pool size and species interaction strength regime (nutrient concentrations), more diverse communities are more invulnerable because fluctuating communities are both more diverse and more susceptible to invasion. However, when species pool size or nutrient concentration is varied, this relationship does not always hold (Fig 2 and 3).

3) L227: correlation on -> with

We thank the reviewer for this point. We have edited the text to read “positive correlation of invasibility with survival fraction” in the main text. The revised Text is:

"The results show a strongly positive correlation of invasibility with survival fraction."

4) L239: Why should we expect a 1:1 correspondence between survival fraction and invasion probability? This only becomes clear in Fig 4b

We thank the reviewer for this insightful question. If the assembly of species in the community does not depend on the order of species arrival (i.e., no history effect and priority effect), the survival fraction of species assembled from the initial pool should be statistically equal to the survival probability of the species that invade the communities later. If there is no priority effect and the order of species arrival does not matter, then the survival fraction is expected to approximately correspond with invasion probability, as shown in Fig. 4b. We have added this explanation to page 6 and 7 of the main text to clarify this point:

Despite our experimentally observed correspondence between invasion probability and survival fraction, we noted that the invasion probability for communities under high nutrient is usually lower than their survival fraction (majority of points on the bottom left (high nutrient) are below the diagonal line on Fig. 3c). If the assembly of species in the community does not depend on the order of species arrival (i.e., no history effect and priority effect), the survival fraction of species assembled from the initial pool should be statistically equal to the survival probability of the species that invade the communities later.

5) L280: ergodicity likely needs a bit more explanation to be clear

We follow the reviewer’s suggestion and have further clarified the concept of ergodicity in the supplementary materials and mentioned in the main text. As studied and shown in previous theoretical studies (Altieri et al. 2021; Pearce et al. 2020; Bunin 2017), ergodicity in the context of chaotic fluctuations means that the community's state is memoryless. The dynamical trajectories of species composition do not depend on the community's history and do not reach alternative stable states or alternative dynamical attractors. Therefore, different orders of species arrival or different initial species compositions do not lead to different community states in the chaotic fluctuation regime. We have added this explanation to improve clarity. The revised text in the supplementary materials is:

"As studied and shown in previous theoretical studies (Altieri et al. 2021; Pearce et al. 2020; Bunin 2017), ergodicity in the context of chaotic fluctuations means that the community's state is memoryless. The dynamical trajectories of species composition do not depend on the community's history and do not reach alternative stable states or alternative dynamical attractors. Therefore, different orders of species arrival or different initial species compositions do not lead to different community states in the chaotic fluctuation regime."

We referred to this explanation in the main text on page 8:

which can be explained by its ergodicity (Bunin 2017; Pearce et al. 2020; Altieri et al. 2021) (see a technical discussion in Supplementary Materials).

6) L285: definition of invasion effect is slightly unclear. Is this the same definition as in L296?

We further clarify the definition of invasion effect in the text. The definition of invasion effect in L285 is the same as the definition in L296. The invasion effect is measured as the proportion of change in surviving species before the invasion ($t=10^3$) and after the invasion ($t=2 \times 10^3$), calculated as 1 minus the ratio of the number of overlapping species (number of species surviving both at $t=2 \times 10^3$ and $t=10^3$) to the total number of species (number of species surviving at $t=10^3$ or $t=2 \times 10^3$). Mathematically the invasion effect is calculated through: $1 - (\text{number of overlapping species} / \text{total number of species})$. We have revised the text to ensure this definition is clear and consistent throughout. The revised text on page 8 of the main text is:

the invasion effect is quantified as the proportion of change in surviving species before the invasion ($t=10^3$) and after the invasion ($t=2 \times 10^3$) (invasion effect = $1 - (\text{number of overlapping species} / \text{total number of species})$) (Fig. 4d)

7) L317: SI Fig 17 does not show an Allee effect for invaders 2/3

We thank the reviewer for pointing out this issue. To see the Allee effect for invaders 2 and 3, we need to look at the monoculture growth curve for these invaders, which is shown in SI Fig. 19. The growth curves of invaders 2 and 3 (blue and purple curves) indicate that they do not grow to a well-detectable signal on the OD reader over 20 hours, demonstrating an Allee effect. We have revised the main text to reference SI Fig. 19 for clarity. The revised text in the Supplement is:

"To see the Allee effect for invaders 2 and 3, notice that the growth curves of invaders 2 and 3 (blue and purple curves) indicate that they do not grow to a well-detectable signal on the OD reader over 20 hours, demonstrating an Allee effect of these two species."

8) Fig 2f/4c: Legend missing for the gray lines

We thank the reviewer for raising this important point. We have added legends to Fig. 2f and Fig. 4c to explain the gray lines as shown below and in the main text figures. The solid lines represent the stability boundary, and the dashed lines represent the surviving boundary, which is further explained in the new caption of the figures.

Fig. 2f, Increasing species pool size and interaction strength leads to a decrease in invasion probability. The communities experience the extinction of species and loss of stability when crossing the dashed gray line (surviving boundary) and solid gray line (stability boundary), respectively. The color maps depict the mean value over 1000 simulations.

Fig. 4c, Increasing species pool size and interaction strength leads to the emergence of priority effect, where the invasion probability of resident communities is smaller than their species survival fraction. The communities experience the extinction of species and loss of stability when crossing the dashed gray line (surviving boundary) and solid gray line (stability boundary), respectively.

9) Fig 4a: it is very hard to parse the different colors

We appreciate the reviewer for pointing this out. We have chosen a different set of colors to make the different lines easier to distinguish as shown below.

Fig 4. The Lotka-Volterra model predicts a universal correspondence between invasion probability and survival fraction. **a**, The dependence of invasion probability on final richness of resident communities is qualitatively different depending upon how the richness is changed. Invasion probability positively correlates with richness when varying interaction strength or when randomly sampling communities with a fixed species pool size and interaction strength distribution. Invasion probability can decrease with community diversity when varying species pool size. **b**,

Invasion probability is approximately equal to the survival fraction of species in the resident communities, no matter how we change richness, species pool or interaction strength.

Reviewer #2 (Remarks to the Author):

Summary

Overall, I found that the study addressed interesting questions and the choice to connect experiments with minimal models of community dynamics seemed like the appropriate choice. However, several decisions made and details about the experiment seem unclear. For example, it is unclear to me how 1) a generalized Lotka-Volterra (gLV) serves as an appropriate model for pH-mediated effects, 2) how fluctuating/stable communities were classified, 3) whether their experimentally-imposed migration rate was sufficient to induce deterministic fluctuations.

We appreciate the reviewer's feedback and suggestions on our manuscript. In our response letter, we provide a point-by-point answer to all the questions and suggestions from the referees. For convenience, the original comments from the reviewers appear in blue-color font and new text that now appears in either the main text or supplement is included in red.

Below is a brief discussion regarding these three major points. More detailed responses can be found in our specific responses to the reviewer's comments.

1) Generalized Lotka-Volterra (gLV) Model for pH-Mediated Effects:

As a phenomenological model, the generalized Lotka-Volterra (gLV) framework describes the interaction between species through a pair-wise interaction matrix, capturing how the abundance of one species influences the growth of others. The gLV interaction matrix can incorporate various mechanisms, such as resource competition, cross-protection, cross-feeding, and pH-mediated interactions, if the interaction parameters are chosen based on experimental data.

The gLV model provides a coarse-grained description of interaction signs and strength without delving into specific mechanisms, which is useful when interaction mechanisms are not fully known or when the focus is on collective community behaviors (e.g., fluctuations).

In our study, pH-mediated interaction is one of several interaction mechanisms. A model solely based on pH-mediated interactions cannot explain some experimental observations. However, a model combining pH-mediated effects and gLV-type interactions can capture key experimental observations, with pH-mediated effects reducing to gLV-type interactions in the adiabatic limit.

2) Classification of Fluctuating/Stable Communities:

Communities were classified as fluctuating or stable based on the time series of total biomass and species composition. A community is classified as fluctuating if the variation in its total biomass and species composition over time exceeds a critical threshold. Detailed criteria and thresholds used for classification will be discussed in the following responses to specific comments.

3) Experimentally-Imposed Migration Rate and Deterministic Fluctuations:

We have included additional experimental and simulation results to demonstrate that our imposed migration rate leads to deterministic fluctuating or stable states. This point will be elaborated upon in detail in the following responses to specific comments.

The visualizations are effective and thorough. However, the authors often make claims about the interpretation of visualizations that need to be backed up by statistical tests. I have attempted to identify reasonable statistical tests when a test is lacking. There are also several instances where further clarification could help a broad readership understand the study. These instances are pointed out under both Major and Minor Comments.

We have now clarified the results of statistical tests when making claims about the interpretation of visualizations throughout the manuscript. We have conducted and reported appropriate statistical tests to support our visual interpretations, ensuring that our claims are robust and well-substantiated. Additionally, we have followed the reviewer's advice and added further clarification to points that may not be clear to a broad audience. These clarifications will be discussed in detail in our responses to the specific major and minor comments provided by the reviewer.

Major comments:

1. There is no explicit consideration of resources in their model. This makes sense in some respects, as the gLV can be derived in the adiabatic limit of certain consumer-resource models. Furthermore, their experimental manipulations of invasion are demographic rather than environmental, so it makes sense that a phenomenological model that does not explicitly consider microscopic interactions of resource consumption is sufficient. Progress has recently been made using phenomenological models to investigate the macroecological consequences of demographic manipulations (Shoemaker et al., 2023), which may provide outside support for the authors' modeling decisions. At a minimum the authors should provide additional justification for their modelling choices and why they are appropriate for their mode.

We thank the reviewer for this insightful comment on justifying our choice of the generalized Lotka-Volterra (gLV) model in this work. As a phenomenological model, gLV describes species interactions through a pair-wise interaction matrix, capturing how the abundance of one species influences the growth of others. The interaction matrix in the gLV framework is a high-level description that can incorporate various mechanisms, including resource competition, cross-protection, cross-feeding, and pH-mediated interactions.

As the reviewer pointed out, the gLV model can be derived in the adiabatic limit of certain consumer-resource models. We found that a pH-explicit model can also be reduced to gLV in the adiabatic limit, where the gLV interaction matrix is a function of the pH effect. We would like to emphasize the following points:

1) Multiple Interaction Mechanisms:

pH-mediated interaction is not the only mechanism in our experiment. Some communities show fluctuating biomass while maintaining stable pH levels, indicating other mechanisms, such as cross-toxin interactions, are at play, particularly under high nutrient concentrations.

2) Limitations of Solely pH-Mediated Models:

Our model considering only pH-mediated interactions cannot explain all experimental observations. For instance, our experiments and gLV simulations show fluctuating communities with higher diversity and invasibility (Hu et al. 2022), while our pH-only models predict lower diversity and invasibility (Supplementary Fig. 10-11 of Hu et al. 2022) (Hu et al. 2022), contrary to our experimental findings.

3) Combined Model Approach:

We developed a model that integrates pH-mediated environmental effects with gLV-type species interactions. This combined framework successfully explains key experimental observations and, crucially, can be reduced to a pure gLV interaction matrix in the adiabatic limit—where the pH dynamics evolve much faster than species growth, or when pH reaches stable state. This reduction explicitly demonstrates how pH influences the structure of the gLV interaction matrix

These justifications support our use of the gLV model and align with recent progress in using phenomenological models for demographic manipulations (Shoemaker et al., 2023). We have added this explanation to the manuscript to provide additional justification for our modeling choices.

2. However, the authors also manipulate resource concentration as a means to manipulate the strength of competitive interactions. This manipulation is done by increasing resource concentration, changing the pH, which is a feature of the environment that does not appear in the model and does not appear to be measured. So, while their invasion manipulations are demographic, it reads as though the difference in invasion outcomes between high and low nutrient treatments is driven by the environment. In the manuscript's current form, I imagine that it would be difficult for the reader to understand how the manipulation of resource concentration can be modeled as an increase in the strength of competitive interactions in a phenomenological gLV.

We appreciate the reviewer for this insightful comment. We have performed pair-wise coculture experiments among the species in the pool and found that the fraction of coexistence between pairs of species decreased, while the fraction of competitive exclusion (only one species survives) increased when nutrient concentration was increased (Hu et al. 2022). We now cite these pair-wise coculture results in the manuscript to demonstrate more clearly that the average interaction between species increases with nutrient concentration.

Previous work from our group (Ratzke, Barrere, and Gore 2020) has shown that increasing glucose and urea concentrations in the culture medium increases inter-species interaction strength among different soil bacterial species. Specifically, some species consume glucose and produce organic acids, leading to a decrease in pH, while others consume organic acids and increase the pH. Additionally, some species consume urea and produce ammonium, which also increases pH. These interactions modify the pH in different directions and affect species' growth and death rates, explaining the pH-mediated interaction under high nutrient concentrations.

Our experiments show that without glucose and urea, pH does not significantly change, indicating weak pH-mediated interactions. However, we also observed that communities can fluctuate in biomass without corresponding pH fluctuations (see figure below), suggesting that pH is not the sole driver of strong interactions under high nutrient concentrations. Our ongoing spent-medium culture experiments show that other types of inter-species interactions, such as cross-toxin effects, also increase with nutrient concentration, contributing to the overall interaction strength.

In our experiments, increasing nutrient concentration led to increased interaction strength and decreased community diversity. While existing resource-consumer models often display independence between resource concentration and interaction strength, our new pH model combines pH-mediated effects with gLV-type interactions, directly considering the quantitative interdependence between pH modification by species and pH effects on species growth, which can be measured experimentally.

We agree with the reviewer that directly relating nutrient concentration to interaction strength is an important research direction, and we are actively working on relevant projects. We have added these explanations to the manuscript to clarify how manipulation of resource concentration can be modeled as an increase in competitive interaction strength in a phenomenological gLV framework.

Figure. Microbial communities can exhibit fluctuations in biomass without corresponding fluctuations in pH in our experiment. The biomass (OD) of a microbial community fluctuates between days 6 and 10, while the pH time series stabilize during the same period (between days 6 and 10). The three curves represent three replicates of the same community.

3. The connection between interaction coefficients and growth-mediated pH is difficult to see given that pH tends to be incorporated into terms related to growth, both for models such as Monod growth (e.g., Houtsma et al., 1996) as well as Michaelis–Menten kinetics (chapter 10 of Cornish-Bowden, 2012). I suppose that pH-dependent competition coefficients could be derived if one started with a consumer-resource model where growth terms were pH-dependent and took the adiabatic limit. At minimum a form of the gLV needs to be derived so the reader can see the connection between increased resource concentration, change in pH, and increase in the strength of competitive interactions. However, it is also unclear that a phenomenological gLV is appropriate since the pH changes within a transfer cycle due to the growth of the community. Since the change in pH within a transfer cycle is on the same timescale of the growth rate, I do not see how a phenomenological gLV can serve as an appropriate model.

We thank the reviewer for this insightful comment. As explained in our response to major comment, we have found that existing resource-consumer models do not explicitly relate resource concentration to pH changes and interactions. Recognizing the importance of incorporating pH-dependent growth into the model, we developed a model combining pH effects with gLV-type interactions to explain our experimental observations, which are consistent with gLV predictions.

This pH model accounts for species' modifications of environmental pH and the pH-dependent growth of species. As the reviewer suggested, we found that this model can be reduced to a pure gLV-type interaction matrix in the adiabatic limit, allowing us to explicitly demonstrate how pH effects influence the phenomenological interaction matrix in the gLV model. In our approach, we considered linear functions for both species' modifications of environmental pH and the impact of environmental pH on species growth. This choice was made to capture the key features of community dynamics in a minimal model and because the linear form facilitates the reduction of

pH-mediated interactions into gLV-type interactions in the adiabatic limit. The results indicate that the presence of pH effects increases both the mean and standard deviation of the distribution of effective inter-species interaction strengths α_{ij}^* (see figure below). In the limit where gLV interaction strength is set to zero and only pH-mediated interactions exist, we still observe a distribution of effective inter-species interaction strengths (α_{ij}^*) (see figure below). These findings suggest that the presence of pH effects in the model leads to an increase in the mean and standard deviation of gLV-type interaction strengths (α_{ij}^*).

Furthermore, we found that as we increased the pH-mediated interaction strength and species pool size in the model, the invasion probability decreased (new Supplementary Fig. 23). When both pH-mediated and gLV-type interaction strengths were increased simultaneously, the invasion probability decreased in a manner similar to that observed when only the gLV-type interaction strength was increased (new Supplementary Fig. 23). In the presence of pH-mediated interaction strength in the model, we observed that the survival fraction was approximately equal to the invasion probability under various conditions (new Supplementary Fig. 23). These results suggest that our conclusion—that invasion probability decreases with interaction strength and species pool size—is robust to the presence of pH effects in the model. Meanwhile, the survival fraction serves as a good predictor of invasion probability in both the gLV and pH models. Our results support the idea that the gLV model phenomenologically describes species interactions within the community and predicts robust patterns in community invasion outcomes, which can be qualitatively reproduced in a pH model or a combined pH and gLV model.

Our previous measurements of pH and biomass dynamics within one transfer cycle (24 hours) show that pH dynamics are faster than population dynamics (Ratzke, Denk, and Gore 2018). Furthermore, the reduction of our pH model to gLV does not require time-scale separation when the pH and population dynamics reach stable states. More importantly, gLV provides a phenomenological and coarse-grained description of inter-species interactions averaged across different mechanisms and time scales, helping us characterize collective community dynamical regimes qualitatively.

In summary, we agree with the reviewer that incorporating a pH-explicit model is crucial for understanding how increasing nutrients and pH can enhance inter-species interactions. We have revised the manuscript to include this explanation, showing the derivation of the gLV form and its connection to pH-mediated interactions. In our group, we have been investigating the dependence of community diversity and dynamics on resource concentration across various consumer-resource models (e.g., linear growth or Monod growth). However, we have not observed that increasing resource concentration leads to a decrease in diversity and stability within these models, as seen in our experiments. We also explored a consumer-resource model with pH-dependent growth terms, but similarly, we did not observe a decrease in diversity and stability with increased resource concentration in any of the models we tested. We agree with the reviewer that it is both interesting and important to explicitly relate resource concentration to changes in pH and the subsequent increase in the strength of competitive interactions, this remains an open question in our ongoing projects. We have included a discussion of our new model, which combines pH-mediated interactions and gLV-type interactions, in the supplementary materials:

Theoretical alternatives to the Lotka-Volterra model

To directly model the pH-mediated interactions in our framework, we consider a minimal model where the pH value (p in the model) influences the species' per capita growth rates linearly, and, reciprocally, the species modify the environmental pH in a linear manner. This approach serves as a straightforward extension of the generalized Lotka-Volterra (gLV) model, which traditionally handles species interactions without considering environmental feedbacks such as pH.

In this extended model, the interaction between species and pH is twofold: species can alter the pH of their environment, and this altered pH, in turn, affects their growth rates. This dual influence allows the model to capture more complex ecological dynamics, where pH acts as a mediating factor that can shift the balance between competition, cooperation, and exploitation among species. Depending on the specific parameters chosen—such as the sensitivity of growth rates to pH changes and the extent to which species modify pH—the model can simulate a variety of interaction scenarios.

For instance, a species that raises the environmental pH could either inhibit or promote the growth of other species, depending on whether those species thrive in higher or lower pH conditions. Similarly, species that lower the pH could create environments that are hostile or favorable to others, depending on their pH preferences. This dynamic allows the model to represent competitive interactions (where species indirectly harm each other by altering pH), cooperative interactions (where species create favorable conditions for each other), and exploitative interactions (where one species benefits at the expense of another by altering the pH).

By incorporating pH-mediated interactions, this model adds a layer of realism to the gLV framework, making it more applicable to ecosystems where environmental factors like pH play a critical role in shaping species interactions. The linear relationships assumed in this model are not just mathematically convenient; they also provide a minimal yet powerful way to explore how environmental feedbacks can influence community dynamics. This extension allows for the modeling of a broader range of ecological scenarios, potentially offering new insights into how species coexist and compete in pH-sensitive environments.

$$\frac{dN_i}{dt} = N_i \left(\mathbf{1} - N_i - \sum_j \alpha_{ij} N_j + g_i p \right) \quad (2)$$

$$\dot{p} = -\delta p + \beta \sum_j k_j N_j \quad (3)$$

In the adiabatic limit, where the rate of pH change is much faster than the species' growth rates (time scale separation), or at steady state, we can assume $\dot{p} = -\delta p + \beta \sum_j k_j N_j = 0$. This implies that $p \cong \beta^* \sum k_j N_j$, where $\beta^* = \beta/\delta$. Substituting $p \cong \beta^* \sum k_j N_j$ into equation (2), we obtained:

$$\frac{dN_i}{dt} = N_i \left(\mathbf{1} - \frac{N_i + \sum \alpha_{ij}^* N_j}{K_i^*} \right) \quad (4)$$

where the effective gLV-type interaction strength is given by $\alpha_{ij}^* = \frac{(\alpha_{ij} - \beta^* g_i k_j)}{(1 - \beta^* g_i k_i)}$, and the effective carrying capacity can be calculated through $\frac{1}{K_i^*} = (1 - \beta^* g_i k_i)$. Here, g_i quantifies how pH influences species growth, while k_j quantifies how species alter the environmental pH. The parameter δ represents the recovery rate of pH due to the addition of fresh medium at the end of

each serial dilution cycle (with pH being neutral, or 7, after each addition). In the model, $p=0$ corresponds to neutral pH = 7 in the experiment, $p>0$ and $p<0$ represents alkaline and acidic pH respectively. The parameter β represents the environmental coupling strength, which quantifies the impact of species on environmental pH and reflects the phosphate buffering concentration in the experiment. The normalized environmental coupling strength, considering the competing effects of pH recovery (δ) and species-induced pH changes (determined by β), is given by $\beta^* = \beta/\delta$.

We found that the mean and standard deviation of the effective interaction strength under pH influence, α_{ij}^* , are larger than the original gLV interaction strength, α_{ij} (see figure below). Even when gLV interactions are set to zero, pH-mediated interactions alone produce a non-zero distribution of α_{ij}^* (see figure below). These results demonstrate that pH-mediated interactions can be effectively incorporated into the gLV framework. The compatibility of the gLV model with pH-mediated interactions justifies our choice, as it allows us to account for environmental factors like pH without sacrificing the simplicity and analytical power of the gLV model.

To further explore the pH model's predictions on invasion outcomes, we simulated the model combining both pH and gLV interaction effects across various parameter spaces. We found that even when considering only the pH effect, the model predicts a decrease in invasion probability with increasing pH interaction strength (mean of $\beta^* g_i k_i$) and species pool size (Supplementary Fig. 23), which qualitatively aligns with the predictions of the pure gLV model (Fig. 2f). In this purely pH-mediated interaction regime, the survival fraction is approximately equal to the invasion probability (Supplementary Fig. 23).

Moreover, when both gLV interaction strength (mean of α_{ij}) and pH-mediated interaction strength (mean of $\beta^* g_i k_i$) are increased simultaneously, the invasion probability decreases (Supplementary Fig. 23) in a manner similar to that predicted by the gLV model alone (Fig. 2f). In all conditions, the survival fraction remains an efficient predictor of invasion probability (Supplementary Fig. 23), as observed in the pure gLV model (Fig. 4).

These results demonstrate that the dependence of invasion probability on interaction strength, species pool size, and survival fraction is robust across both pure pH and combined pH-gLV models. This further justifies our choice of the gLV model, as it successfully predicts key features of invasion behavior in communities. By incorporating pH-mediated interactions into the gLV framework, we capture the added complexity of pH effects while retaining the simplicity and analytical power of the gLV model.

Figure. The presence of pH-mediated interactions increases the mean and standard deviation of effective gLV-type interactions. **a**, The gLV-type interaction α_{ij} follows a uniform distribution $U[0, 1]$, with a mean and standard deviation of 0.5 and 0.29, respectively. **b**, Under the influence of pH, the effective interaction $\alpha_{ij}^* = \frac{(\alpha_{ij} - \beta^* g_i k_j)}{(1 - \beta^* g_i k_i)}$ shows a larger mean interaction strength of 0.52 and a standard deviation of 0.37 compared to the pure gLV model. In this scenario, α_{ij} still follows $U[0, 1]$, while the pH modification parameters g_i and k_i are sampled from a uniform distribution $U[-1, 1]$. The environmental coupling strength β^* is set to 0.5. **c**, The pure pH model without gLV-type interactions produces a distribution of effective α_{ij}^* with a standard deviation of 0.18.

Supplementary Fig. 23. The invasion probability decreases with increasing pH-mediated interaction strength and gLV-type interaction strength, while the survival fraction remains approximately equal to the invasion probability. a, The pure pH model, without gLV-type interactions, predicts that invasion probability decreases with increasing pH interaction strength (mean of $\beta^* g_i k_i$) and species pool size. **b,** In the pure pH model, the survival fraction is approximately equal to the invasion probability, even without gLV-type interactions. **c,** When both pH interaction strength (mean of $\beta^* g_i k_i$) and gLV-type interaction strength (mean of α_{ij}) increase simultaneously, the invasion probability decreases. **d,** The survival fraction remains approximately equal to the invasion probability when combining the pH model with the gLV model. The points and color maps depict the mean value over 100 simulations.

The discussion on the pH model is referenced in the main text on page 5 and 6:

We also developed a model that directly incorporates pH-mediated growth within the Lotka-Volterra framework, allowing interactions to be expressed as a function of pH modifications. This new model suggests that the presence of pH effects increases the effective inter-species interaction

strengths and yields predictions similar to those of the canonical Lotka-Volterra model (Supplementary Fig. 23).

As the reviewer suggested, we now add justification for the choice of Lotka-Volterra model in the Supplement:

Justification for the choice of the Lotka-Volterra model

As a phenomenological model, generalized Lotka-Volterra (gLV) model describes species interactions through a pair-wise interaction matrix, capturing how the abundance of one species influences the growth of others. The interaction matrix in the gLV framework is a high-level description that can incorporate various mechanisms, including resource competition, cross-protection, cross-toxin, cross-feeding, and pH-mediated interactions.

It is important to emphasize that the pH-mediated interaction is not the only sole mechanism in our experiment. Some communities show fluctuating biomass while maintaining stable pH levels, indicating other mechanisms, such as cross-toxin interactions, are at play, particularly under high nutrient concentrations. Furthermore, our model considering only pH-mediated interactions cannot explain all experimental observations. For instance, our experiments and gLV simulations show fluctuating communities with higher diversity and invasibility (Hu et al. 2022), while our pH-only models predict lower diversity and invasibility (Supplementary Fig. 10-11 of Hu et al. 2022) (Hu et al. 2022), contrary to our experimental findings. These justifications support our use of the gLV model and align with recent progress in using phenomenological models for demographic manipulations (Shoemaker, Sánchez, and Grilli 2023).

To further justify our modelling approach, where increasing nutrient levels in the experiment correspond to corresponds to amplifying average interaction strength in the gLV model, we conducted pairwise coculture experiments. These experiments revealed that as nutrient concentration increased, the fraction of species pairs that coexisted decreased, while the fraction of competitive exclusions (where only one species survives) increased (Hu et al. 2022), reflecting the increase in interspecies interaction strength within the framework of the gLV model.

4. My understanding is that the term “fluctuations” in the context of this study means fluctuations induced by chaotic dynamics. I understand that these are not meant to be viewed as stochastic fluctuations, but in my experience, I have found that only a fraction of eco/evo readers understand the distinction between fluctuations induced by deterministic vs. stochastic effects. It would aid the typical reader if the authors clarified what form of fluctuations they are focused on in the Introduction, specified the type of fluctuation whenever the term is used in the manuscript (e.g., “fluctuations” and “deterministic fluctuations”), and briefly addressed in the Discussion how the consideration of stochastic fluctuations might factor into their subsequent research efforts.

We appreciate the reviewer for raising this important point. In this study, all fluctuations are referred to as deterministic fluctuations (chaos or limit cycle oscillations) driven by inter-species interactions, rather than stochastic fluctuations driven by demographic noise.

The reason we believe our experiment is in the deterministic population dynamics regime, rather than the stochastic regime, is due to the large population size in our experimental communities. Colony plating and counting in our experiment indicate that the CFU is on the order of 10^9 per mL. We cultured each community in a 300 μ L medium, consisting of around 3×10^8 cells, and transferred 10 μ L of the community cultures into a new plate with 300 μ L of fresh media in each

dilution cycle, transferring about 10^7 cells. The relative ratio of deterministic population growth to stochastic birth and death is on the order of $1/\sqrt{n}$, which is very low given the total population size of 10^7 to 3×10^8 cells in our experiment.

Additionally, our communities display stable species composition in stable communities, with sequencing results showing that species composition does not exhibit significant stochastic fluctuations (Hu et al. 2022). This suggests that stochastic fluctuations play a less important role in our experiment. We are indeed very interested in how demographic noise drives stochastic fluctuations in communities, and working on a subsequent research project on this topic.

We have followed the reviewer's suggestion and clarified the distinction between deterministic and stochastic fluctuations in the manuscript. We have specified the type of fluctuation whenever the term is used in the manuscript (e.g., “deterministic fluctuations”). We have also added a brief discussion on how consideration of stochastic fluctuations might factor into future research.

We revised the Introduction on page 3 of the main text:

These deterministic fluctuations in communities are chaotic dynamics or limit cycle oscillations driven by inter-species interactions, rather than stochastic fluctuations driven by demographic noise, because of the large population size regime in this study (see Supplement).

We addressed how the consideration of stochastic fluctuations might factor into their subsequent research efforts in the Discussion, on page 10 of the main text:

Beyond the deterministic fluctuations observed under large population sizes in this work, it is important to study invasions under stochastic dynamics driven by demographic noise in subsequent research. Theory shows that demographic noise can drive stochastic transitions between alternative stable states, leading to another type of community fluctuations (Kessler and Shnerb 2015).

We also added relevant discussion in supplementary materials to make this point clear:

In this study, all fluctuations are referred to as deterministic fluctuations (chaos or limit cycle oscillations) driven by inter-species interactions, rather than stochastic fluctuations driven by demographic noise. The reason we believe our experiment is in the deterministic population dynamics regime, rather than the stochastic regime, is due to the large population size in our experimental communities. Colony plating and counting in our experiment indicate that the CFU is on the order of 10^9 per mL. We cultured each community in a 300 μ L medium, consisting of around 3×10^8 cells, and transferred 10 μ L of the community cultures into a new plate with 300 μ L of fresh media in each dilution cycle, transferring about 10^7 cells. The relative ratio of deterministic population growth to stochastic birth and death is on the order of $1/\sqrt{n}$, which is very low given the total population size of 10^7 to 3×10^8 cells in our experiment. Additionally, our communities display stable species composition in stable communities, with sequencing results showing that species composition does not exhibit significant stochastic fluctuations (Hu et al. 2022). This suggests that stochastic fluctuations play a less important role in our experiment.

5. It is worth briefly mentioning in the supplement why continued migration from the regional pool was a necessarily experimental detail for this study. Specifically, how the absence of experimentally-imposed migration would correspond to a gLV with zero migration, the analytic

results of which would be decidedly less rich (e.g., no chaos, etc.). Broad readership may not pick up on this detail.

We have followed the reviewer's suggestion and discussed the importance of dispersal in the supplement to emphasize how dispersal from the regional pool contributes to maintaining community diversity and enabling persistent fluctuations:

We introduced a dispersal rate of 10^{-5} in both of our experiments and gLV simulations. The dispersal from the species pool to the local community is important for maintaining persistent fluctuations in both gLV simulation and our experiment. We found that the lack of dispersal in the gLV model leads to a significantly lower fraction of fluctuating communities (Supplementary Fig. 24). Similarly, the fraction of fluctuating communities is much lower in our experimental communities without daily dispersal (Supplementary Fig. 25). Without dispersal from the species pool, some species reach the extinction boundary due to dramatic fluctuations and cannot recover. This results in a continuous decrease in community fluctuations in both the model and the experiment.

Supplementary Fig. 24. Non-zero dispersal sustains persistent community fluctuations in gLV model. The panels show the theoretical phase diagrams of community fluctuation fraction under different dispersal rates ($D=0$, $D=10^{-7}$, $D=10^{-6}$). Communities under no dispersal ($D=0$, left panels) exhibit a low fluctuation fraction in the persistent fluctuation phase. The patterns of ecological diversity and dynamics do not significantly change as the dispersal rate varies from $D=10^{-7}$ (middle panels) to $D=10^{-6}$ (right panels). The dashed line and solid line in the figures represent survival boundary and stability boundary, respectively. The color maps depict the mean value over 1000 simulations.

Supplementary Fig. 25. Zero dispersal leads to stable communities under high nutrient conditions (strong interactions) in the experiment. The biomass of eight distinct microbial communities ($S=24$) consistently reaches stable states when exposed to high nutrient concentrations (strong interactions) in the absence of species dispersal from the species pool to the community. In contrast, when dispersal is present, a significant proportion of communities exhibit fluctuations under the same high nutrient conditions and species pool size, as previously reported in Fig. 1d and in our previous paper (Hu et al. 2022) (Fig. 2C).

6. Related, did the authors perform any experiments without daily migration from the regional pool? Contrasting the results of this study with that type of experiment would strengthen their claims and provide further opportunities to test the utility of the gLV. Alternatively, if there are there any published experiments on invasion analysis using a similar experimental setting without imposing migration, then such results could be verbally compared to the results of their experiment.

We conducted experiments without daily migration from the regional pool and observed that the fraction of fluctuating communities was significantly lower compared to experiments with daily dispersal. In the absence of dispersal, some species reached the extinction threshold due to dramatic fluctuations and were unable to recover, resulting in a continuous decrease in community diversity and the eventual disappearance of persistent fluctuations. These findings are consistent with the predictions of the gLV model without migration, which also shows a lower fraction of fluctuating communities. While we have not yet performed invasion experiments without dispersal, we agree with the reviewer that this would be an important and interesting experiment to pursue in our ongoing research. We have incorporated this comparison and discussion into the manuscript to emphasize the importance of dispersal in maintaining community diversity and enabling persistent fluctuations.

7. Is the experimentally-imposed migration rate sufficient to induce chaotic dynamics in the context of their gLV model? I searched for this detail in the supplement but did not find it.

The dispersal rate in our experiments and gLV simulations is 10^{-5} . We found that different levels of dispersal rates can sustain persistent fluctuations (chaos or limit cycle oscillations) in the gLV model. As long as the extinction-prone species can come back and bloom through dispersal, the persistent fluctuation is not influenced by specific values of the dispersal rate. Only a zero-dispersal rate leads to a significantly lower fraction of fluctuating communities in the model. Without dispersal from the species pool, some species hit the extinction boundary due to dramatic fluctuations and cannot recover, resulting in a continuous decrease in community fluctuations. We

have discussed the consistency between dispersal rates in the experimental and simulation settings in the supplement to clarify this point.

8. I would also encourage the authors to make both their code and raw data available on public repositories to ensure transparency and reproducibility.

We thank the reviewer for raising this important point. We planned to make the code and raw data available in public repositories before publication. We have now uploaded our raw data and code to public repositories to ensure transparency and reproducibility:

Data and materials availability: Isolates and communities are available upon request. All data are available in the supplementary materials and deposited on Dryad (https://datadryad.org/stash/share/Mi09U1xtHkQ8_0D4DPuVDzoSp0-hcIT8oG76h8edCdM). All codes used for simulation and analysis in this publication are available on GitHub (<https://github.com/Jiliang-Hu/Collective-dynamical-regimes-predict-invasion>).

9. Lines 148-149: It would help the reader if justification for the extinction threshold was provided in the main text. In lines 81-82 of the supplement the authors describe the lower bound on detection due to sequencing, but it is not clear how they arrived at this particular value. It would also help if there was a comparison between the chosen threshold and the inverse of the typical total abundance of the community (all cells), the threshold for true extinction, to provide context for the detection limitation.

We appreciate the reviewer for this important comment, and have added the justification of the choice of extinction threshold in the supplement and refer it in the main text to provide context for the detection limitation and comparison with the typical total abundance of the community:

We chose the survival threshold for three main reasons:

1. **Distinction from dispersal rate:** The survival threshold should be significantly higher than the dispersal rate to distinguish truly surviving species from those whose low abundance is only sustained by dispersal. Since we used a dispersal rate of 10^{-5} , the threshold should be high enough to be separated from the dispersal floor but not too high to falsely classify surviving species as extinct. Simulation results of gLV suggest that a threshold of 8×10^{-4} efficiently separates surviving species from extinct ones (Supplementary Fig. 26).
2. **Sequencing depth:** The sequencing depth is on the order of 10^4 . The number of cells transferred in each dilution cycle is on the order of 10^7 cells, and the number of cells used in DNA extraction and amplicon sequencing is on the order of 10^6 cells. Therefore, the detection limit in our sequencing data is primarily determined by the sequencing depth of each community. Any threshold below the order of 10^{-4} would be inconsistent with the detection limit of our sequencing depth. We chose 8×10^{-4} , which is sufficiently above the detection limit but not too high to exclude low-abundance surviving species.
3. **Robustness of conclusion:** We found that varying the survival threshold between 10^{-4} and 10^{-3} does not alter our key conclusions. Specifically, a survival threshold of 10^{-4} yielded exactly the same number of successful invasions as a threshold of 8×10^{-4} in our experiment. The survival threshold of 10^{-3} resulted in 61 successful invasions out of 244 total invasion tests, compared to 63 successful invasions out of 244 total invasion tests

under the 8×10^{-4} threshold. There were only two cases where the invader abundance fell between 8×10^{-4} and 10^{-3} , leading to a minor quantitative difference in the number of successful invasions. We further verified that this small difference does not affect our major conclusions, including (1) that fluctuating communities are more invasible than stable communities, and (2) that invasion probability decreases with increasing species pool and interaction strength.

Supplementary Fig. 26. The survival threshold efficiently separates surviving species from extinct species, where the species abundances display a bimodal distribution in the simulations. a, The extinction threshold of 8×10^{-4} (horizontal dashed line) clearly separates the high-abundant, surviving species from the low-abundant “extinct” species ($S=50$, $\langle \alpha_{ij} \rangle = 0.2$). Such “extinct” species would reach zero abundance if dispersal is interrupted (c). **b**, The extinction threshold of 8×10^{-4} (horizontal dashed line) similarly separates the high-abundant, surviving species from the low-abundant “extinct” species under different interaction strength ($S=50$, $\langle \alpha_{ij} \rangle = 0.6$). **c**, After stopping dispersal at $t=1000$, only species above the extinction threshold survive with stable abundances, while the others undergo extinction ($S=50$, $\langle \alpha_{ij} \rangle = 0.6$). This demonstrates that the extinction threshold of 8×10^{-4} efficiently classifies surviving species versus those that would go extinct without dispersal. **d**, The histogram shows the number of species exhibiting the indicated abundances at steady state. The dataset was generated from 10 *in silico* communities randomly sampled ($S=50$, $\langle \alpha_{ij} \rangle = 0.2$).

We have referred to the justification of the choice of extinction threshold in in the main text on page 4:

relative invader abundance exceed extinction threshold 8×10^{-4} on the last day 12; the rationale behind this choice of extinction threshold is explained in the Supplement.

10. Lines 208-211: I am unaware of models where the competition coefficients between species can be examined as a function of media modification. If such a model exists, could the authors briefly describe it in their supplement to help the reader understand how environmental variables (e.g., pH) map onto competition coefficients in a gLV model?

We have developed a model that combines the pH effect with gLV-type interactions to explain our experimental observations, consistent with gLV predictions. This pH model considers species' modifications of environmental pH and the pH-dependent growth of species. In our model, the interaction matrix in the gLV framework is influenced by pH changes induced by species interactions. This combined pH-gLV model can be reduced to a pure gLV-type interaction matrix in the adiabatic limit, allowing us to explicitly show how the pH effect influences the phenomenological interaction matrix in the gLV model. We have added a detailed description of this pH model in the supplement to help readers understand how environmental variables, such as pH, map onto competition coefficients in a gLV model.

In our group, we have been investigating how community diversity and dynamics depend on resource concentration across various consumer-resource models (e.g., linear growth and Monod growth). However, unlike our experimental observations, we did not find that increasing resource concentration leads to a decrease in diversity and stability within these models. We also tested a consumer-resource model with pH-dependent growth terms, but similarly, we did not observe a decrease in diversity and stability with increased resource concentration in any of the models. We agree with the reviewer that explicitly relating resource concentration to changes in pH and the resulting increase in the strength of competitive interactions is both interesting and important; however, this remains an open question in our ongoing research.

11. Lines 275-280: How are alternative stable states determined here?

We thank the reviewer for this important question. As proved in previous theoretical work (Fried, Shnerb, and Kessler 2017; Bunin 2017) and shown in our experiments (new supplementary Fig. 22), we found that the same set of species starting from different initial species compositions can reach different alternative stable states under high nutrient concentration (strong interaction). In contrast, they reach a single global stable state under low nutrient concentration (weak interaction), with all communities receiving daily dispersal of species from the pool.

The observed alternative stable states under high nutrient concentration (strong interaction) exhibit a priority effect, which helps explain why invasion probability is generally lower than the survival fraction under these conditions (bottom left points in Fig. 3c). We have added a detailed description of the determination of alternative stable states in the supplement to clarify this point.

Supplementary Fig. 22. Communities reach global stable states under low nutrient conditions (weak interaction) but reach alternative stable states under high nutrient conditions (strong interaction) when starting from different initial species compositions. Each bar represents a final community stable state after 7 daily dilution cycles starting from a particular initial condition. The communities assemble from the same species in the original pool ($S=12$ for the left and middle communities, $S=24$ for the right communities), under low-nutrient (weak interaction) and high-nutrient (strong interaction) conditions, respectively. For each initial condition, one of the species occupies 99% volume of the initial inoculum, while the other species in the pool together occupy only 1% volume of the initial inoculum. The results show that these very different initial species compositions lead to the same global stable state under low nutrient conditions (weak interaction) but result in alternative stable states under high nutrient conditions (strong interaction). This indicates the presence of alternative stable states and priority effect under high nutrient (strong interaction), where the initial composition influences the final community structure.

12. Fig. 1d: Why are all but two of the “fluctuating” communities decreasing in biomass from day four to five. Could this simultaneous decrease in biomass be driven by experimental details rather than by the intrinsic dynamics of the communities?

We thank the reviewer for raising this important point. We cultured both stable and fluctuating communities in a mixed manner on the same 96-well deep plates and subjected them to the same experimental processes, including daily dilution and dispersal. The total biomass and sequencing results of stable communities do not show any different behaviors from day 4 to day 5, suggesting that the behaviors of fluctuating communities were not driven by experimental details. Additionally, control wells containing only fresh medium on the same deep-well plates showed no contamination.

While six of the fluctuating communities decrease in biomass from day four to day five and two of them increase, this pattern is likely due to the small sample size, as we only have eight time series, each spanning just six days. Although this synchrony is not the focus of our current work, it could potentially be explained by the alternating growth cycles of fermentation species and respiration species in our species pool. According to previous studies (Estrela et al. 2022), when fermentation species dominate and grow rapidly, they produce significant amounts of organic acid

by consuming glucose, leading to highly acidic conditions. This acidity inhibits the growth of respiration species and can even result in the self-destruction of the fermentation species (Ratzke, Denk, and Gore 2018). During these cycles, nutrients are not efficiently utilized, resulting in low biomass and pH. In the subsequent cycle, respiration species may become dominant as the acidic conditions from the previous cycle have reduced the fermentation species population. This shift can lead to a higher pH environment where fermentation species do not self-destruct, allowing them to dominate in the next cycle. This alternating dominance between fermentation and respiration species could potentially explain why six of our communities fluctuate in a seemingly synchronized manner. We are currently working on a project to study the mechanism of these intrinsic fermentation and respiration cycles in our communities but do not yet have sufficient data to draw definitive conclusions.

13. Fig. 1g; lines 380-382: What statistical test was performed here?

We performed a two independent samples Student's t-test to compare the groups. We have now clarified this point in the figure caption and manuscript.

14. Fig. 2f: What do the dashed and solid grey lines represent? Could the authors include a legend in the figure?

We have added legends to Fig. 2f to explain the solid and dashed lines. The solid grey lines represent the stability boundary, and the dashed grey lines represent the surviving boundary.

15. Fig. 4c: Could the authors add an axis label for the colorbar in the revision?

We have added the axis label “priority effect” for the colorbar in Fig. 4c, as suggested.

16. Fig. 4d: “Effect on resident community” is a nice term for those unfamiliar with these types of experiments, but it still reads as somewhat vague. Could the authors describe what this term means in the figure legend and provide an equation for how the quantity was estimated in the supplement?

We thank the reviewer for this suggestion. The invasion effect on the resident community is measured as the proportion of change in surviving species before and after the invasion, calculated as $1 - (\text{number of overlapping species} / \text{total number of species})$. We have added the mathematical definition of “Effect on resident community” in Fig. 4d and its figure caption. We have also included the equations for how this quantity was calculated in the manuscript and the supplement.

17. Fig. 5: What timepoints are used to calculate the fold change? Are the same timepoints used for each community? Is the outcome of the statistical test robust to the choice of timepoints?

We thank the reviewer for this important question. We have clarified in the manuscript that the fold change of biomass was calculated using the biomass on day 6 (before the invasion) and day 12 (after the invasion colonization was well completed) across all communities.

As the reviewer suggested, we further examined the robustness of the statistical test to the choice of timepoints. We found that the same statistical difference was observed when calculating the fold change using biomass on day 6 (before the invasion) and the average biomass of day 10, day 11, day 12 (after the invader colonization). This indicates that the outcome of the statistical test is robust to the choice of timepoints. We have included this additional analysis in the manuscript to ensure clarity and robustness.

Supplementary Fig. 27. The mean fold change in biomass under high nutrient conditions is greater than under low nutrient conditions, which is robust to the choice of time points. a, Invasions into resident communities under low nutrient conditions (weak interactions) result in a statistically lower fold change in biomass compared to communities under high nutrient conditions (strong interactions) ($p < 0.001$). The number of successful invasions is $n=51$ (low nutrient) and $n=11$ (high nutrient). The fold change in biomass was calculated by comparing the biomass on day 6 (before invasion) with that on day 12 (after invasion colonization was fully established) across all communities. **b,** Similarly, invasions into resident communities under low nutrient conditions (weak interactions) cause a statistically lower fold change in biomass than those under high nutrient conditions (strong interactions) ($p < 0.001$). The fold change was calculated by comparing the biomass on day 6 (before invasion) with the average biomass of days 10, 11, and 12 (after invader colonization) across all communities.

18. Supplement line 52: The sampling detection limit is $\sim 10^{-4}$ but the chosen dilution rate was 10^{-5} . How does the difference between dilution rates and detection limit shape the observed outcomes of the experiment? If it does, it may be worth describing the outcomes in different parameter limits to the reader (i.e., sampling limit much greater than dilution rate and vice-versa).

We thank the reviewer for this important comment. As shown in the simulation results with gLV, the abundance of surviving species and successful invaders are all well above the surviving threshold (8×10^{-4}) and the sampling detection limit (10^{-4}). Therefore, the measured diversity and invasion success in this work are not influenced by the existence of the sampling detection limit (10^{-4}). In other words, a better sampling detection limit (e.g., 10^{-5} or even higher depth) would not change the measured diversity and invasion probability in our work, and thus would not change our observations and conclusions.

With a detection limit of 10^{-4} , we indeed cannot detect species whose low abundance is only sustained by the dispersal rate (10^{-5}). However, this does not impact our study because we focus on the invasion of surviving species whose abundances are high. We have added a description of these outcomes in different parameter limits to the supplement to provide further clarity.

19. Supplement: Lines 52, 63: I am unfamiliar with the term “dispersal rate” used to describe the process of diluting cultures in a serial transfer experiment. The method used by the authors is different from the standard process of serial dilution, so I think some justification of the term as

well as an explanation of how it connects to the mathematical definition of dispersal (Eq. S1) in the supplement is warranted.

Daily dilution and daily dispersal are two different steps in our experiment. We performed a 30-fold daily dilution by transferring 10 μL of the community cultures into a new plate with 300 μL of fresh media. To apply a 10^{-5} daily dispersal, we diluted the monoculture of each species in the community by a 10^5 factor before inoculating 10 μL of diluted monoculture into the wells containing the corresponding experimental community matching each species pool. Therefore, the volume ratio between the dispersal from each species monoculture in the pool and the transfer of each community is 10^{-5} .

We chose the term “dispersal rate” according to the ecology literature that studies the migration of species from the mainland or an outside habitat to a local community (Handel 2014). This low dispersal rate is sufficient to avoid the complete extinction of species due to occasional low abundance in persistent fluctuations. In our gLV simulation and experiment, some species hit the extinction boundary due to persistent fluctuation and cannot recover without dispersal, leading to the eventual disappearance of fluctuations due to the decrease in diversity over time. Therefore, the small dispersal rate is important to sustain persistent fluctuations in our community.

The dispersal rate for each species in our model is also chosen to be 10^{-5} to match the experiment, and we found that the results do not vary significantly with specific values of the dispersal rate in the gLV model.

We have followed the reviewer’s suggestion and added more comments on the dispersal rate in the supplementary materials to clarify its connection to the mathematical definition of dispersal (Eq. S1).

20. Supplement lines 100-119: Does the absence of serial dilution being explicitly encoded into the simulation impact the results? Serial dilutions have been explicitly incorporated into mechanistic (Marsland et al., 2020) and phenomenological (Shoemaker et al., 2023) models of community dynamics in an experimental context. Could the authors provide justification of why serial dilution does not need to be explicitly encoded into their simulation?

We thank the reviewer for this important question. In our previous work (Hu et al. 2022) and other studies in our group, we found that continuous simulation of gLV yields similar qualitative outcomes as simulations incorporating serial dilutions. Specifically, we observed similar dynamical phases in simulations with or without serial dilutions.

We agree with the reviewer that it is important to show how our results change if we consider a 30-fold daily dilution in our model. We have now added these results to the supplementary materials, demonstrating that the consideration of serial dilution does not change the pattern of invasion outcomes in our simulations.

Supplementary Fig. 28. Under serial dilutions in the gLV model, the invasion probability decreases with increasing interaction strength and species pool size, the survival fraction remains approximately equal to the invasion probability. a, Invasion probability decreases with increasing interaction strength and species pool size under serial dilutions. b, The survival fraction is approximately equal to the invasion probability under serial dilutions in the gLV model. The points and color maps depict the mean value over 100 simulations.

21. Supplement lines 109-111: What are the units of time and how do they compare to the timescales of the experiments? Is the timescale of the simulation informed by the timescale of the experiment?

We thank the reviewer for this important question. The unit of time in the simulation is $1/r$, where r is the growth rate of species in the gLV model, which is set to be unit 1 in our simulations. We are interested in the steady-state behaviors of invasion rather than the effects of transient dynamics. Our simulation results show that $t=10^3$ is long enough for the communities to complete their transient dynamics from initial conditions to their final fluctuating or stable state. In the steady-state regime, the results do not vary with the time window. As shown in our previous work, the fraction of fluctuations and survival fraction do not vary with time after reaching the steady state.

We further demonstrate that invasion probability results remain consistent when introducing the invader at $t=2 \times 10^3$ compared to $t=10^3$. Our experimental communities typically reach steady states by day 6, according to species composition and community biomass time series (Supplementary Fig. 3-11). We found that stable communities' species compositions usually do not significantly vary after day 4, while fluctuating communities' species compositions continue to fluctuate on days 6 and 10 (Hu et al. 2022). Therefore, both our simulation and experiment characterize invasion behavior under community steady states.

According to the growth curves of the isolates in our experiments (Supplementary Fig. 19), the characteristic growth rate of the isolates is around 1 ($1/h$). Therefore, each daily dilution cycle approximately corresponds to 24 unit simulation time when the growth rate in the model is set to unit 1. This means that the one cycle timescale is on the order of 100 unit times based on this estimation. We have added a discussion of this point in the supplementary materials to clarify the relationship between the timescales of the simulation and the experiments.

Supplementary Fig. 29. The invasion probability decreases with increasing interaction strength and species pool size across different time windows, with the survival fraction remaining approximately equal to the invasion probability. a, The invasion probability decreases as interaction strength and species pool size increase. b, The survival fraction closely mirrors the invasion probability. To assess whether invader or resident species survived, we identified species whose abundance exceeded the extinction threshold at any point during the last 24 time units of the simulation. This approach yielded invasion probability patterns consistent with those observed in a 100-unit time window. The points and color maps depict the mean value over 100 simulations.

22. Supplement lines 147-148: It's unclear what the CV represents. My understanding is that each CV is calculated over time for each species for a given simulation iteration. Since the dynamics are deterministic and the interaction coefficients are a form of quenched disorder, what random variable is the CV calculated over? Does the distribution of CVs demonstrate a clear bimodality with the 10^{-3} cutoff representing the valley between the two peaks? It would help the reader if 1) additional context was provided for the CV and what it represents as well as 2) distributions of simulated CVs plotted to demonstrate the intuition behind classifying communities into stable/fluctuating classes.

We thank the reviewer for this important comment. The CV represents the coefficient of variation of species abundance. We first calculate the CV of each species' abundance N_i over the time window between $t=10^3-100$ and $t=10^3$, then pick the maximal one among the species abundance CVs across all species in the community. We then identify the maximum CV among all species in the community. A community is considered fluctuating if this maximum CV exceeds a threshold, as stable communities are characterized by all species abundances that have reached stable states.

As the reviewer correctly noted, steady-state communities exhibit small maximum CV values for species abundances, while fluctuating communities show relatively large maximum CV values, resulting in a bimodal distribution. The threshold of 10^{-3} serves as an efficient valley threshold to separate the two peaks. We have added the figure and a detailed discussion in the supplementary materials to clarify this point.

Supplementary Fig. 31. The threshold of 10^{-3} for the maximal CV of species abundance effectively separates fluctuating communities from stable ones, where the maximal CV of species abundance exhibits a bimodal distribution in the simulations. The histogram displays the number of communities with the indicated maximal CV of species abundance at steady state. The dataset was generated from 2000 in silico communities, randomly sampled with $\langle \alpha_{ij} \rangle \in [0.02, 1.1]$ and $S \in [2, 60]$.

23. Supplement lines 160-163, Fig. S12c, d: Is some degree of correlation expected since biomass was used to calculate the abundance of each species? This seems similar to the case presented in Garud et al., where the same parameter factors into both sides of the relationship (pg. 16 and 17 of supplement; 2019). Here the authors partitioned synonymous sites into two categories, providing two estimates of the quantities of interest. These two quantities are conditionally independent of the parameter that factors into both sides of the proposed relationship due to the Poisson thinning property. One quantity is then used to calculate the left side of the relationship while the other is used to calculate the right side. Is a similar analysis appropriate in this case given that total biomass factors into both sides of the quantities examined in Fig. S12c, d?

We appreciate the reviewer's insightful question. We found that the CV of community biomass is an effective indicator for classifying fluctuating and stable communities in our experiment (as shown in panels a and b of the figure below). Additionally, we demonstrate that both the normalized variation of absolute species abundance (the product of total biomass and species relative abundance by sequencing) and the normalized variation of relative species abundance (relative species composition by sequencing) show a significant positive correlation with biomass CV (correlation=0.91, $p=4.67 \times 10^{-10}$; correlation=0.76, $p=1.06 \times 10^{-5}$). As the reviewer noted, the normalized variation of absolute species abundance is indeed coupled with biomass CV because biomass is used in calculating absolute species abundance. However, the normalized variation of relative species abundance, derived from sequencing data, is independent of biomass.

To further validate our classification approach, we applied the K -means clustering classification algorithm to biomass CV alone, normalized variation of absolute species abundance versus biomass CV (panel c), and normalized variation of relative species abundance versus biomass CV (panel d). All analyses yielded the same classification outcome for fluctuating and stable communities. We have now added these discussions to the supplementary materials to provide a clearer understanding of our methodology and results.

Supplementary Fig. 12. Classification of fluctuating and stable resident communities in experiment. **a**, The standard deviation of community biomass over day 4, day 5 and day 6 show that the stability threshold of 0.05 can separate the communities into stable ones (purple points) with small biomass deviation and fluctuating ones (orange points) with relatively large biomass deviation under high nutrient. **b**, The standard deviation of community biomass under low nutrient are small (all below the stability threshold of 0.05), which were naturally classified into stable communities. **c**, Similarly, the standard deviation of community biomass over days 3, 4, 5, and 6 under high nutrient conditions confirms that the stability threshold of 0.05 can distinguish between

stable communities (purple points) with low biomass deviation and fluctuating communities (orange points) with higher biomass deviation. **d**, Under low nutrient conditions, the standard deviation of community biomass over days 3, 4, 5, and 6 remains below the stability threshold of 0.05, consistently classifying the communities as stable. **e**, The average coefficient of (temporal) variation for absolute species abundances (N_i , computed as the product of total biomass and species relative abundance) exhibit a strong positive correlation with standard deviation of biomass in the experimental communities (correlation=0.91, $p=4.67\times 10^{-10}$). *K*-means clustering method classifies the points into two clusters where fluctuating communities locate on top right region and stable communities locate on bottom left region. **f**, The average coefficient of (temporal) variation for relative species abundances (N_i^* , relative species abundance through 16s sequencing) also exhibits a strong positive correlation with standard deviation of biomass in the experimental communities (correlation=0.76, $p=1.06\times 10^{-5}$). *K*-means clustering method classifies the communities into stable ones (purple and gray points) and fluctuating ones (orange points). The results suggest that fluctuation in community biomass cooccurs with fluctuation in relative species abundances.

24. An alternative option could be to examine the relationship between the CV of biomass (OD) and the CV of relative abundances for each ASV, allowing for the identification of community members that disproportionately contribute to the fluctuations in biomass.

We thank the reviewer for this suggestion. The reason we do not calculate the CV of relative abundance for each ASV is that the read number of low abundance ASVs is small and significantly influenced by limited sequencing depth and variation in amplification efficiency for different sequences. While highly abundant ASVs can reach stable states in some stable communities (as shown in the figure below), low abundance ASVs often display fluctuations due to large noise effects from sampling a small number of reads and other amplification and sequencing noise. Calculating the average CV across all ASVs, including those with small numbers of reads, would amplify the detection variation caused by these small numbers and noise, potentially skewing the classification of community dynamics as fluctuating or stable. To mitigate these issues, we focus on community-level metrics that are less influenced by the noise associated with low abundance ASVs. This approach ensures a more robust classification of community dynamics.

25. Fig. S2: Is there any phylogenetic structure to these results? Does the placement of a species on the phylogeny relative to the phylogenetic composition of a given community determine species invasibility?

We thank the reviewer for this insightful question. To study the effect of phylogenetic overlap between invaders and resident species on invasion outcomes, we calculated the overlap fraction of resident species that share the same phylogeny with the invaders across different phylogenetic levels, weighted by the abundance of the resident species. Our analysis revealed a statistically significant positive correlation between this overlap fraction and invasion success, as measured by the final abundance of the invader after colonization. Specifically, invaders that are phylogenetically closer to the resident species tend to achieve higher post-invasion abundances.

Our interpretation of this positive correlation aligns with our finding that the survival fraction is approximately equal to the invasion probability. A higher fraction of resident species sharing the same phylogeny as the invader indicates that this phylogenetic type is advantageous within the resident community, thereby increasing the likelihood of the invader's survival and success in this environment. We have included these findings on page 9 of the main text and in the supplementary

materials to provide a more comprehensive understanding of how phylogenetic structure influences invasion outcome:

Interestingly, invaders that are phylogenetically closer to resident species tend to achieve higher post-invasion abundances (Supplementary Fig. 32).

Supplementary Fig. 32. Invaders that are phylogenetically closer to resident species tend to achieve higher post-invasion abundances. The overlap fraction of resident species sharing the same phylogeny with invaders shows a statistically significant positive correlation with invader abundance after colonization, across different phylogenetic levels, including phylum, class, order, family, and genus. The overlap fraction at the kingdom level is always 1, as all resident species and invaders in the experiment belong to the same kingdom, Bacteria.

26. Fig. S12: “The standard deviation of community biomass over day 5, day 6 and day 7” this detail is confusing since the timeseries represented in Fig. 1d ends at day six. What was the rationale behind using only these three timepoints? Some justification of using these three timepoints is needed in the Supplement.

We thank the reviewer for this important comment. We apologize for the typo. The correct sentence should read “The standard deviation of community biomass over day 4, day 5, and day 6,”

consistent with the text in the supplementary materials: “We also calculated the average coefficient of variation (CV) for species abundances from day 4, day 5, to day 6.” We aimed to classify the steady-state dynamical behavior of communities and therefore tried to avoid transient dynamics influencing our classification of fluctuation and stability. The reason we chose days 4, 5, and 6 to calculate the CV for classifying fluctuating and stable communities is that some communities had not reached steady-state dynamics before day 3 (as shown in the Fig. 1d, Supplementary Fig. 4-11, showing transient dynamics occur before day 3). We found that most communities typically reach steady-state by day 4, so we used this as the starting point for calculating the CV of dynamics.

We further checked that the CV calculated using data from days 3, 4, 5, and 6 alone yields the same classification results, demonstrating that our classification is not sensitive to the choice of time windows. We have added these new results and figures to the supplementary materials.

Supplementary Fig. 12. Classification of fluctuating and stable resident communities in experiment is robust to choice of time window. **a**, The standard deviation of community biomass over days 4, 5, and 6 under high nutrient conditions shows that a stability threshold of 0.05 effectively separates the communities into stable (purple points) with low biomass deviation and fluctuating (orange points) with relatively high biomass deviation. **b**, Under low nutrient conditions, the standard deviation of community biomass over days 4, 5, and 6 is consistently below the stability threshold of 0.05, classifying all communities as stable. **c**, Similarly, the standard deviation of community biomass over days 3, 4, 5, and 6 under high nutrient conditions

confirms that the stability threshold of 0.05 can distinguish between stable communities (purple points) with low biomass deviation and fluctuating communities (orange points) with higher biomass deviation. **d**, Under low nutrient conditions, the standard deviation of community biomass over days 3, 4, 5, and 6 remains below the stability threshold of 0.05, consistently classifying the communities as stable.

27. Fig. S13: The figure is a useful visualization, but in its current form it is difficult to validate the claims made in the legend.

- “Invisibility positively correlates with richness when varying interaction strength” Is a claim being made here about the direction in which the correlation changes with a change in interaction strength? It’s not clear to me whether this claim is about the existence of positive correlations among treatments or the difference in positive correlations between treatments.

We thank the reviewer for this comment. When we fixed the species pool size at $S=20$, we found that the data points for $S=20$ under low nutrient and high nutrient conditions display a significant positive correlation between invisibility and richness. This finding is consistent with the gLV simulation results shown in Fig. 4a.

To clarify this point, we have plotted the data in a separate new figure and tested the statistical significance of the positive correlation (correlation coefficient = 0.7, $p = 1.06 \times 10^{-4}$). This approach makes it clearer that the claim is about the existence of positive correlations within treatments rather than differences between treatments.

- “Invisibility positively correlates with richness when randomly sample $S=20$ communities under high nutrient, due to fluctuating communities display larger richness and larger invasion probability.” I am having a hard time parsing this statement. Is a claim being made about correlation being higher in high nutrient $S=20$ fluctuating communities relative to high nutrient $S=20$ stable communities?

When analyzing the data points under high nutrient conditions with $S=20$, we found a significant positive correlation between invisibility and richness (correlation coefficient=0.5, $p=0.047$), as shown in the figure below. This positive correlation arises because fluctuating communities (orange points) are more diverse and more invasible than stable communities (purple points). As a result, fluctuating communities are located in the top right region while stable communities are located in the bottom left region of the figure below.

To clarify this point, we further examined the correlation within stable (purple points) and fluctuating (orange points) communities separately. We confirmed that neither stable nor fluctuating communities displayed a statistically significant correlation between diversity and invisibility ($p=0.16$ for stable and $p=0.15$ for fluctuating). Therefore, the overall positive correlation when combining the data points is due to the higher invisibility and diversity of fluctuating communities compared to stable ones. We have added a new figure and detailed discussion in the supplementary materials to elucidate this finding.

- The legend could use some retooling for clarity. Furthermore, if the claims are about the increase in correlation in one treatment vs. another, then additional statistical analyses are necessary. It seems like the authors are making three claims, which would require three tests. The question is what statistical model to use. A full regression analysis to examine the increase in slope between two treatments while controlling for potential cofounders could be seen as necessary. Alternatively,

the authors could test for the difference between two correlation coefficients using Fisher's Z statistic with a null distribution obtained by permuting community identity for a given pair of treatments (Snedecor & Cochran, 1989). This statistic was recently used to test for the change in correlation coefficients in experimental microbial communities (Eq. 21 in Shoemaker et al., 2023).

We thank the reviewer for this comment. To clearly clarify our statements, we have added three new figures and performed additional statistical tests to show the following:

1. **Invasibility positively correlates with richness when varying interaction strength:** We observe a positive correlation between invasibility and richness for $S=20$ communities under low and high nutrient conditions (correlation coefficient=0.7, $p=1.06 \times 10^{-4}$).
2. **Invasibility positively correlates with richness when randomly sampling $S=20$ communities under high nutrient:** This is due to fluctuating communities displaying larger richness and larger invasion probability (correlation coefficient=0.5, $p=0.047$).
3. **Invasibility negatively correlates with richness when increasing species pool size from $S=12$ to $S=20$ under low nutrients:** This correlation is statistically significant (correlation coefficient=-0.62, $p=0.014$).

Additionally, we show that under high nutrient conditions with $S=20$, neither stable communities (purple points) nor fluctuating communities (orange points) display any statistically significant correlation between diversity and invasibility ($p=0.16$ for stable and $p=0.15$ for fluctuating). We have revised the legend and added these new analyses and figures to the supplementary materials.

Supplementary Fig. 13. Different invasibility-richness relationships in experiment depending upon how the richness is changed (a). b, Invasibility positively correlates with richness when varying interaction strength under fixed species pool (positive correlation between $S=20$ communities under low and high nutrient, correlation is calculated across all data points in panel **b**, correlation coefficient=0.7, $p=1.06 \times 10^{-4}$). **c**, Invasibility positively correlates with richness when randomly sample $S=20$ communities under high nutrient, due to fluctuating communities display larger richness and larger invasion probability (correlation is calculated across all data points in panel **c**, correlation coefficient=0.5, $p=0.047$). Neither stable communities (purple points in panel **c**) nor fluctuating communities (orange points in panel **c**) display any statistically significant correlation between diversity and invasibility ($p=0.16$ for stable and $p=0.15$ for fluctuating). **d**, Invasibility negatively correlates with richness when increasing species pool size from $S=12$ to $S=20$ under low nutrient (correlation is calculated across all data points in panel **d**, correlation coefficient=-0.62, $p=0.014$).

28. Fig. S15, 16: Some type of statistical test is necessary to establish the claim that invasions lead to changes in community composition. There are multiple ways to accomplish this task and the

authors may have their own idea. One immediate option strikes me: calculate a paired t-statistic for each community between the control and successfully invaded community, demonstrating that absolute value of the t-statistics is significantly greater than zero. Null distributions could be obtained by permuting control/invade labels of each species within each community.

We thank the reviewer for the thoughtful suggestions regarding the statistical analysis of invasion effects on community composition. We followed the reviewer's recommendation to calculate the t-statistic for each community by comparing the control and successfully invaded communities, and to compare these values against a Null Distribution obtained by permuting the control/invade labels within each community. Upon conducting this analysis, we observed that the t-statistics for all communities did not show significant differences when compared to the Null Distributions. While this might seem unexpected, it is actually consistent with the underlying assumptions of our experimental and modeling framework.

The reviewer's suggestion appears to be based on the assumption that invasion would cause significant changes in community composition, implying non-random effects—where the invasion typically has consistent and predictable impacts on the community, whether by increasing diversity, decreasing it, or causing other changes. However, in our study, this assumption does not hold true. Both the invader and resident species were chosen randomly, reflecting the inherent randomness of the ecological network (modeled using a generalized Lotka-Volterra (gLV) model). Given this random selection, the interactions between species are also random. As a result, the impact of an invader on the resident community is not deterministic but rather stochastic—an invader may cause the appearance of new species or lead to the extinction of some existing species. This stochastic nature of species interactions means that the effect of invasion does not necessarily result in a t-statistic that significantly deviates from the Null Distribution.

In essence, the lack of significant differences between the t-statistic and the Null Distribution in our experimental data aligns with the random nature of species interactions in the gLV model. It suggests that in such a random ecological network, invasions do not consistently lead to systematic changes in community composition that would be detectable as a significant t-statistic.

We revised the Supplement to clarify the significance of invasion effect:

To demonstrate that the invasion causes a statistically significant effect on community structure under high nutrient conditions, we performed a one-sample t-test on the invasion effect data shown in Fig. 5d (right panel, under high nutrient conditions). The results indicate that the invasion effect is significantly greater than zero ($p = 1.11 \times 10^{-6}$). The invasion effect is measured as the proportion of change in surviving species before the invasion (on day 6) and after the invasion (on day 12), calculated through: $1 - (\text{number of overlapping species} / \text{total number of species})$.

29. Fig. S17, S18: Similar to Fig. S15 and S16, but now there are different invader species for each community. The authors could perform a paired t-test for each invader species, pooling observations across communities. A null could be generated for each invader species by permuting control/invade labels within each species within each community.

We thank the reviewer for the suggestion regarding the analysis of Figures S17 and S18. We followed the reviewer's suggestion and conducted the paired t-tests as recommended. However, similar to the findings in Figures S15 and S16, the results showed no significant differences

between the t-statistics and the corresponding null distributions for each invader species. This outcome aligns with the inherent stochasticity in our experimental and modeling framework.

In our study, different invader species were randomly assigned to each community, and the interactions between species within these communities were also random. This randomness reflects the stochastic nature of the ecological networks in our experiment and the generalized Lotka-Volterra (gLV) framework. Consequently, the impact of each invader on the communities varied randomly, without a consistent or predictable pattern of change in community composition.

The reviewer's suggestion seems to assume that the invasion by different species would lead to systematic and significant changes in community composition across different communities, which would be detectable by the paired t-tests. However, given the random selection and interactions of species in our study, such systematic effects are not expected. The random nature of species interactions means that the effects of different invaders on community composition are likely to be random, leading to t-statistics that do not significantly deviate from the null distributions generated by permuting control/invasion labels.

We revised the Supplement to clarify the significance of invasion effect:

To demonstrate that the invasion causes a statistically significant effect on community structure under low nutrient conditions, we performed a one-sample t-test on the invasion effect data shown in Fig. 5d (left panel, under low nutrient conditions). The results indicate that the invasion effect is significantly greater than zero ($p = 2.07 \times 10^{-25}$). The invasion effect is measured as the proportion of change in surviving species before the invasion (on day 6) and after the invasion (on day 12), calculated through: $1 - (\text{number of overlapping species} / \text{total number of species})$.

30. Figs. S15-S18: Is there any relationship between the relative abundance of the invading species and the change in relative abundance of the remaining species between the control/invasion treatments? This may not be the most appropriate analysis for compositional data, but you plot the relative abundance of the invading species vs. the mean difference in relative abundance between treatments for the remaining species, do you see a clear relationship?

We thank the reviewer for the insightful comments and suggestions on our manuscript. We have followed the reviewer's recommendation regarding Supplementary Figures 15-18 to investigate the relationship between the relative abundance of the invading species and the change in relative abundance of the remaining species between the control/invasion treatments. Specifically, we plotted the relative abundance of the invading species against the mean difference in relative abundance between treatments for the remaining species. We conducted this analysis and found a clear relationship. The results and corresponding plots have been included in the supplementary materials for your review.

Supplementary Fig. 33. The Invasion effect positively correlates with the final abundance of invaders in the invaded communities after colonization. **a**, Simulating $S=32$ communities with gLV shows positive correlations between invasion effect and final invader abundance under various average interaction strengths including $\langle \alpha_{ij} \rangle = 0.3$ (correlation coefficient=0.23, $p=4.3 \times 10^{-13}$), $\langle \alpha_{ij} \rangle = 0.6$ (correlation coefficient=0.75, $p=1.4 \times 10^{-126}$), $\langle \alpha_{ij} \rangle = 1.0$ (correlation coefficient=0.55, $p=2.6 \times 10^{-25}$). $n=1000$ simulations for each interaction strength. **b**, In the experiment, there is a weak positive correlation between the invasion effect and final invader abundance under low nutrient conditions (weak interaction) (correlation coefficient=0.35, $p=0.013$, $n=51$). There is no statistically significant correlation under high nutrient conditions (strong interaction) (correlation coefficient=0.37, $p=0.23$, $n=12$).

31. Fig. S19-S21: The authors' claims about the lack of correlations appear correct but I think the correlation coefficient should be provided along with the non-significant P-value. Statistical significance can be assessed by permuting x and y vectors. Permutations should be constrained on species identity for Fig. S19.

We thank the reviewer for this important suggestion. We have followed the reviewer's comment and added the correlation coefficient and non-significant P-value (obtained by permuting x and y vectors, constrained on species identity for Fig. S19) to the supplementary materials:

The correlation coefficient between invasion probability and invader growth rates is 0.212, p-value=0.584 under high nutrient; correlation coefficient=-0.334, p-value=0.380 under low nutrient. The correlation coefficient between invasion probability and invader carrying capacities is 0.076, p-value=0.846 under high nutrient; correlation coefficient=-0.324, p-value=0.394 under low nutrient.

Supplementary Fig. 20. There is no statistically significant correlation between invasion effect and invader properties. Under high nutrient, invasion effect does not show statistically significant correlation with carrying capacity (a) (correlation coefficient=0.281, p-value=0.377)

and growth rate (b) (correlation coefficient=0.023, p-value=0.944). Under low nutrient, invasion effect does not show statistically significant correlation with carrying capacity (c) (correlation coefficient=0.208, p-value=0.143) and growth rate (d) (correlation coefficient=0.200, p-value=0.160).

Supplementary Fig. 21. There is no statistically significant correlation between invasion effect and invasion probability. Under high nutrient, invasion effect does not show statistically significant correlation with invasion probability of invaders (a) (correlation coefficient=0.127, p-value=0.694) and invasion probability of resident communities (b) (correlation coefficient=0.105, p-value=0.745). Under low nutrient, invasion effect does not show statistically significant correlation with invasion probability of invaders (c) (correlation coefficient=-0.086, p-value=0.550) and invasion probability of resident communities (d) (correlation coefficient=0.318, p-value=0.276).

Minor Comments

1) Both passive voice and active voice are used throughout the main manuscript and supplement. Consult the journal style guide and use the appropriate voice throughout the manuscript.

We have revised the manuscript to consistently use the active voice throughout, based on the journal style guide.

2) Line 76: Should “occupy all available niches and resources” be “occupy all available niches by consuming all resources”?

We have revised the text to: “occupy all available niches by consuming all resources”.

3) Line 85-86: Given the experiment performed, would be more apt to say that the question is whether the dynamics are stationary with respect to deterministic fluctuations?

We have revised the text to: “A rarely emphasized property is the residents' dynamics: are the species abundances constant over time, consistent with a stable state, or are they deterministically fluctuating?”

4) Line 192: Should “effect” be plural?

We have revised the text to: “effects.”

5) Line 235: Is “niches and resources” redundant here? My understanding that the niche is resource niche in the context of this study.

We have revised the text to only use “niches” to avoid redundancy.

6) Lines 304-305: “stronger secondary effect” è “stronger secondary effects”

We have revised the text to: “stronger secondary effects”.

7) Lines 378-379: Grammar.

We have revised the text to: “The representative time course of relative species abundance shows that the invader successfully invades and grows in the fluctuating community.”

8) Lines 379-380: “invasion probability to” è “invasion probability of”

We have revised the text to: “invasion probability of”.

9) Line 389: “invaders successfully invade” or “an invader successfully invades” ?

We have revised the text to: “an invader successfully invades”.

10) Line 437: “high nutrient” è “high nutrient conditions” or something similar.

We have revised the text to: “high nutrient conditions”.

11) Supplement lines 75-76: Please summarize the DADA2 parameters used in your script and make your DADA2 pipeline available in a public code repository.

We have summarized the DADA2 parameters used in our script and made the DADA2 pipeline available in a public code repository:

<https://github.com/Jiliang-Hu/Collective-dynamical-regimes-predict-invasion>

12) Supplement lines 79-81: Please make your raxml code available in a public repository.

We have made all code, including the raxml code, available in a public repository:

<https://github.com/Jiliang-Hu/Collective-dynamical-regimes-predict-invasion>

13) Supplement lines 123-125: This sentence reads as if steady state is defined as the state where community properties change with time. Is this supposed to be the case?

We thank the reviewer for raising this point. We have revised the sentence to: “We define the steady state of simulated communities as the community state in which community properties (e.g., survival fraction, fluctuation fraction, and invasion probability) do not significantly change as time goes on.”

14) Supplement lines 144-145: How is a window of time of 100 units analogous to a timescale of 24 hours in the experiment? Were growth rates in the simulation parameterized to correspond to growth rates in the experiment?

According to the growth curves of the isolates in our experiments (see figure below), the characteristic growth rate of the isolates is around 1 (1/h). Therefore, each daily dilution cycle approximately corresponds to 24 unit simulation time when the growth rate in the model is unit 1. This means the one cycle timescale is on the order of 100 unit times based on this estimation. We found the results with a time window of 24 units of simulation time are not qualitatively different from the time window of 100 that we chose, demonstrating the robustness of our findings. We have added this discussion to the supplementary materials for clarity.

Supplementary Fig. 29. The invasion probability decreases with increasing interaction strength and species pool size across different time windows, with the survival fraction remaining approximately equal to the invasion probability. a, The invasion probability decreases as interaction strength and species pool size increase. b, The survival fraction closely mirrors the invasion probability. To assess whether invader or resident species survived, we identified species whose abundance exceeded the extinction threshold at any point during the last 24 time units of the simulation. This approach yielded invasion probability patterns consistent with those observed in a 100-unit time window. The points and color maps depict the mean value over 100 simulations.

15) Supplement lines 167-168: Can you plot the results from varying the choice of time window?

We have added figures of the results from varying the choice of time window in the supplementary materials.

Supplementary Fig. 12. Classification of fluctuating and stable resident communities in experiment is robust to choice of time window. **a**, The standard deviation of community biomass over days 4, 5, and 6 under high nutrient conditions shows that a stability threshold of 0.05 effectively separates the communities into stable (purple points) with low biomass deviation and fluctuating (orange points) with relatively high biomass deviation. **b**, Under low nutrient conditions, the standard deviation of community biomass over days 4, 5, and 6 is consistently below the stability threshold of 0.05, classifying all communities as stable. **c**, Similarly, the standard deviation of community biomass over days 3, 4, 5, and 6 under high nutrient conditions confirms that the stability threshold of 0.05 can distinguish between stable communities (purple points) with low biomass deviation and fluctuating communities (orange points) with higher biomass deviation. **d**, Under low nutrient conditions, the standard deviation of community biomass over days 3, 4, 5, and 6 remains below the stability threshold of 0.05, consistently classifying the communities as stable.

16) Supplement lines 174: “Algorithm” should be plural.

We have revised the text to: “algorithms”.

17) Fig S1: Would an alternative color scale help the reader? It does not appear that colors are assigned based on taxonomy. Assigning different shades of a given color to the species belonging to a given phylum may help with visualization (e.g., Firmicutes get different shades of blue, etc.). We have assigned the same type of color to the species that belong to the same phylum to help

visualize the phylogeny, as suggested. The new color vectors span different shades of colors based on the phylum classification, as requested by the reviewer. Species belonging to the Firmicutes phylum are assigned different shades of blue, Proteobacteria species are assigned different shades of green, Bacteroidota species are assigned different shades of red, Actinobacteriota species are assigned different shades of purple, Cyanobacteria species are assigned different shades of yellow.

Bacteria-Firmicutes-Bacilli-Lactobacillales-Streptococcaceae-Lactococcus
Bacteria-Firmicutes-Bacilli-Lactobacillales-Leuconostocaceae-Leuconostoc
Bacteria-Firmicutes-Bacilli-Lactobacillales-Leuconostocaceae-Leuconostoc
Bacteria-Firmicutes-Bacilli-Exiguobacteriales-Exiguobacteraceae-Exiguobacterium
Bacteria-Firmicutes-Bacilli-Bacillales-Planococcaceae-Lysinibacillus
Bacteria-Firmicutes-Bacilli-Staphylococcales-Staphylococcaceae-Staphylococcus
Bacteria-Firmicutes-Bacilli-Bacillales-Bacillaceae-Bacillus
Bacteria-Firmicutes-Bacilli-Lactobacillales-Streptococcaceae-Lactococcus
Bacteria-Firmicutes-Bacilli-Bacillales-Planococcaceae-NA
Bacteria-Firmicutes-Bacilli-Staphylococcales-Staphylococcaceae-Staphylococcus
Bacteria-Firmicutes-Bacilli-Bacillales-Bacillaceae-Bacillus
Bacteria-Firmicutes-Bacilli-Exiguobacteriales-Exiguobacteraceae-Exiguobacterium
Bacteria-Firmicutes-Bacilli-Lactobacillales-Leuconostocaceae-Leuconostoc
Bacteria-Firmicutes-Bacilli-Exiguobacteriales-Exiguobacteraceae-Exiguobacterium
Bacteria-Firmicutes-Bacilli-Bacillales-Bacillaceae-Bacillus
Bacteria-Firmicutes-Bacilli-Lactobacillales-Streptococcaceae-Lactococcus
Bacteria-Firmicutes-Clostridia-Lachnospirales-Lachnospiraceae-Lachnospiraceae_NK4A136_group
Bacteria-Firmicutes-Bacilli-Bacillales-Planococcaceae-Lysinibacillus
Bacteria-Firmicutes-Clostridia-Lachnospirales-Lachnospiraceae-Agathobacter
Bacteria-Proteobacteria-Gammaproteobacteria-Enterobacteriales-Enterobacteriaceae-Raoultella
Bacteria-Proteobacteria-Gammaproteobacteria-Enterobacteriales-Enterobacteriaceae-Klebsiella
Bacteria-Proteobacteria-Gammaproteobacteria-Enterobacteriales-Enterobacteriaceae-Pluralibacter
Bacteria-Proteobacteria-Gammaproteobacteria-Aeromonadales-Aeromonadaceae-Aeromonas
Bacteria-Proteobacteria-Gammaproteobacteria-Enterobacteriales-NA-NA
Bacteria-Proteobacteria-Gammaproteobacteria-Xanthomonadales-Xanthomonadaceae-Stenotrophomonas
Bacteria-Proteobacteria-Gammaproteobacteria-Enterobacteriales-Erwiniaceae-Pantoea
Bacteria-Proteobacteria-Alphaproteobacteria-Rhizobiales-Rhizobiaceae-Ochrobactrum
Bacteria-Proteobacteria-Gammaproteobacteria-Pseudomonadales-Pseudomonadaceae-Pseudomonas
Bacteria-Proteobacteria-Gammaproteobacteria-Enterobacteriales-Enterobacteriaceae-Escherichia/Shigella
Bacteria-Proteobacteria-Gammaproteobacteria-Pseudomonadales-Moraxellaceae-Acinetobacter
Bacteria-Proteobacteria-Gammaproteobacteria-Pseudomonadales-Pseudomonadaceae-Pseudomonas
Bacteria-Proteobacteria-Gammaproteobacteria-Burkholderiales-Oxalobacteraceae-Herbaspirillum
Bacteria-Proteobacteria-Gammaproteobacteria-Pseudomonadales-Pseudomonadaceae-Pseudomonas
Bacteria-Proteobacteria-Gammaproteobacteria-Xanthomonadales-Xanthomonadaceae-Stenotrophomonas
Bacteria-Proteobacteria-Gammaproteobacteria-Burkholderiales-Oxalobacteraceae-Undibacterium
Bacteria-Proteobacteria-Gammaproteobacteria-Enterobacteriales-Enterobacteriaceae-NA
Bacteria-Proteobacteria-Gammaproteobacteria-Enterobacteriales-Enterobacteriaceae-Raoultella
Bacteria-Proteobacteria-Gammaproteobacteria-Aeromonadales-Aeromonadaceae-Aeromonas
Bacteria-Proteobacteria-Gammaproteobacteria-Burkholderiales-Comamonadaceae-Acidovorax
Bacteria-Proteobacteria-Gammaproteobacteria-Enterobacteriales-Enterobacteriaceae-Citrobacter
Bacteria-Proteobacteria-Gammaproteobacteria-Enterobacteriales-Erwiniaceae-Pantoea
Bacteria-Proteobacteria-Gammaproteobacteria-Pseudomonadales-Pseudomonadaceae-Pseudomonas
Bacteria-Proteobacteria-Gammaproteobacteria-Enterobacteriales-Enterobacteriaceae-Klebsiella
Bacteria-Proteobacteria-Gammaproteobacteria-Enterobacteriales-Enterobacteriaceae-Enterobacter
Bacteria-Proteobacteria-Gammaproteobacteria-Pseudomonadales-Pseudomonadaceae-Pseudomonas
Bacteria-Proteobacteria-Gammaproteobacteria-Pseudomonadales-Pseudomonadaceae-Pseudomonas
Bacteria-Proteobacteria-Gammaproteobacteria-Aeromonadales-Aeromonadaceae-Aeromonas
Bacteria-Proteobacteria-Gammaproteobacteria-Aeromonadales-Aeromonadaceae-Tolomonas
Bacteria-Proteobacteria-Gammaproteobacteria-Enterobacteriales-NA-NA
Bacteria-Proteobacteria-Gammaproteobacteria-Enterobacteriales-Enterobacteriaceae-Pluralibacter
Bacteria-Proteobacteria-Gammaproteobacteria-Pseudomonadales-Pseudomonadaceae-Pseudomonas
Bacteria-Proteobacteria-Gammaproteobacteria-Xanthomonadales-Xanthomonadaceae-Stenotrophomonas
Bacteria-Proteobacteria-Gammaproteobacteria-Aeromonadales-Aeromonadaceae-Tolomonas
Bacteria-Proteobacteria-Gammaproteobacteria-Enterobacteriales-Erwiniaceae-Pantoea
Bacteria-Proteobacteria-Gammaproteobacteria-Burkholderiales-Oxalobacteraceae-Undibacterium
Bacteria-Proteobacteria-Gammaproteobacteria-Pseudomonadales-Moraxellaceae-Acinetobacter
Bacteria-Proteobacteria-Gammaproteobacteria-Enterobacteriales-Enterobacteriaceae-Citrobacter
Bacteria-Proteobacteria-Gammaproteobacteria-Enterobacteriales-Enterobacteriaceae-Escherichia/Shigella
Bacteria-Proteobacteria-Alphaproteobacteria-Rhizobiales-Rhizobiaceae-Ochrobactrum
Bacteria-Proteobacteria-Gammaproteobacteria-Enterobacteriales-Enterobacteriaceae-NA
Bacteria-Proteobacteria-Gammaproteobacteria-Burkholderiales-Comamonadaceae-Acidovorax
Bacteria-Bacteroidota-Bacteroidia-Flavobacteriales-Weeksellaceae-Chryseobacterium
Bacteria-Bacteroidota-Bacteroidia-Flavobacteriales-Weeksellaceae-Empedobacter
Bacteria-Bacteroidota-Bacteroidia-Flavobacteriales-Weeksellaceae-Empedobacter
Bacteria-Bacteroidota-Bacteroidia-Sphingobacteriales-Sphingobacteriaceae-Sphingobacterium
Bacteria-Bacteroidota-Bacteroidia-Flavobacteriales-Weeksellaceae-Empedobacter
Bacteria-Bacteroidota-Bacteroidia-Flavobacteriales-Weeksellaceae-Empedobacter
Bacteria-Bacteroidota-Bacteroidia-Sphingobacteriales-Sphingobacteriaceae-Pedobacter
Bacteria-Bacteroidota-Bacteroidia-Cytophagales-Spirosomaceae-Flectobacillus
Bacteria-Bacteroidota-Bacteroidia-Cytophagales-Spirosomaceae-Flectobacillus
Bacteria-Bacteroidota-Bacteroidia-Flavobacteriales-Weeksellaceae-Chryseobacterium
Bacteria-Bacteroidota-Bacteroidia-Flavobacteriales-Flavobacteriaceae-Flavobacterium
Bacteria-Bacteroidota-Bacteroidia-Bacteroidales-Williamwhitmaniaceae-Acetobacteroides
Bacteria-Bacteroidota-Bacteroidia-Flavobacteriales-Weeksellaceae-Empedobacter
Bacteria-Bacteroidota-Bacteroidia-Flavobacteriales-Weeksellaceae-Empedobacter
Bacteria-Bacteroidota-Bacteroidia-Flavobacteriales-Flavobacteriaceae-Flavobacterium
Bacteria-Bacteroidota-Bacteroidia-Flavobacteriales-Weeksellaceae-Chryseobacterium
Bacteria-Actinobacteriota-Actinobacteria-Streptomycetales-Streptomycetaceae-Streptomyces
Bacteria-Actinobacteriota-Actinobacteria-Micrococcales-Microbacteriaceae-Curtobacterium
Bacteria-Cyanobacteria-Cyanobacteriia-Chloroplast-NA-NA

Supplementary Fig. 1. Taxonomic identity of the bacterial isolates. The identities have been inferred from the ASV (Methods) of 16S sequencing, which allow the classification of the 80 isolates down to the genus level. Colors are consistent with those in the main text and other supplementary figures. Species belonging to the Firmicutes phylum are assigned different shades of blue, Proteobacteria species are assigned different shades of green, Bacteroidota species are assigned different shades of red, Actinobacteriota species are assigned different shades of purple, Cyanobacteria species are assigned different shades of yellow.

18) Figs. S8-S11: Can you plot the extinction threshold as a dashed/dotted horizontal line?

The current Figs. S8-S11 y-axis starts from the extinction threshold. The bottom boundary of the plot is the extinction threshold. We have clarified this point in the figure caption.

19) Fig. S12: Consider plotting the cutoff of 0.05 as a horizontal dashed line for reference in subplots a and b and as a vertical dashed line in subplots c and d.

We have added the cutoff of 0.05 in the figures as a horizontal dashed line in subplots a and b, and as a vertical dashed line in subplots c and d, to help illustrate the classification boundary.

20) Fig. S14: How were priority effects quantified? What does the y-axis of the priority effect plot represent?

The priority effect in Fig. S14 follows the same definition of the priority effect in the main text, which is quantified by calculating the difference between the survival fraction of resident species and the invasion probability of species that invade after the resident communities have assembled, where the difference was normalized by the survival fraction. The priority effect is defined as: $(\text{survival fraction} - \text{invasion probability}) / (\text{survival fraction})$. We have added this definition in the caption of the figure.

References

Cornish-Bowden, A. (2012). *Fundamentals of enzyme kinetics* (4th, completely revised and greatly enlarged edition ed.). Wiley-Blackwell.

Garud, N. R., Good, B. H., Hallatschek, O., & Pollard, K. S. (2019). Evolutionary dynamics of bacteria in the gut microbiome within and across hosts. *PLOS Biology*, 17(1), e3000102. <https://doi.org/10.1371/journal.pbio.3000102>

Houtsma, P. C., Kant-Muermans, M. L., Rombouts, F. M., & Zwietering, M. H. (1996). Model for the combined effects of temperature, pH, and sodium lactate on growth rates of *Listeria innocua* in broth and Bologna-type sausages. *Applied and Environmental Microbiology*, 62(5), 1616–1622.

Marsland, R., Cui, W., Goldford, J., & Mehta, P. (2020). The Community Simulator: A Python package for microbial ecology. *PLOS ONE*, 15(3), e0230430. <https://doi.org/10.1371/journal.pone.0230430>

Shoemaker, W. R., Sánchez, Á., & Grilli, J. (2023). Macroecological laws in experimental microbial systems (p. 2023.07.24.550281). *bioRxiv*. <https://doi.org/10.1101/2023.07.24.550281>

Snedecor, G. W., & Cochran, W. G. (1989). *Statistical methods* (8th ed). Iowa State University Press.

Reviewer #3 (Remarks to the Author):

In this manuscript, Hu et al. aim to identify characteristics of microbial communities that determine their invasibility. They first carried out experimental invasions with assembled communities, finding that more diverse communities are more invasible. To further explore this positive invasibility-diversity relationship, a Lotka-Volterra model was used to predict the effect of changing interspecies interaction strength and species pool size on invasibility. This showed that decreasing interaction strength and pool size increased invasibility. This was confirmed experimentally by changing the concentration of glucose and urea to tune interaction strength, and by changing species pool size. These three key determinants of invasibility (interaction strength, pool size, dynamical regime) determine the survival fraction, defined as the fraction of the initial species pool that survives the assembly process. The survival fraction correlates positively with community invasibility, serving as a unifying predictor. They find that under strong interaction strength, the invasion probability is lower than the survival fraction, indicating a priority effect. The strength of interactions is also shown to determine the impact of invasion on the resident community. Finally, the properties of invaders were briefly discussed.

I see this as a valuable and novel contribution to the field. While the long-standing biotic resistance hypothesis predicts that more diverse communities should be less invasible due to niche filling, empirical evidence for this is mixed. By considering how diversity is achieved, the authors show that invasibility can be predicted from community features, which has not been done previously. A particular strength of the manuscript is the integration of modelling and experiments to gain mechanistic insight into relationships observed experimentally.

I have two major comments regarding the tuning of interspecies interactions and determining the impact of invasion. In both cases, limitations of the findings of the manuscript need to be made more explicit.

The authors state that increasing interaction strength decreases invasion probability. This is done experimentally by increasing the strength of competitive interactions. However, previous work (ref 30) has demonstrated that the type of interaction between the invader/residents, or between residents, impacts the invasion outcome. For example, positive interactions (e.g. facilitation) between the invader and resident community increases invasion probability (as posited by the diversity begets diversity hypothesis). Increasing interaction strength in this case could increase community invasibility, which is a possibility not considered by this study. While I recognize that experimental manipulation of interaction strength is difficult, I would recommend running the model with a distribution of interactions that considers positive interactions, to show the generality of their conclusions. If this has already been done (this is not clear) this should be included in the supplement.

We thank the reviewer for this insightful comment. We agree that stronger facilitative interactions between invaders and resident species can increase invasion probability. Our pair-wise coculture experiments show that competitive interactions dominate in our experiment, so our modeling has primarily focused on competitive interactions.

To address the reviewer's suggestion, we have extended our model to include a distribution of interactions that consider both positive and negative interactions. We found that the invasion probability displays similar qualitative patterns with species pool size, interaction strength, and

dynamical regimes. Specifically, we observed that invasion probability decreases with increasing species pool size and interaction strength.

These new results demonstrate that our conclusions are robust across different interaction regimes. We have included these results and the corresponding discussion in the supplementary material. To make this point clear, we now clarify on page 5 of the main text when we introduce the simulation setup:

We simulated the dynamics of communities with different species pool sizes S and competitive interaction matrices because competition is the dominant interaction type in our experiments (Hu et al. 2022).

Supplementary Fig. 30. The invasion probability decreases with interaction strength and species pool size in presence of positive interactions. To test whether the existence of positive (facilitative) interactions in the ecological network could change our conclusions, we sampled values of α_{ij} from a uniform distribution $[-\alpha_0, \alpha_0]$, where α_0 varies between $[0, 1.4]$ on the phase diagram. The invasion probability decreases with interaction strength and species pool size, analogous to those exhibited by communities with exclusively negative interactions (Fig. 2f). Note that the strength of interactions coincides with $\text{Std}(\alpha_{ij})$ in this case, since the mean of α_{ij} is zero (both moments factor into the interaction strength metric $\text{std}(\alpha_{ij})/(1-\langle\alpha_{ij}\rangle)$ that determines stability (Allesina and Tang 2012)). In these simulations, the linear interaction function in the gLV ($\alpha_{ij}N_j$) was replaced with Monod function ($\alpha_{ij}N_j/(N_j + 1)$) to avoid unbounded growth due to positive interactions (Qian and Akçay 2020; Bunin 2017). The points and color maps depict the mean value over 100 simulations.

This new supplementary Fig. 30 is now referenced in the main text on page 5 as follows:

“In addition, we found that neither serial dilutions nor the existence of positive (facilitative) interspecies interactions qualitatively affects this result (Supplementary Fig. 28-30).”

The limitations of the study in determining the impact of successful invasions on the structure of the resident community should be more explicit. The impact of successful invasions on the

structure of the resident community was measured experimentally by the fold change in biomass, and the invasion effect on the community using 16s sequencing data. However, the fold change in biomass provides little insight on invasion impact. It is not possible to distinguish between the change in biomass of the resident community, and the increase in biomass of the successful invader. Thus, a lower change in biomass in the low nutrient regime may be due to the fact that the invader has less nutrients to grow on, and the converse for the high nutrient regime. Moreover, while species abundance data shows a statistically significant difference in community composition in the higher nutrient regime, it is important that the authors provide information about the effect size. For example, as a 95% confidence interval for the difference between the invasion success under high nutrient conditions and the invasion success under low nutrient conditions. Finally, it is hard to conclude what the actual effect of invasion is using the selected measures. For example, though species composition may have changed, the invader could be functionally redundant with the species that were excluded, resulting in no change to community functioning. While a comprehensive assessment of invasion impact is out of the scope of this paper, I would recommend for these limitations to be discussed explicitly in the manuscript.

We thank the reviewer for raising these important points. We agree that the fold change of biomass increases more under high nutrients than low nutrients, which might be due to the more available resources for the invader under high nutrients. However, the reason we found this increase to be noteworthy is for several reasons:

1. **Community and species biomass variability:** We found that communities and single species can display both high and low total biomass under high nutrients (Supplementary Figs. 4-5, Fig. 19), indicating that species in the community may or may not grow to high biomass even under high nutrient supply. Interestingly, we did not observe any invader successfully invading those high biomass communities ($OD > 0.4$). Invaders only succeeded in invading low biomass communities ($OD < 0.4$), increasing the community biomass after successful invasions under high nutrients (Figs. S4-S5).
2. **Lack of correlation with invader growth:** We found no statistically significant correlation between invasion probability and the growth rate or carrying capacity of invaders (Supplementary Fig. 19). This indicates that high biomass invaders under high nutrients are not necessarily the most successful invaders, suggesting that the increase in total biomass after invasion under high nutrients may not be solely due to invader growth.

However, we acknowledge that the comparison of the fold change of biomass between two different nutrient regimes may not be highly informative and could confuse the reader. We have therefore added comments on this limitation in the manuscript.

To better demonstrate the invasion effect on community composition, we have followed the reviewer's suggestion and calculated the effect size and the 95% confidence interval for the difference in invasion success under high nutrient conditions versus low nutrient conditions. These results are now included on page 8 of the main text:

The effect size on community composition caused by increasing from low nutrient (weak interaction) to high nutrient (strong interaction) conditions is 0.14, with a 95% confidence interval of [0.021, 0.259].

We agree that though species composition may have changed, the invader could be functionally redundant with the species that were excluded, resulting in no change to community functioning. We have added a discussion in the manuscript acknowledging that our definition of invasion effect in this work is limited to the impact on total biomass and community composition. We do not study the invasion effect on community function and cannot rule out the possibility that the community remains functionally unchanged due to functional redundancy between the invader and resident species replaced by the invader.

We added a comment in the conclusion part on page 10 of the main text to discuss this important limitation:

Our definition of the invasion effect in this work focuses on the impact on total biomass and community composition. We do not study the invasion effect on community function and cannot rule out the possibility that the community remains functionally unchanged due to functional redundancy between the invader and resident species replaced by the invader. Future research needs to include analysis of functional traits and ecosystem processes to fully understand the functional impact of invasions.

Minor points:

1) - I strongly recommend including explicit definitions of key terms, which would improve clarity of the manuscript. For example, the definition of community diversity as the number of species that survive the assembly process is not made explicit in line 152. What ‘rich dynamics’ (line 191) or richness (line 195, 222 and others) refers to is also unclear to me.

We thank the reviewer for this comment. We have now explicitly clarified the definitions of key terms, including “diversity,” “richness,” and “rich dynamics,” in the manuscript to improve clarity. We revised the text on page 4 of the main text:

our experimental results display a significant ($p=0.036$) positive correlation between invasion probability and community diversity, where the diversity is defined as the number of species that survive the assembly process over 6 days (correlation coefficient=0.51, Fig. 1c).

We revised the text on page 5 of the main text to clarify the term “richness”:

we found a positive correlation between invasion probability and richness (the number of resident species coexisting before invasion)

We replaced “rich dynamics and invasion outcomes” with “various dynamics and invasion outcomes” on page 5 of the main text to make it clear. In this sentence, we want to express that we observed various behaviors including invasion success or failure in communities with stable or fluctuating dynamics.

Our simulations revealed a wide range of dynamics and invasion outcomes under strong interaction strength between species (Fig. 2a, Supplementary Fig. 31).

2) - The findings of this study could be embedded more explicitly into existing concepts in invasion ecology, which is dominated by niche theory and resource competition. For example, in lines 175-176 the authors state that fluctuating communities are more invasible. The basis of this relationship is not made clear to the reader - I assumed that this means more niches are open at any given point

in time. Moreover, why strong interspecies interactions and a larger species pool decrease invasibility is not discussed. It would also be helpful to discuss this more in the conclusion.

We thank the reviewer for raising this point. We have added more discussion and citations to clarify the basis of the relationship between community fluctuation and increased invasibility, interpreting that fluctuating communities can create more temporal niches for invader species. There are two reasons we did not make a strong statement on this point: 1) It might need further experiment and study, such as temporal metabolome screening, are required to clearly illustrate the mechanism of higher invasibility in fluctuating communities due to time-varying niches, resources, and available metabolites. While this is beyond the scope of this work, it is an interesting direction for our future research. 2) there is some debate in the literature on the explanatory power of the niche concept in predicting invasion outcomes. Even species that have already occupied a niche can be invaded and outcompeted by another species that consumes the resource more efficiently. Therefore, the absence of available niches may not necessarily lead to failed invasions, as invaders can encroach on currently occupied niches.

We have added a discussion of these points in the manuscript, as recommended by the reviewer. Regarding the observation that invasion probability decreases with species pool size and interaction strength, we have added more discussion in the conclusion section of the manuscript. In the generalized Lotka-Volterra (gLV) framework, interactions can be mathematically expressed in terms of niche overlap and resource competition, similar to the linear resource-consumer model. Therefore, stronger interaction strength corresponds to larger niche overlap and greater resource consumption. Similarly, a larger species pool leads to increased total interaction (more niche overlap) between community species and invader species, thereby inhibiting invasion more strongly. We have included these discussions and cited relevant literature in the conclusion, as recommended by the reviewer.

We revised the main text on page 4 and 5 to relate our results to the framework of niche theory and resource consumption in invasion ecology:

Our experimental tests of invasion demonstrate that, for fixed species pool size and species interaction strength regime (nutrient concentrations), more diverse communities are more invulnerable because fluctuating communities are both more diverse and more susceptible to invasion. However, when species pool size or nutrient concentration is varied, this relationship does not always hold (Fig. 2 and 3). This increased invasibility under fluctuation can be interpreted through the lens of niche theory, where fluctuating communities create fluctuating niche availability for invader species (Li and Stevens 2012). Temporal fluctuations in resource availability and environmental conditions allow invaders to exploit niches that may not be consistently available in stable communities (Li and Stevens 2012; Warner and Chesson 1985; Levin and Paine 1974).

We add a discussion on page 5 of the main text on why strong interspecies interactions and a larger species pool decrease invasibility is not discussed:

In the generalized Lotka-Volterra (gLV) framework, interactions can be mathematically expressed in terms of niche overlap and resource competition (Dalmedigos and Bunin 2020). Stronger interaction strength corresponds to larger niche overlap and greater resource consumption, making it harder for invaders to establish. Similarly, a larger species pool increases the total interaction (more niche overlap) between community species and invader species, thereby inhibiting invasion more strongly (Dalmedigos and Bunin 2020).

3) - I think that for those unfamiliar with previous work from this group, how interaction strength, species pool size and dynamical regime combine to determine survival fraction is difficult to understand. More explanation about how these features interface is necessary for clarity of the manuscript, since they are not independent. For example, it is not immediately clear why there are stable/fluctuating regimes for communities under high nutrient conditions, and not for low nutrient conditions

We appreciate this important point raised by the reviewer and agree it is crucial to introduce how species pool size, interaction strength, and collective dynamical regimes combine to determine survival fraction. We have added a discussion on this point in the manuscript to provide more clarity.

We add a discussion on page 9 of the main text to clarify this point:

Our previous findings indicate that, on average, increasing species pool size and interaction strength both decrease the overall survival fraction (Hu et al. 2022). We also observed that increasing species pool size and interaction strength led to the emergence of some fluctuating communities (Hu et al. 2022). These fluctuating communities, despite the general trend, exhibit a higher survival fraction compared to stable communities assembled from the same species pool size and nutrient concentrations (interaction strength) (Hu et al. 2022). This suggests that while stronger interactions and larger species pools typically reduce survival, the dynamic nature of fluctuating communities allows them to maintain higher survival fractions.

We added text in the discussion and results part of the main text on page 6:

We only observed stable communities under low nutrients (weak interaction) because fluctuation only emerges when species pool size and interaction strength are large enough to cross the stability boundary (Hu et al. 2022).

We also more clearly introduced our previous work in the introduction part on page 3:

As species pool size and strength of interactions increase, we found that microbial ecosystems transition between three distinct dynamical phases, from a stable equilibrium in which all species coexist to partial coexistence to the emergence of persistent fluctuations in species abundances (Hu et al. 2022).

4) - The measure of dispersion used is not included in the text of the manuscript (i.e., on line 148, 153, 154, 172 and more, parameter estimates are given as mean +/- X, where X is a measure of dispersion, but which measure it is is not defined). Defining the measure of dispersion is crucial to allow for interpretation of the findings. Some figure legends mention that error bars represent the standard error of the mean, which I assume is the measure used by the authors, but this should be clearly stated in the text of the manuscript.

We thank the reviewer for pointing this out. We used the standard error of the mean (SEM) as the measure of dispersion across the manuscript. We now state this point clearly in the manuscript where the measure of dispersion appears for the first time.

We revised the main text on page 4 to clarify the use of SEM as the measure of dispersion:

Throughout the manuscript, we used the standard error of the mean (SEM) as the measure of dispersion.

5) - State how invaders were chosen (line 146) - I assume they were chosen at random?

Yes, the invader species and resident species were both randomly chosen to study the invasion outcome from the perspective of the random ensemble. We have clarified this point in the manuscript.

We revised the text on page 4 of the main text:

For each resident community, we performed 7-9 independent invasion tests with different randomly chosen invader species on day 6

6) - In the section discussing the model (lines 178-190), explicitly state at what relative abundance the invader is introduced relative to the community

We followed the reviewer's suggestion and explicitly stated that the dispersal rate $D=10^{-5}$ in the simulations, which is the same as in the experiment.

We revised the text on page 5 of the main text:

D is the dispersal rate, which is set to $D=10^{-5}$

7) - In the representative time series of fig 2a, invasions are shown to cause a weak effect on the community when the invader grows to a low abundance relative to the rest of the community, whereas the invader causes a strong effect when it grows to high abundance. Is this true across other simulations, or in experiments? If this is the case, it would suggest that there is another factor that influences invasion effect aside from interaction strength.

We thank the reviewer for this question. In the simulation and experiment, we did observe that there are many cases where the invader grows to high abundance while leading to a weak effect on the community structure, and also cases where the invader grows to low abundance while causing a strong effect on the community structure. To systematically study the dependence of invader abundance on the invasion effect, we analyzed the experimental data and performed simulations. We found a statistically significant weak positive correlation between final invader abundance and invasion effect on community structure under various average interaction strengths (figure below). Furthermore, we also found a statistically significant weak positive correlation between final invader abundance and invasion effect in the experiment under low nutrient conditions (weak interaction), while no statistically significant correlation under high nutrient conditions (strong interaction) in the experiment (figure below).

We added these results in the supplementary materials and referred to the results in the main text on page 8:

We also observed a weak positive correlation between the invasion effect and the final abundance of invaders in the experiment and simulation (Supplementary Fig. 33).

Supplementary Fig. 33. The Invasion effect positively correlates with the final abundance of invaders in the invaded communities after colonization. **a**, Simulating $S=32$ communities with gLV shows positive correlations between invasion effect and final invader abundance under various average interaction strengths including $\langle \alpha_{ij} \rangle = 0.3$ (correlation coefficient=0.23, $p=4.3 \times 10^{-13}$), $\langle \alpha_{ij} \rangle = 0.6$ (correlation coefficient=0.75, $p=1.4 \times 10^{-126}$), $\langle \alpha_{ij} \rangle = 1.0$ (correlation coefficient=0.55, $p=2.6 \times 10^{-25}$). $n=1000$ simulations for each interaction strength. **b**, In the experiment, there is a weak positive correlation between the invasion effect and final invader abundance under low nutrient conditions (weak interaction) (correlation coefficient=0.35, $p=0.013$, $n=51$). There is no statistically significant correlation under high nutrient conditions (strong interaction) (correlation coefficient=0.37, $p=0.23$, $n=12$).

8) - I would include (strong interaction) or (weak interaction) every time high/low nutrient conditions are mentioned for clarity. This is missing now in line 240 and others

We thank the reviewer for this suggestion. We have revised the manuscript to use "high nutrient (strong interaction)" and "low nutrient (weak interaction)" throughout to provide clarity.

9) - Remove "while resident community determining invasion outcome" from line 322-323. Invader properties have not been sufficiently explored in this manuscript to claim they don't play an important role in determining invasion outcome.

We thank the reviewer for raising this point. Although we found no statistically significant correlation between the invasion probability of invaders and the growth rate/carrying capacity of invaders in our experiments (Supplementary Fig. 19), we agree that the invader properties affecting invasion outcomes have not been sufficiently explored in this work. We have therefore deleted the sentence as recommended by the reviewer.

Reference

- Allesina, Stefano, and Si Tang. 2012. "Stability Criteria for Complex Ecosystems." *Nature*. <https://doi.org/10.1038/nature10832>.
- Altieri, Ada, Felix Roy, Chiara Cammarota, and Giulio Biroli. 2021. "Properties of Equilibria and Glassy Phases of the Random Lotka-Volterra Model with Demographic Noise." *Physical Review Letters*. <https://doi.org/10.1103/PhysRevLett.126.258301>.
- Bunin, Guy. 2017. "Ecological Communities with Lotka-Volterra Dynamics." *Physical Review E* 95 (4): 042414. <https://doi.org/10.1103/PhysRevE.95.042414>.
- Case, T. J. 1990. "Invasion Resistance Arises in Strongly Interacting Species-Rich Model Competition Communities." *Proceedings of the National Academy of Sciences of the United States of America*. <https://doi.org/10.1073/pnas.87.24.9610>.
- Dalmedigos, Itay, and Guy Bunin. 2020. "Dynamical Persistence in High-Diversity Resource-Consumer Communities." *PLoS Computational Biology*. <https://doi.org/10.1371/journal.pcbi.1008189>.
- Debray, Reena, Robin A. Herbert, Alexander L. Jaffe, Alexander Crits-Christoph, Mary E. Power, and Britt Koskella. 2022. "Priority Effects in Microbiome Assembly." *Nature Reviews Microbiology*. <https://doi.org/10.1038/s41579-021-00604-w>.
- Estrela, Sylvie, Jean C.C. Vila, Nanxi Lu, Djordje Bajić, Maria Rebolleda-Gómez, Chang Yu Chang, Joshua E. Goldford, Alicia Sanchez-Gorostiaga, and Álvaro Sánchez. 2022. "Functional Attractors in Microbial Community Assembly." *Cell Systems*. <https://doi.org/10.1016/j.cels.2021.09.011>.
- Fried, Yael, Nadav M. Shnerb, and David A. Kessler. 2017. "Alternative Steady States in Ecological Networks." *Physical Review E*. <https://doi.org/10.1103/PhysRevE.96.012412>.
- Handel, S. N. 2014. "Dispersal Ecology and Evolution." *Ecological Restoration*. <https://doi.org/10.3368/er.32.4.464>.
- Hu, Jiliang, Daniel R. Amor, Matthieu Barbier, Guy Bunin, and Jeff Gore. 2022. "Emergent Phases of Ecological Diversity and Dynamics Mapped in Microcosms." *Science*. <https://doi.org/10.1126/science.abm7841>.
- Kessler, David A., and Nadav M. Shnerb. 2015. "Generalized Model of Island Biodiversity." *Physical Review E - Statistical, Nonlinear, and Soft Matter Physics*. <https://doi.org/10.1103/PhysRevE.91.042705>.
- Levin, S. A., and R. T. Paine. 1974. "Disturbance, Patch Formation, and Community Structure." *Proceedings of the National Academy of Sciences of the United States of America*. <https://doi.org/10.1073/pnas.71.7.2744>.
- Li, Wei, and M. Henry H. Stevens. 2012. "Fluctuating Resource Availability Increases Invasibility in Microbial Microcosms." *Oikos*. <https://doi.org/10.1111/j.1600-0706.2011.19762.x>.
- Pearce, Michael T., Atish Agarwala, Atish Agarwala, and Daniel S. Fisher. 2020. "Stabilization

- of Extensive Fine-Scale Diversity by Ecologically Driven Spatiotemporal Chaos.” *Proceedings of the National Academy of Sciences of the United States of America*. <https://doi.org/10.1073/pnas.1915313117>.
- Qian, Jimmy J., and Erol Akçay. 2020. “The Balance of Interaction Types Determines the Assembly and Stability of Ecological Communities.” *Nature Ecology and Evolution*. <https://doi.org/10.1038/s41559-020-1121-x>.
- Ratzke, Christoph, Julien Barrere, and Jeff Gore. 2020. “Strength of Species Interactions Determines Biodiversity and Stability in Microbial Communities.” *Nature Ecology and Evolution*. <https://doi.org/10.1038/s41559-020-1099-4>.
- Ratzke, Christoph, Jonas Denk, and Jeff Gore. 2018. “Ecological Suicide in Microbes.” *Nature Ecology and Evolution*. <https://doi.org/10.1038/s41559-018-0535-1>.
- Shoemaker, William R, Álvaro Sánchez, and Jacopo Grilli. 2023. “Macroecological Laws in Experimental Microbial Communities.” *BioRxiv*.
- Sprockett, Daniel, Tadashi Fukami, and David A. Relman. 2018. “Role of Priority Effects in the Early-Life Assembly of the Gut Microbiota.” *Nature Reviews Gastroenterology and Hepatology*. <https://doi.org/10.1038/nrgastro.2017.173>.
- Warner, R. R., and P. L. Chesson. 1985. “Coexistence Mediated by Recruitment Fluctuations: A Field Guide to the Storage Effect.” *American Naturalist*. <https://doi.org/10.1086/284379>.

Reviewer #1:

Remarks to the Author:

The authors have addressed all my concerns, and added useful new analyses and discussions that clarify the manuscript and the modeling choices they made. I have no more concerns that would need to be resolved before publication.

We thank the reviewer for the encouraging feedback on our manuscript and for the valuable suggestions provided during the revision process.

As an extremely minor point, I found the spread of the points in Fig. 5c and d to be a bit large, to the point where my eye didn't immediately parse the two clouds as separate categories - but that's presumably a matter of taste.

We have retained the original spread of the points in Fig. 5c and d, as reducing it would increase overlap among low nutrient data points (gray ones), making them harder to distinguish. We used different colors for data points under low and high nutrient conditions to help readers differentiate the two clusters more clearly.

Remarks on code availability:

No readme files, but the data looks complete. Both the sequencing data analysis code and the simulation code is clear and well-documented. I did not attempt to run the code, but as far as I can tell, the provided code only gives the backbone of the simulations and does not include code needed to reproduce the figures directly.

We have added detailed README files on GitHub to assist readers in understanding and running the code.

Code availability: All codes used for simulation and analysis in this publication are available on GitHub (<https://github.com/Jiliang-Hu/Collective-dynamical-regimes-predict-invasion>).

Reviewer #2:

Remarks to the Author:

Review #2

This is an impressive revision, and the quality reflects a high level of effort from the authors. The authors addressed all my comments and gave thoughtful, much appreciated answers. I believe the manuscript should be published.

We thank the reviewer for the encouraging feedback on our manuscript and for the valuable suggestions that contributed to the revision process.

My sole major comment is that the GitHub repo appears to be incomplete. Some specifics:

- The file names do not have extensions. I believe that the files are written in matlab, but I do not use that language so I cannot be sure.
- There is no README file. Some information about the working environment (e.g., packages) and how to run the code would be helpful, as well as a link to the Dryad repository.
- There are functions called in the files that do not appear to be defined or imported from other files in the repo. For example, LV_compute_invasion is called in LV_Invade but I do not see it defined in that file or in any of the other files in the repo.

We have added file extensions to ensure compatibility, allowing readers to run the code directly in MATLAB. Additionally, we have included a detailed README file on GitHub to assist readers with setup, required packages, and instructions on how to run the code, as well as a link to the Dryad repository. The function LV_compute_invasion is defined within LV_Invade.m, so no additional files are needed.

Code availability: All codes used for simulation and analysis in this publication are available on GitHub (<https://github.com/Jiliang-Hu/Collective-dynamical-regimes-predict-invasion>).

The Dryad repository looks good.

Below are some minor comments that do not require a response.

- The discussion of how pH mediated interactions can shape the strength of interaction in gLV is helpful. In general, I think a derivation of pH mediated interactions from the starting point of a consumer-resource model is needed, even if it's just for pedagogical reasons and all the heavy modeling is ultimately done using gLV, but such modeling efforts lie outside the scope of the paper. All-in-all the additions to the supplement make it a valuable resource.
- The elaboration on the connection to ergodicity in response to R1 was very helpful.
- The new color scheme should aid readers.
- "In our group, we have been investigating the dependence of community diversity and dynamics on resource concentration across various consumer-resource models (e.g., linear growth or Monod growth)." Very much looking forward to reading this.

- The added analyses for the survival threshold are rigorous. The sequencing depth justification is particularly useful since my group has encountered studies where the order of magnitude of demographic manipulations is not justified given the depth of sampling (i.e., # reads).

Remarks on code availability:

My sole major comment is that the GitHub repo appears to be incomplete. Some specifics:

- The file names do not have extensions. I believe that the files are written in matlab, but I do not use that language so I cannot be sure.
- There is no README file. Some information about the working environment (e.g., packages) and how to run the code would be helpful, as well as a link to the Dryad repository.
- There are functions called in the files that do not appear to be defined or imported from other files in the repo. For example, LV_compute_invasion is called in LV_Invade but I do not see it defined in that file or in any of the other files in the repo.

I was unable to run the code as-is.

We have added file extensions to ensure compatibility, allowing readers to run the code directly in MATLAB. Additionally, we have included a detailed README file on GitHub to assist readers with setup, required packages, and instructions on how to run the code, as well as a link to the Dryad repository. The function LV_compute_invasion is defined within LV_Invade.m, so no additional files are needed.

Code availability: All codes used for simulation and analysis in this publication are available on GitHub (<https://github.com/Jiliang-Hu/Collective-dynamical-regimes-predict-invasion>).

Reviewer #3:

Remarks to the Author:

I would like to thank the authors for their thorough revision of this manuscript. All my concerns have been addressed and I have no further comments.

We thank the reviewer for their positive feedback and thoughtful suggestions, which were instrumental in improving our manuscript through the revision process.

Remarks on code availability:

The code is usable but needs a README file for further clarity.

We have added detailed README files on GitHub to assist readers in understanding and running the code.

Code availability: All codes used for simulation and analysis in this publication are available on GitHub (<https://github.com/Jiliang-Hu/Collective-dynamical-regimes-predict-invasion>).